# Sustainable Preparation Behavior for Kitchen Staff in Order to Limit Food Waste

**DOI:** 10.3390/foods12163028

**Published:** 2023-08-12

**Authors:** Min-Yen Lu, Wen-Hwa Ko

**Affiliations:** 1 Program in Nutrition and Food Science, Fu Jen Catholic University, New Taipei City 242062, Taiwan; 2 Department of Restaurant, Hotel and Institutional Management, Fu Jen Catholic University, New Taipei City 242062, Taiwan

**Keywords:** surplus food, sustainable preparation behavior, food waste, kitchen staff, ethical sustainability, self-efficacy

## Abstract

The concepts of culinary sustainability and avoiding the waste of surplus food have become important sustainability trends today. How the handling of surplus food can be integrated into the catering industry is a topic of concern in the industry. Kitchen staff are the vital soul of any restaurant, and we intend to discuss how kitchen staff actually behave and explore factors that influence their behaviors in order to develop an implementation model for food waste prevention. Therefore, this study explored a model of ethical sustainability, professional competence, self-efficacy, sustainable food preparation objectives, and sustainable food promotion and behavior focused on limiting food waste, using structural equation modeling (SEM) to understand the relationship between various constructs. This study used a questionnaire and surveyed employees who had been employed for more than 6 months in Taiwan. From May to August 2022, 500 questionnaires were distributed; 415 valid questionnaires were retrieved, yielding a 90.2% recovery rate. According to the structural equation modeling analysis between the dimensions, ethical sustainability should have a positive influence on professional competence in food waste prevention and self-efficacy. Professional competence in food waste prevention affected self-efficacy and behavioral intentions during food preparation; self-efficacy also significantly affected behavioral intentions towards sustainable food preparation. Similarly, behavioral intention had a positive influence on promoting sustainable behaviors. There is a significant relationship between all constructs in this study. Professional competence in food waste prevention was found to be the mediating factor between ethical sustainability and behavioral intentions toward sustainable food preparation, and self-efficacy was the mediating factor between professional competence in food waste prevention and behavioral intentions toward sustainable food preparation.

## 1. Introduction

Food is an important daily necessity. With advancements in the global economy, equipment, and food production efficiency, an abundant supply of food can provide for more people. From production to consumption, food is often discarded by consumers even though it is safe to eat because they have bought too much or have insufficient knowledge about the shelf life of food. About 1.3 billion tons of food is wasted in the world every year, accounting for about 1/3 of production [1,2]. The problem of food waste has been included in the World Health Organization’s sustainable development goals (SDG 17) in order to reduce global human food waste by half and to reduce food loss by 2030 [2]. Targeting food service and family experiences has been shown to be an appropriate way to address national and international sustainable goals [1].

Due to limited resources, which affect the supply of normal diets, the problem of surplus food management has attracted great attention from academics and the industry. Another problem is the commercial and economic losses caused by waste in the food supply process, especially by those foods that are still edible, which cannot be distributed, eaten, and sold for various reasons and are eventually discarded [3].

The Food and Agriculture Organization (FAO) of the United Nations has estimated that the annual global food waste across the entire food chain is USD 2.6 trillion. The Waste and Resources Action Plan estimated that food waste in the UK totals around 10 million tons per year, 70% of which was avoidable [4]. Food waste is an urgent and serious issue in Western countries. The increase in the amount of food waste also directly affects environmental changes, including the pollution of packaging containers and greenhouse gas emissions. In Poland, 53% of food waste is produced by consumers. In order to reduce consumer food waste, it is necessary to understand the behavioral factors [5]. The key problem of food waste has been primarily due to the supply chain, from production, preparation, and processing to transportation and consumption, all of which require considerable costs and resources. Therefore, from consumers and industry players, we come to discuss the issue of supply and demand. The predicted population growth will require an adequate food system that includes the sustainable development of natural ecological resources [6,7]. Generally, the further back in the food chain, the more serious the food waste situation [8].

In restaurants, government intervention and enhanced waste disposal collection are necessary. Research has shown that changing consumer behavior, combined with corporate social responsibility, mitigates food waste [9]. The results of the study showed that many restaurants have prioritized reducing food waste. However, most of the final factors affecting food waste in restaurants were quality and specification issues during production. Generally, food waste was the worst in casual restaurants, and large meal portions resulted in more food waste [10].

The chefs are the souls of a restaurant. In addition to making good dishes, a good chef can also improve the product value while maintaining their social responsibility. Food oversupply has also been related to consumers’ perceived concept of preservation. Consumers may purchase more than they need in order to save time [11,12]. Among these food behaviors, discarding reusable ingredients as kitchen waste was the most severe (28.8% of ingredients were considered kitchen waste and discarded) [13]. Storing and sorting food in an orderly manner (for example, stacking old and new food or sorting by frequency of use) and reordering regularly can reduce food waste [14,15]. According to Secondi, Principato, and Laureti [16], one of the best courses of action to avoid wasting food is to estimate portion sizes correctly. Misplaced, forgotten, or expired leftovers are an issue when storing food [17].

Social cognitive theory (SCT) explains human behavior based on the dynamic interactions of individual experiences, the actions of others, and environmental influences [18]. Self-efficacy is the belief in one’s ability to change motivations, cognitive resources, and courses of action in order to meet the needs of a particular situation [19,20,21,22]. Research has found that self-efficacy predicted several job-related outcomes, such as work attitudes, training proficiency, and job performance. Bandura [23] pointed out that self-efficacy is the result of interactions with the external environment, personal and professional competency, and learning effectiveness, which can then motivate behavior. Kitchen staff play a key role in food production from the kitchen to the table, and a sustainability-focused food revolution should be initiated to reduce carbon emissions and mitigate food waste to preserve the planet’s ecosystem.

Therefore, the attitudes of kitchen staff in avoiding food waste can affect how they handle food, and sustainable cooking can provide substantial results in the future for environmental and food sustainability. This study explored the perspectives of kitchen staff towards limiting food waste by evaluating behavioral patterns such as ethical sustainability, professional competence in food waste prevention, self-efficacy, behavioral intentions towards sustainable food preparation, and sustainable food promotion from a practical perspective. This research can help promote sustainable cooking and food preparation, from the individual to the restaurant and then to the catering industry. By encouraging consistent sustainability in the catering industry, we can more effectively reduce food waste.

## 2. Theoretical Background and Hypotheses Development

### 2.1. Surplus Food and Food Waste

A food surplus distinguishes between the concepts of availability, surplus, recyclability, and waste (ASRW) to distinguish it from food waste. Available food is described as “all food in the food supply chain”, which consists of human consumption, surplus food, and food scraps. There is no generally accepted definition of food waste and leftovers because there have been different interpretations from different perspectives [24].

According to the FAO, food waste consists of three distinct problems. Food loss (FL) is the loss of original food weight or nutritional value. These losses were primarily due to processes in the supply chain, such as poor management, errors, and irregular processes (such as improper farming and harvesting), resulting in reduced edible weight. Food waste (FW) is the discarding of edible food, whether expired or spoiled, for reasons such as oversupply, not having a proper sales strategy, or excessive or unplanned purchases by individual consumers. It is also the waste of edible food due to improper distribution, storage, transportation, and preparation, both in the home and in restaurants [25].

Moreover, food waste occurs along the food chain, including production, distribution, retail, and consumption [26]. Types of waste at all stages of the food chain are shown in Figure 1. Wasted food is any food that is lost due to spoilage or being discarded. “Waste food”, as opposed to “food waste”, refers to food that can further serve a valuable function [2].

According to scholars [27], surplus food can be divided into four categories: (1) Unprofitable crops refer to the market phenomenon after the harvest of agricultural products, and the surplus of grain sales is generated due to a stable sale price. When production is much higher than demand, it can lead to an oversupply in the market. (2) Non-perishable processed foods are foods with a long shelf life (such as dry goods, canned foods, etc.), damaged packaging, incorrect packaging, promotions near the expiration date, etc. (3) Perishable fresh foods are fresh fruits and vegetables, bread, dairy products, refrigerated ready-to-eat foods, etc.; products that are not sold because they are close to their expiration date but are still safe to eat; and foods that agricultural producers and importers may not be able to sell because the fruit or vegetables are irregular in shape and size. (4) Perishable foods include ready-to-eat foods such as sandwiches, cakes, and pastries [27].

### 2.2. Social Cognitive Theory

Social cognitive theory (SCT) was first proposed by Bandura, who combined the concepts of behaviorism and social learning [18]. Behavior is generated by the interaction between the individual and the environment rather than determined by any single aspect. These factors include environmental impacts (the overall social environment, organizational policies, and culture), individual cognition and personal factors (personal motivation, attitude), and behavior (intention and willingness), all of which interact. Figure 2 shows the dynamic formation of the person, the environment, and behavior in SCT, where personal behavior is affected by the environment and influenced by personal subjective cognition. The psychological process of self-introspection is achieved through the inner driving force of the individual, and the person feels and reacts to their external environment due to their inner thoughts rather than being stimulated by the environment alone. Therefore, this research hypothesized that environmental factors can influence personal factors, and personal factors can influence behavioral factors.

Bandura [28] suggested that persons with high self-efficacy can have more enthusiasm, broader vision, broader thinking, and, therefore, courage to accept challenges. To avoid repeating the failures of others, most individuals use the success of others as a guide. Rimal [29] also pointed out that SCT is an important basis for predicting food handling processes. Intention and self-efficacy can predict food-handling behavior [30]. Bandura [23] also suggested that personal past performance affected self-efficacy, and higher self-efficacy can also affect future performance. Behavioral improvements can be enhanced by offering continuous reminders during food handling [31].

Bandura [23] considered SCT the most suitable for explaining dynamic human behavior and regarded cognition and self-regulation as the primary framework of causality. The development of this model can cultivate the confidence of individuals to use certain abilities. However, it can also be affected by many internal factors, such as the level of self-efficacy, which will then result in different behaviors. Self-efficacy refers to a person’s belief in their own capabilities and their degree of confidence in being able to achieve a specific outcome.

### 2.3. Ethical Sustainability Affects Personal Behavior

Waston and Meah [32] indicated that even if consumers had an environmental motivation to avoid food waste, the real motivation to reduce food waste is the “ethic of thrift”. People avoid wasting food because it causes feelings of guilt that are then combined with environmental concerns [17]. Food represents the physical and symbolic connection between humans and nature [33]. A strong negative attitude towards food waste can be explained by our respect for nature, and this can hopefully be extended to the community [34].

Therefore, food waste can be perceived as a waste of precious resources, which would be unfair from an anthropocentric perspective [35]. Food waste is considered immoral, so people need to be called upon to change their behaviors and attitudes concerning food waste [36,37]. Therefore, food is a valuable resource, as it is not always readily available. A change in attitude can benefit both humans and nature by reducing the negative aspects of food production and improving the balance between humans and nature [38]. The “moral” rule for food waste behavior can be “don’t waste food”. With this in mind, our core ethical commitment is to avoid wasting food through more sensible consumption. Doing so could protect and preserve the world in which we live [39].

As indicated by the self-perception theory, previous behaviors affect future actions. The theory suggests that if individuals are encouraged to behave in environmentally conscious ways, they are more likely to make environmentally conscious decisions [40,41]. This theory reinforces environmental self-identity, or an individual’s perception of themselves as a person who is environmentally conscious [42]. In addition, many previous studies found that social norms and the behaviors of social groups had a positive influence on behavior [33,43].

Therefore, the following hypotheses were developed:

**H1:** 
*Attitudes towards ethical sustainability affect professional competence in food waste prevention.*


**H2:** 
*Attitudes towards ethical sustainability affect self-efficacy.*


### 2.4. Relationships among Professional Competence, Self-Efficacy, and Food Waste Prevention

There are main professional competencies, including food knowledge, skills, communication skills, business concepts, and food knowledge for kitchen staff [44]. The four major fields of culinary courses include culinary arts, food science, life science, business and information [45]. The skills of a chef are food science, product preparation, finance, personal management, marketing, language, and computer skills [46]. Professional competence in food waste prevention includes two aspects: hygiene knowledge and culinary knowledge. Professional attitudes towards food waste prevention include three dimensions, namely personal moral attitudes, attitudes towards food ingredients, and education and training. Skill competence comprises two dimensions: dish analysis and meal planning management [47]. In the restaurant and hospitality industry, both food preparers and consumers are responsible for food waste. Food waste has also been identified as a problem in schools, convention stores, bars, restaurants, canteens, and hotels [48].

The concept of self-efficacy first appeared in an article by Bandura [20]. It refers to an individual’s belief of whether they can successfully complete a task. The emphasis was on ability, which can promote individual actions in certain areas and affect future activities. This aspect is related to personal values. Bandura [18] also defined self-efficacy as “in order to deal with an impending situation, an individual makes a subjective judgment on whether he or she can successfully perform the required behavior. This judgment will decide how much effort he or she will put in when he or she personally faces difficulties, and how long it will last”. That is, self-efficacy can affect whether a person will persist in an action after setting goals. This is also related to whether a person has the motivation to overcome obstacles in the learning process and the learning strategies adopted when solving problems.

There are some research studies between perceived behavioral control and self-efficacy; the theory of reasoned action (TRA) [49,50] and the theory of planned behavior (TPB) models [51] were shown to determine kitchen workers’ behavioral intentions toward food waste prevention. The TRA suggested that previous attitudes and existing subjective norms were two important influences on a person’s behavioral intentions to implement a certain behavior [52]. Based on these previous studies, this study aimed to understand whether ethical sustainability and professional competence in food waste prevention were important indicators of behavioral intentions among kitchen staff. According to the TRA and TPB, a person’s attitude can indicate a favorable or unfavorable assessment of an action and thus can predict behavioral intentions [51]. Therefore, the following hypotheses were developed:

**H3:** 
*Professional competence can positively influence self-efficacy in food waste prevention.*


**H4:** 
*Professional competence in food waste prevention can positively influence sustainable behavioral intentions during food preparation.*


**H5:** 
*Self-efficacy can positively influence sustainable behavioral intentions during food preparation.*


**H6:** 
*Sustainable behavioral intentions during food preparation can positively influence the promotion of sustainable preparation.*


### 2.5. Mediating Role

Based on the iceberg model proposed by Spencer and Spencer [53], professional competence includes knowledge, skills, motivation, traits, and self-concept. Explicit traits, including knowledge and skills, are both easily affected by the external environment, trainable, and can be enhanced through education and learning. Implicit traits, such as self-concept, traits, and motivation, cannot be easily explored and developed and, therefore, need to be inherent in the individual. The capacity for efficient food handling is the most important factor in preventing food waste [54]. Self-efficacy is important during food preparation and is also linked to preventing food waste [55].

There is a considerable relationship between the self-efficacy of food handling competence and general self-efficacy. Therefore, if you can enhance your personal self-efficacy, you can also enhance your competence in food handling. It is necessary to improve professional knowledge and skills, as it can improve self-efficacy and food-handling behavior [56,57,58]. In addition, through food handling and cooking classes, self-efficacy can be enhanced, and behavioral ability can be improved [59]. Therefore, the following hypotheses were developed:

**H7:** 
*Professional competence in food waste prevention mediates the attitudes toward ethical sustainability and the sustainable behavioral intentions during food preparation.*


**H8:** 
*Self-efficacy mediates the professional competence in food waste prevention and the sustainable behavioral intentions during food preparation.*


## 3. Methodology

### 3.1. Data Collection

In the beginning, a pre-study was conducted on the reliability of the study. The first questionnaire included questions related to all the constructs of this study. The second part obtained the demographics of the participants, including gender, age, education, job title, work experience, and restaurant type.

To ensure the validity of the questionnaire, we invited three professors to evaluate the validity of the predicted measurements. One hundred pilot-study questionnaires were returned with a valid return rate of 100%. The research process included item and factor analyses to determine the final questionnaire. All items in the questionnaire had good reliability (Cronbach’s a >0.70). In addition, according to the feedback of the participants, some items were revised.

This study included employees in Taiwan’s catering industry (including hotels, restaurants, and group-meal companies) as the research subjects. The convenience sampling and purposive sampling methods were adopted, and assistance from individual practitioners in the catering industry who met the sampling criteria was sought through telephone interviews and e-mails. From May to August 2022, questionnaires were sent and returned by mail. Out of 500 questionnaires, there are 415 valid questionnaires, which yielded a 90.2% valid response rate.

### 3.2. Measures

A questionnaire survey method was adopted for this study. The measurement scales in this study are all from previous related research. The questionnaires used were divided into 6 parts: part 1, ethical sustainability, including 3 items; part 2, professional competence in food-waste prevention scale, including 29 items for 7 dimensions; part 3, self-efficacy, including 5 items; part 4, sustainable food behavioral intentions during food preparation, including 3 items; part 5, sustainable food promotion, including 4 items; and part 6, personal demographic.

For items measuring ethical sustainability, a three-item scale was adapted from the work of Lindeman and Väänänen [60]. This scale was based on a 5-point Likert scale, ranging from 1 (“strongly disagree”) to 5 (“strongly agree”). The Cronbach’s α value of this scale was 0.832. The scale of professional competence in food waste prevention was according to the professional competence in food waste prevention scale of Ko and Lu [47], with 29 questions of 7 dimensions and a 5-point Likert scale ranging from 1 (“strongly disagree”) to 5 (“strongly agree”). The average mean of the questions in each dimension was taken as the average mean for analysis, and the total Cronbach’s α value of the scale was 0.926. To measure self-efficacy, 5 items from Chen, Gully, and Eden [61] were adapted and rated by a 5-point Likert scale, ranging from 1 (“strongly disagree”) to 5 (“strongly agree”). Finally, sustainable behavioral intentions during food preparation were measured using 10 items from Fang, Wang, and Hsu [62], and there were 3 questions. The Cronbach’s α value of this scale was 0.907. The sustainable preparation promotion was adapted from the 4-item scale of Teng, Chih, and Wang [63]. The Cronbach’s α value of this scale was 0.850. These items were rated by a 5-point Likert scale, ranging from 1 (“strongly disagree”) to 5 (“strongly agree”).

### 3.3. Data Analysis

The model comprised a measurement model and a causal structural model. Before data analysis, we performed structural equation modeling (SEM) using the AMOS 21.0 application software. Structural equation modeling is a multivariate statistical analysis technique commonly used in social sciences; it is used to analyze the structural relationship between various facets, and the technique is a combination of factor analysis and multiple regression analysis.

The potentially disguised relationship between variables was explored by confirmatory factor analysis (CFA). Path analysis was used in the study to evaluate the fit of the structural model and the hypothesized relationships among the latent variables [64]. As suggested by Anderson and Gerbing [64], we started with a CFA to determine whether the measurement model was acceptable.

## 4. Results

The subjects were mostly male in an age range of 21–40 years (70.1%). The education level was mostly high school and university. Non-management positions accounted for 71.8%, working experience of 1–4 years accounted for 28.2%, and 4–12 years accounted for 35.6%. Most of the subjects in this study are young people who have worked for more than 4 years in hotel restaurants, independent restaurants, and restaurant chains (including central kitchens) (Table 1).

### 4.1. Measurement Model

The measurement model is shown in Table 2. The model data of x^2^/df = 2.77; RMSEA = 0.071; GFI = 0.87; AGFI = 0.84; CFI = 0.98; NFI = 0.97; and SRMR = 0.043 are acceptable [65]. The loading factor ranged from 0.65 to 0.89 for all items [66]. The composite reliability (CR) values ranged from 0.88 to 0.95 for all the constructs, indicating that all constructs had good internal reliability [67]. Furthermore, the convergent validity using the average extracted variance (AVE) for each construct exceeded 0.50 in the measurement model [68]. Table 3 shows that the facets were ranging from 0.459 to 0.755. In addition, the square root of AVE for each construct was also higher than the cross-correlation among all variables, so this model had discriminant validity [68].

### 4.2. Structural Model for Path Analysis of Ethical Sustainability, Professional Competence in Food waste Prevention, Self-Efficacy, and Sustainable Behavioral Intentions during Food Preparation

The structural models were x^2^/df = 3.10; GFI = 0.86; AGFI = 0.83; CFI = 0.98; NFI = 0.97; RMSEA = 0.077; and SRMR = 0.067, which indicated the good fit of the structural model [65]. The path analysis showed that ethical sustainability positively affected professional competence in food waste prevention (b = 0.51) and self-efficacy (b = 0.14), and professional competence in food waste prevention positively affected self-efficacy (b = 0.65) and sustainable behavioral intentions during food preparation (b = 0.43). Self-efficacy also positively affected sustainable behavioral intentions during food preparation (b = 0.41), and sustainable behavioral intentions during food preparation positively affected sustainable preparation promotion (b = 0.800). Therefore, the results supported H1, H2, H3, H4, H5, and H6 (Figure 3).

### 4.3. Mediating Effect

In order to understand whether professional competence in food waste prevention and self-efficacy are the influencing factors between ethical sustainability and behavior, the mediating effect was further explored using the Preacher and Hayes [69] bootstrapping techniques to test the mediating effect by using 95% confidence intervals and 10,000 bootstraps for the study. Ethical sustainability showed a significant overall impact on sustainable preparation promotion (Table 4). Ethical sustainability showed a positive significant direct effect and indirect effect on self-efficacy. These results indicated that professional competence in food waste prevention played a partial mediating role in the relationships between ethical sustainability and self-efficacy (supported H7). Similarly, professional competence in food waste prevention had a significant direct effect and indirect effect on sustainable behavioral intentions during food preparation. These results indicated that self-efficacy had a partial mediating role between professional competence in food waste prevention and sustainable behavioral intentions during food preparation. Therefore, H8 was supported.

## 5. Discussion

Ethical sustainability positively affected professional competence in food waste prevention and self-efficacy, while professional competence in food waste prevention was the mediating factor between ethical sustainability and sustainable behavioral intentions during food preparation. Edwards and Mercer [70] strongly suggested that lifestyle choices, such as diet and career choices, have an ethical basis. Waston and Meah [32] explained that the primary driver for lower food waste is an “ethic of thrift”, which is the ethical motivation to act appropriately and “be thrifty”. The ethical rule for food waste behavior is “don’t waste food”. Therefore, the core ethical commitment in the study was to avoid wasting food, thus making consumption more reasonable [39]. Therefore, kitchen staff must have a personal imperative to prevent food waste in order to establish relevant behaviors and build professional competence in that arena.

Based on social cognitive theory in the context of food waste, females were more cognitively advanced than males. Females more closely monitored the reduction of food waste, so they were more motivated [71]. This contributed to their identification of more motivational factors that could have had a positive effect on their behavior. The environment affected the individuals and, in turn, their behavior. If actual catering producers were more aware of food waste, it could help enhance their professional competence in food waste prevention.

The research on food waste has shown that preventing food waste requires different culinary skills and knowledge so that leftover food can be re-purposed [72,73]. During storage, leftovers were often misplaced or left in the refrigerator for too long and spoiled [14,15,17]. Therefore, professional competence in food preparation is important.

Professional competence in food waste prevention positively affected self-efficacy and sustainable behavioral intentions during food preparation. Self-efficacy was the mediating factor between professional competence in food waste prevention and sustainable behavioral intentions during food preparation. Lin [74] defined knowledge-sharing as a culture of social interaction, and the composition of this culture includes knowledge exchange, experiential inheritance, and mutual learning among employees because the growth of knowledge could increase self-efficacy. Ding and Huang [75] suggested that mutual assistance, cooperation, and knowledge-sharing are critical for obtaining a competitive advantage. Bandura [20] suggested that individuals with high self-efficacy can have more enthusiasm, broader vision, broader thinking, and, thus, more courage to accept challenges, and these behaviors are predominantly guided by the successful experiences of others. Self-efficacy and behavioral intentions could, indeed, predict food-handling behaviors [30]. Relevant educational policies should support consumers in developing self-efficacy related to sustainable food handling. Previous research points to the need for self-efficacy to enhance knowledge and skills, but building self-efficacy needs to be based on actual experience [45,56,58,64].

Self-efficacy can be conceptualized with different levels of linkages, ranging from general self-efficacy (a person’s ability to exercise self-control in different situations) to specific self-efficacy (related to a person’s ability to control behavior in specific settings (such as work)) to task-related self-efficacy (such as confidence in being able to correctly sort leftover waste) [76]. Domain- and task-related self-efficacy is positively correlated with general self-efficacy [77]. There are different levels of self-efficacy in food waste, and only by deeply understanding the root causes of food waste can we truly understand the factors of household food waste [76]. Therefore, enhancing professional competence in food waste prevention can improve self-efficacy and help promote more sustainable practices around food preparation and surplus food.

## 6. Conclusions and Suggestions

### 6.1. Conclusions

According to the data analysis results, the dimensions of ethical sustainability, professional competence in food waste prevention, self-efficacy, sustainable preparation intention, and sustainable preparation promotion were all found to be significantly and positively correlated. This study first analyzed the CFA among the various dimensions, and the results showed that this model had a good fit. All indices passed the standards, indicating a compliant fit. This model could interpret the impact of each dimension. Ethical sustainability positively affected professional competence in food waste prevention and self-efficacy, and professional competence in food waste prevention positively affected self-efficacy and sustainable behavioral intentions during food preparation. Similarly, self-efficacy positively affected sustainable behavioral intentions during food preparation, and sustainable behavioral intentions during food preparation significantly and positively affected sustainable preparation promotion. Therefore, H1, H2, H3, H4, H5, and H6 all had empirical support. Moreover, professional competence in food waste prevention was the factor mediating ethical sustainability to sustainable behavioral intentions during food preparation, while self-efficacy was the factor mediating professional competence in food waste prevention to sustainable behavioral intentions during food preparation. Both H7 and H8 were supported.

### 6.2. Management Suggestions and Implications

This study shed light on the factors influencing food waste prevention in the catering industry and can help the government and catering stakeholders to make specific recommendations for promoting food waste prevention. The results showed that the ethical sustainability attitudes of kitchen staff affected professional competence in food waste prevention and self-efficacy. Ethical sustainability among kitchen staff was related to their individual experiences and work/education environments. Therefore, the awareness and education provided by companies and governments concerning food waste prevention could indirectly affect the concept of ethical sustainability among kitchen staff. Influencing the knowledge, attitudes, and skills of professional competence in food waste prevention and continuously promoting better strategies for surplus food are important factors for governmental and corporate sustainability promotion. Self-efficacy indicates the individual believes that they can successfully perform a certain task or challenge, and they have the self-confidence to complete the task successfully [18,23]. We need more consumers who are aware of and committed to food waste prevention, and we need strong policymakers with the right strategies to promote food waste prevention systematically. In addition, the cultivation of professional competence in food waste prevention could affect self-efficacy and could have a significant impact on behavioral intentions through self-efficacy. For example, a company could regularly hold competitions for the creative preparation of surplus food in order to encourage self-reflection in catering practitioners [78]. Competitions or mutual internal audits of surplus food and food waste prevention among catering practitioners could be helpful in understanding limitations and sharing effective approaches, which could then further improve the reduction of food waste company-wide.

### 6.3. Practical Suggestions and Implications

It has been shown that the greater the professional competence of employees, the greater the increase in self-efficacy. Sustainability issues have become an important consideration in the catering industry in recent years. The responsibility and prevention of food waste are not only related to consumers but, rather, are closely related to the policies and factors that govern the entire supply chain. It is an encouraging sign that food waste prevention is now a major socio-economic issue globally.

Implicit traits could be adjusted through a longer period of education and training, psychological counseling, or accumulated practical training [37]. According to our results, the level of performance depends on the level of self-efficacy and professional competence. More time must be spent on the psychological development of professional competence before effectiveness can be achieved. Related courses such as sustainability ethics, workplace ethics, and professional ethics are also indispensable components. In these courses, such topics can be discussed via case studies, peer experience sharing, and industry experience seminars for exploring individual perceptions that contribute to the improvement of self-efficacy and overall behavior motivation.

Kitchen staff are crucial for food preparation and management in the catering industry. The knowledge–attitude–behavior model [79] suggested that increased environmental knowledge can enhance environmental awareness and concern, leading to more environmentally conscious behaviors. This knowledge also generated positive thoughts, concerns, and practices [80]. In other words, the environment changes the person, and only personal changes can lead to changes in attitudes toward surplus food and more sustainable efforts when processing ingredients and preparing food. These can, then, aid the catering industry in food waste prevention, sustainable ethics, the sustainable use of ingredients, sustainable cooking, and environmental protection.

### 6.4. Research Limitations and Future Studies

There were some limitations that should be considered. First, we primarily focused on those working in hotels and restaurants, as well as those working for group meal providers as a whole. To understand the differences in their environments, they should be considered separately. In the future, chain restaurants and central catering factories may be considered as the subjects, and different results may be obtained since other products in these catering sectors have been more standardized.

Since survey methods are used to collect data over a short period of time, the results may be somewhat biased because the research instrument was a self-report survey [81]. Therefore, future studies should design more rigorous research procedures, as the data vary due to different data sources. Finally, this study involved quantitative research, which could be supplemented by qualitative research in the future to understand issues associated with its implementation. We could have collected more factors affecting food waste in order to explore the actual situation of sustainable preparation behavior, such as the types of service and the ranks of kitchen staff, as well as to discuss the factors regulating sustainable preparation behavior.

## Figures and Tables

**Figure 1 foods-12-03028-f001:**
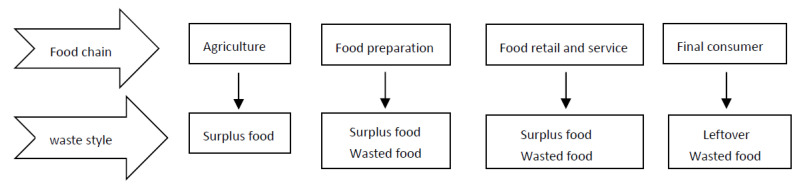
Food waste styles in all stages of the food chain.

**Figure 2 foods-12-03028-f002:**
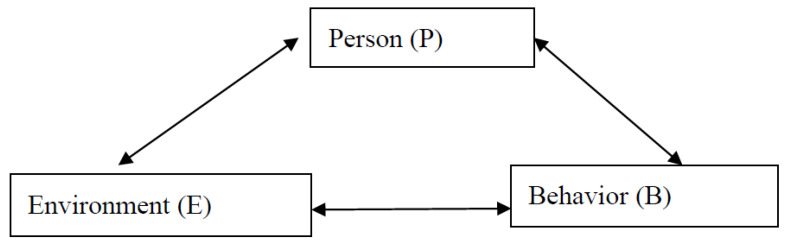
Interactive model of social cognitive theory.

**Figure 3 foods-12-03028-f003:**
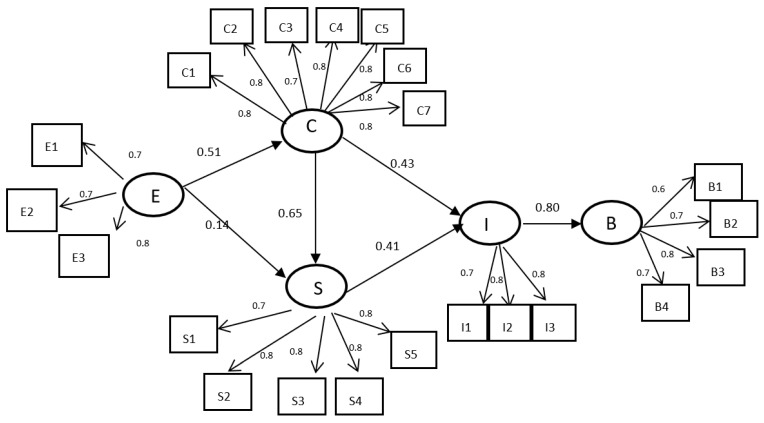
Structural model for path analysis of ethical sustainability, professional competence in food waste prevention, self-efficacy, and sustainable behavioral intentions during food preparation. (E: ethical sustainability; C: professional competence in food waste prevention; S: self-efficacy; I: sustainable behavioral intentions during food preparation; and B: sustainable preparation promotion).

**Table 1 foods-12-03028-t001:** Sociodemographic details of subjects.

Background Variables	Category	Number of Persons	Percentage (%)
Gender	Male	217	61.8
	Female	134	38.2
Age	Below 20	8	2.3
	21–30	139	39.6
	31–40	107	30.5
	41–50	65	18.5
	51–60	27	7.7
	Older 61	5	1.4
Educational level	Junior high school or below	14	4
	High school/vocational high school	130	37
	College	67	19.1
	University	130	37
	Graduate school	10	2.8
Job position	Management	99	28.2
	Non-management	252	71.8
Working years	1 year or less	24	6.8
	More than 1 to 4 years (inclusive)	99	28.2
	More than 4 to 8 years (inclusive)	64	18.2
	More than 8 to 12 years (inclusive)	61	17.4
	More than 12 to 16 years (inclusive)	34	9.7
	More than 16 to 20 years (inclusive)	29	8.3
	More than 20 years	40	11.4
Employment	Restaurant in hotel	120	34.2
	Individual restaurant	136	38.7
	Group meal/central kitchen	82	23.4
	Others	13	3.7

**Table 2 foods-12-03028-t002:** Measurement model estimated parameters.

Items	Std. Factor Loadings	Cronbach’s α	CR	AVE
**E. Ethical sustainability**		0.832	0.836	0.630
E1. I will choose to use the food produced through a method that does not harm the animals (for example, enough space for raising).	0.77			
E2. I can consider environmentally friendly ways to prepare food and try to avoid food waste.	0.79			
E3. I will consider ways that will not disrupt the ecological balance to avoid waste in the food preparation process.	0.82			
**C. Professional competence in food waste prevention ^a^**		**0.926**	**0.930**	**0.655**
C1. Hygiene knowledge	0.80			
C2. Cooking knowledge	0.85			
C3. Personal moral attitude	0.72			
C4. Attitude towards ingredients	0.81			
C5. Education and training attitude	0.83			
C6. Dish analysis skills	0.81			
C7. Planning and management skills	0.84			
**S. Self-efficacy**		0.907	0.910	0.670
S1. I can set a goal to avoid food waste, and I think I can get the desired result.	0.76			
S2. I will be able to successfully overcome several challenges to avoid food waste.	0.83			
S3. I believe that I can effectively perform several different tasks to avoid food waste.	0.84			
S4. Compared with others, I can complete most tasks well to avoid food waste.	0.82			
S5. Even if I encounter difficulties in food waste prevention, I can try my best to achieve the goal.	0.84			
**I. Sustainable behavioral intentions during food preparation**		0.879	0.843	0.696
I1. I am willing to take food preparation activities for a healthy and sustainable diet.	0.78			
I2. I will encourage others to participate in healthy and sustainable food preparation together.	0.86			
I3. I will adjust and change my food preparation habits to promote sustainable positive effects of food preparation on health and the environment.	0.86			
**B. Sustainable preparation promotion behavior**		0.850	0.856	0.600
B1. I implement green procurement (e.g., purchase local food, support green diet).	0.65			
B2. I implement and supervise others to reduce food waste in preparation.	0.77			
B3. I participate in the action of a healthy and sustainable diet.	0.89			
B4. I support the sustainable food/snacks activities organized by the government.	0.77			

^a^ Twenty-nine items measured for professional competence loaded onto seven factors.

**Table 3 foods-12-03028-t003:** Relevance among various dimensions and the correlations among all the constructs.

Variable	Average Mean	Standard Deviation	E ^1^	C	S	I	B
E. Ethical sustainability	4.10	0.58	0.818 ^2^				
C. Professional competence in food waste prevention	4.39	0.55	0.459	0.789			
S. Self-efficacy	4.12	0.51	0.712	0.497	0.806		
I. Sustainable behavioral intentions during food preparation	4.23	0.61	0.680	0.537	0.665	0.844	
B. Sustainable preparation promotion behavior	4.13	0.61	0.680	0.519	0.721	0.755	0.777

N = 351; ^1^ E: ethical sustainability; C: professional competence in food waste prevention; S: self-efficacy; I: sustainable behavioral intentions during food preparation; B: sustainable preparation promotion; and ^2^ the square root of the AVE for discriminant validity.

**Table 4 foods-12-03028-t004:** The indirect, direct, and total effects for the hypothetical model.

	Bootstrapping			
	Estimate	Bias-Corrected		
		Lower	Upper	Two-Tailed Significance
Indirect effects				
Ethical sustainability→Professional competence in food waste prevention→Self-efficacy	0.329	0.224	0.432	0.007 **
Professional competence in food waste prevention→Self-efficacy→Sustainable behavioral intentions during food preparation	0.266	0.181	0.397	0.004 **
Direct effects				
Ethical sustainability→Professional competence in food waste prevention	0.510	0.385	0.603	0.019 *
Ethical sustainability→Self-efficacy	0.140	0.032	0.277	0.021 *
Professional competence in food waste prevention→Self-efficacy	0.645	0.483	0.740	0.009 **
Professional competence in food waste prevention→Sustainable behavioral intentions during food preparation	0.425	0.250	0.617	0.006 **
Self-efficacy→Sustainable behavioral intentions during food preparation	0.413	0.245	0.577	0.018 *
Sustainable behavioral intentions during food preparation→Sustainable preparation promotion	0.798	0.721	0.859	0.009 **
Total effects				
Ethical sustainability→Sustainable preparation promotion	0.328	0.234	0.420	0.018 *

** *p* < 0.01, * *p* < 0.05.

## Data Availability

The data used to support the findings of this study can be made available by the corresponding author upon request.

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
