# Peer review of "Sustainable Preparation Behavior for Kitchen Staff in Order to Limit Food Waste"

_foods, 2023, doi:10.3390/foods12163028_

Round 1

Reviewer 1 Report

Comments and Suggestions for Authors

Dear authors,

the topic of your manuscript is valuable. 

However, I have several remarks:

1. The manuscript needs English editing.

2. The references are not according to MDPI guidelines.

3. The manuscript is poorly constructed. Please, check MDPI guidelines. Reorganise the manuscript.

4. The aim of the study is not described. You should describe it clearly.

5.  Tables 2, 3 and 4  : There is a point before the numbers. Why?

6. Figure 2. It is not reader-friendly. Please revise the figure and explain it better.

7. In the section "Practical Implications" :

"The academic knowledge of kitchen staff must also be continuously improved."

The academic education is not obligatory for kitchen staff. The sentence is not clear. You have to revise it.

8. You must separate the conclusion and the discussion.

Comments on the Quality of English Language

The manuscript needs English editing.

Author Response

To Editor-in-Chief and reviewers:

Thank you for your comments and suggestions. I appreciate the time and effort that you and the reviewers have dedicated to providing your valuable feedback on my manuscript. I had major revised in this paper as final attached. Please see our comments below.

The following are Reviewer comments:

Overall

  1. The manuscript needs English editing.
  2. The references are not according to MDPI guidelines.

Ans: I had made major revise in all sections.

I had revised paper due to English language editing by MDPI. Please see the certification as below, and I also to check MDPI guideline to check all paper. Please see the revised paper. And, I also change title as “Sustainable Preparation Behavior for Kitchen Staff in order to Limit Food Waste”.

The following are our responses to the Reviewers’ 1 comments:

  1. The aim of the study is not described. You should describe it clearly.

Ans:  I have added a statement in Literature, which should help to clear up the issue. Please see below.

Food is an important daily necessity. With advancements in the global economy, equipment, and food production efficiency, an abundant supply of food can provide for more people. From production to consumption, food is often discarded by consumers even though it is safe to eat because they have bought too much or have insufficient knowledge about the shelf life of food. It is estimated that one-third (1.3 billion tons) of the food produced globally for consumption is wasted every year [1,2]. The problem of food waste has been included in the World Health Organizations’ sustainable development goals (SDG 17) in order to reduce global human food waste by half and to reduce food loss by 2030. Food service and family experiences have been shown to be an appropriate way to address national and international sustainable goals [1].

The problem of surplus food management has also attracted significant attention from academics and other concerned practitioners because the normal dietary sources of a population are often interrupted due to limited resources, and experts believe that it is possible to reduce food insecurity via surplus food management. In addition to the commercial economic losses due to food production and waste, the social and environmental losses have also been significant, especially when foods that were still edible were not able to be distributed and sold, for various reasons, and were eventually discarded [3].

The Food and Agriculture Organization (FAO) of the United Nations has estimated that the annual global food waste across the entire food chain was USD 2.6 trillion. The Waste and Resources Action Plan estimated that food waste in the UK totaled around 10 million tons per year, 70% of which was avoidable [4]. Food waste is an urgent and serious issue in Western countries. The increase in the amount of food waste directly affects environmental changes, including greenhouse gas emissions and packaging pollution. In Poland, 9.2 million tons of grain have been lost every year, 53% of which was produced by consumers. In order to reduce consumer food waste, it was necessary to understand the behavioral factors [5]. The key problem of food waste has been primarily due to the supply chain, from production, preparation, and processing to transportation and consumption, all of which require considerable costs and resources. Therefore, practitioners and consumers should pay close attention to this issue from both the supply and the demand side. Reducing food waste is an important factor for sustainable development. The predicted population growth will require an adequate food system that includes the sustainable development of natural ecological resources [6,7]. Food waste occurs at all stages of the food supply chain, and the further down the supply chain, the greater the negative environmental, economic, and social impacts of food waste [8].

In restaurants, government intervention and enhanced waste disposal collection are necessary. Research has shown that changing consumer behavior, combined with corporate social responsibility, mitigated food waste [9]. The results of the study showed that many restaurants have prioritized reducing food waste. However, most of the final factors affecting food waste in restaurants were quality and specification issues during produ711ction. Generally, food waste was the worst in casual restaurants, and large meal portions resulted in more food waste [10].

The chefs are the souls of a restaurant. In addition to making good dishes, a good chef can also improve the product value while maintaining their social responsibility. Food oversupply has also been related to consumers' perceived concept of preservation. Consumers may purchase more than they need in order to save time [11, 12]. Among these food behaviors, discarding reusable ingredients as kitchen waste was the most severe (28.8% of ingredients were considered kitchen waste and discarded) [13]. Storing and sorting food in an orderly manner (for example, stacking old and new food, or sorting by frequency of use), and reordering regularly could reduce food waste [14, 15]. According to Secondi, Principato, and Laureti [16], one of the best courses of action to avoid wasting food was to estimate portion sizes correctly. Misplaced, forgotten, or expired leftovers were an issue when storing food [17].

Social cognitive theory (SCT) explains human behavior based on the dynamic interactions of individual experiences, the actions of others, and environment influences [18]. Self-efficacy is the belief in one's ability to change motivations, cognitive resources, and courses of action in order to meet the needs of a particular situation [19-22]. Research has found that self-efficacy predicted several job-related outcomes, such as work attitudes, training proficiency, and job performance. Bandura [23] pointed out that self-efficacy was the result of interactions with the external environment, personal professional competency, and learning effectiveness, which could then motivate behavior. Kitchen staff play a key role in food production from the kitchen to the table, and a sustainability-focused food revolution should be initiated for reducing carbon emissions and mitigating food waste to preserve the planet’s ecosystem.

Therefore, the attitudes of kitchen staff to avoid food waste could affect how they handle food, and sustainable cooking could provide substantial results in the future for environmental and food sustainability. This study explored the perspectives of kitchen staff towards limiting food waste by evaluating the behavioral patterns such as ethical sustainability, professional competence in food-waste prevention, self-efficacy, behavioral intentions towards sustainable food preparation, and sustainable food promotion from a practical perspective. This research could help promote sustainable cooking and food preparation, from the individual to the restaurant, and then to the catering industry. By encouraging consistent sustainability in the catering industry, we can more effectively reduce food waste.

---------------------------------------------------------------------------

  1. Tables 2, 3 and 4 : There is a point before the numbers. Why

Ans: I had deleted and check. Please see the revised paper.

-------------------------------------------------------------------------------------

  1. Figure 2. It is not reader-friendly. Please revise the figure and explain it better.

Ans: I had revised as below. Please see the revised paper.

Figure 3. Path analysis of each construct for overall structure model

(E: Ethical sustainability; C: Professional competence in food-waste prevention; S: Self-efficacy; I: Sustainable behavioral intentions during food preparation; B: Sustainable preparation promotion)

  1. In the section "Practical Implications" :"The academic knowledge of kitchen staff must also be continuously improved."The academic education is not obligatory for kitchen staff. The sentence is not clear. You have to revise it.

 Ans:  I had revised as below.

Practical Implications

It has been shown that the greater the professional competence of employees, the greater the increase in self-efficacy. Sustainability issues have become an important consideration in the catering industry in recent years. The responsibility and prevention of food waste are not only related to consumers but, rather, are closely related to the policies and factors that govern the entire supply chain. It is an encouraging sign that food-waste prevention is now a major socio-economic issue globally.

Implicit traits could be adjusted through a longer period of education and training, psychological counseling, or accumulated practical training [37]. According to our results, the level of performance depends on the level of self-efficacy and professional competence. More time must be spent on the psychological development of professional competence before effectiveness can be achieved. Related courses such as sustainability ethics, workplace ethics, and professional ethics are also indispensable components. In these courses, such topics can be discussed via case studies, peer experience sharing, and industry experience seminars for exploring individual perceptions that contribute to the improvement of self-efficacy and overall behavior motivation.

Kitchen staff are crucial for food preparation and management in the catering industry. The knowledge–attitude–behavior model [79] suggested that increased environmental knowledge could enhance environmental awareness and concern, leading to more environmentally conscious behaviors. This knowledge also generated positive thoughts, concerns, and practices [80]. In other words, the environment changed the person, and only personal changes could lead to changes in attitudes towards surplus food and more sustainable efforts when processing ingredients and preparing food. These could, then, aid the catering industry in food-waste prevention, sustainable ethics, the sustainable use of ingredients, sustainable cooking, and environmental protection.

-----------------------------------------------------------------------------------------

5.5 You must separate the conclusion and the discussion.

Ans: I had separate already, please see revised paper.

Sustainable Preparation Behavior for Kitchen Staff in order to Limit Food Waste

Abstract

The concepts of culinary sustainability and avoiding the waste of surplus food have become important sustainability trends today. How to integrate the handling of surplus food in the catering industry is a topic of concern in the industry. Kitchen staff are the vital soul of any restaurant, and we intend to discuss how kitchen staff actually behave and to explore factors that influence their behaviors in order to develop an implementation model for food-waste prevention. Therefore, this study explored a model of ethical sustainability, professional competence, self-efficacy, sustainable food preparation objectives, and sustainable food promotion and behavior focused on limiting food waste.  This study used a questionnaire and surveyed employees who had been employed for more than 6 months. From May to August 2022, 500 questionnaires were distributed; 415 valid questionnaires were retrieved, yielding a 90.2% recovery rate. According to the structural equation modeling analysis between the dimensions, ethical sustainability should have had a positive influence on professional competence in food-waste prevention and self-efficacy. Professional competence in food-waste prevention affected self-efficacy and behavioral intentions during food preparation; self-efficacy also significantly affected behavioral intentions towards sustainable food preparation. Similarly, behavioral intention had a positive influence on promoting sustainable behaviors. This model was a good fit for the acquired data. Professional competence in food-waste prevention was found to be the mediating factor between ethical sustainability and the behavioral intentions towards sustainable food preparation, and self-efficacy was the mediating factor between the professional competence in food-waste prevention and the behavioral intentions towards sustainable food preparation.

Keywords: surplus food, sustainable preparation behavior, food waste, kitchen staff, ethical sustainability, self- efficacy

  1. Introduction

Food is an important daily necessity. With advancements in the global economy, equipment, and food production efficiency, an abundant supply of food can provide for more people. From production to consumption, food is often discarded by consumers even though it is safe to eat because they have bought too much or have insufficient knowledge about the shelf life of food. It is estimated that one-third (1.3 billion tons) of the food produced globally for consumption is wasted every year [1,2]. The problem of food waste has been included in the World Health Organizations’ sustainable development goals (SDG 17) in order to reduce global human food waste by half and to reduce food loss by 2030. Food service and family experiences have been shown to be an appropriate way to address national and international sustainable goals [1].

The problem of surplus food management has also attracted significant attention from academics and other concerned practitioners because the normal dietary sources of a population are often interrupted due to limited resources, and experts believe that it is possible to reduce food insecurity via surplus food management. In addition to the commercial economic losses due to food production and waste, the social and environmental losses have also been significant, especially when foods that were still edible were not able to be distributed and sold, for various reasons, and were eventually discarded [3].

The Food and Agriculture Organization (FAO) of the United Nations has estimated that the annual global food waste across the entire food chain was USD 2.6 trillion. The Waste and Resources Action Plan estimated that food waste in the UK totaled around 10 million tons per year, 70% of which was avoidable [4]. Food waste is an urgent and serious issue in Western countries. The increase in the amount of food waste directly affects environmental changes, including greenhouse gas emissions and packaging pollution. In Poland, 9.2 million tons of grain have been lost every year, 53% of which was produced by consumers. In order to reduce consumer food waste, it was necessary to understand the behavioral factors [5]. The key problem of food waste has been primarily due to the supply chain, from production, preparation, and processing to transportation and consumption, all of which require considerable costs and resources. Therefore, practitioners and consumers should pay close attention to this issue from both the supply and the demand side. Reducing food waste is an important factor for sustainable development. The predicted population growth will require an adequate food system that includes the sustainable development of natural ecological resources [6,7]. Food waste occurs at all stages of the food supply chain, and the further down the supply chain, the greater the negative environmental, economic, and social impacts of food waste [8].

In restaurants, government intervention and enhanced waste disposal collection are necessary. Research has shown that changing consumer behavior, combined with corporate social responsibility, mitigated food waste [9]. The results of the study showed that many restaurants have prioritized reducing food waste. However, most of the final factors affecting food waste in restaurants were quality and specification issues during produ711ction. Generally, food waste was the worst in casual restaurants, and large meal portions resulted in more food waste [10].

The chefs are the souls of a restaurant. In addition to making good dishes, a good chef can also improve the product value while maintaining their social responsibility. Food oversupply has also been related to consumers' perceived concept of preservation. Consumers may purchase more than they need in order to save time [11, 12]. Among these food behaviors, discarding reusable ingredients as kitchen waste was the most severe (28.8% of ingredients were considered kitchen waste and discarded) [13]. Storing and sorting food in an orderly manner (for example, stacking old and new food, or sorting by frequency of use), and reordering regularly could reduce food waste [14, 15]. According to Secondi, Principato, and Laureti [16], one of the best courses of action to avoid wasting food was to estimate portion sizes correctly. Misplaced, forgotten, or expired leftovers were an issue when storing food [17].

Social cognitive theory (SCT) explains human behavior based on the dynamic interactions of individual experiences, the actions of others, and environment influences [18]. Self-efficacy is the belief in one's ability to change motivations, cognitive resources, and courses of action in order to meet the needs of a particular situation [19-22]. Research has found that self-efficacy predicted several job-related outcomes, such as work attitudes, training proficiency, and job performance. Bandura [23] pointed out that self-efficacy was the result of interactions with the external environment, personal professional competency, and learning effectiveness, which could then motivate behavior. Kitchen staff play a key role in food production from the kitchen to the table, and a sustainability-focused food revolution should be initiated for reducing carbon emissions and mitigating food waste to preserve the planet’s ecosystem.

Therefore, the attitudes of kitchen staff to avoid food waste could affect how they handle food, and sustainable cooking could provide substantial results in the future for environmental and food sustainability. This study explored the perspectives of kitchen staff towards limiting food waste by evaluating the behavioral patterns such as ethical sustainability, professional competence in food-waste prevention, self-efficacy, behavioral intentions towards sustainable food preparation, and sustainable food promotion from a practical perspective. This research could help promote sustainable cooking and food preparation, from the individual to the restaurant, and then to the catering industry. By encouraging consistent sustainability in the catering industry, we can more effectively reduce food waste.

  1. Theoretical background and hypotheses development

2.1 Surplus food and food waste

The conceptual model of availability, surplus, recoverability, and waste (ASRW) provided a clear definition of surplus food and distinguished it from food waste by introducing the concepts of availability, surplus, recoverability, and waste. According to this model, available food was defined as "all food in the food supply chain", which consisted of three parts: "human consumption", "surplus food", and "food scraps”. There has been no universally accepted definition of food waste and surplus food because there have been different interpretations from different perspectives [24].

According to the FAO, food waste consisted of three distinct problems. Food loss (FL) was the reduction in weight or nutritional value of food originally intended for human consumption. These losses were primarily due to processes in the supply chain, such as poor management, errors, and irregular processes (such as improper farming and harvesting), resulting in reduced edible weight. Food waste (FW) was the discarding of edible food, whether expired or spoiled, due to reasons such as oversupply, lack of a sales strategy, or excessive or unplanned purchases by individual consumers. It was also the waste of edible food due to improper distribution, storage, transportation, and preparation, both in the home and in restaurants [25].

Moreover, food waste occurs at all stages of the food chain: production, distribution, retail, and consumption processes [26]. Types of waste at all stages of the food chain are shown in Figure 1. Wasted food is any food that is lost due to spoilage or being discarded. "Waste food" as opposed to "food waste" refers to food that has not been used for its intended purpose but could be further used as a valuable resource [2].

Figure 1. Food waste styles in all stages of the food chain

According to scholars [27], surplus food could be divided into four categories: (1) Unprofitable crops refer to the market phenomenon after the harvest of agricultural products, and the surplus of grain sales is generated due to a stable sale price. When production is much higher than demand, it can lead to an oversupply in the market. (2) Non-perishable processed foods are foods with a long shelf life (such as dry goods, canned foods, etc.), damaged packaging, incorrect packaging, promotions near the expiration date, etc. (3) Perishable fresh foods are fresh fruits and vegetables, bread, dairy products, refrigerated ready-to-eat foods, etc.; products that are not sold because they are close to their expiration date but are still safe to eat; and foods that agricultural producers and importers may not be able to sell because the fruit or vegetables are irregular in shape and size. (4) Perishable foods include ready-to-eat foods such as sandwiches, cakes, and pastries [27].

2.2 Social Cognitive Theory

Social cognitive theory (SCT) was first proposed by Bandura, who combined the concepts of behaviorism and social learning [18]. Behavior is generated by the interaction between the individual and the environment, rather than determined by any single aspect. These factors include environmental impacts (the overall social environment, organizational policies, and culture); individual cognition and personal factors (personal motivation, attitude); and behavior (intention and willingness), all of which interact. Figure 2 shows the dynamic formation of the person, the environment, and behavior in SCT, where personal behavior is affected by the environment and influenced by personal subjective cognitions. The psychological process of self-introspection is achieved through the inner driving force of the individual, and the person feels and reacts to their external environment due to their inner thoughts, rather than being stimulated by the environment alone. Therefore, this research hypothesized that environmental factors could influence personal factors, and personal factors could influence behavioral factors.

          Figure 2. Interactive model of social cognitive theory

Bandura [28] suggested that persons with high self-efficacy could have more enthusiasm, broader vision, broader thinking, and, therefore, courage to accept challenges. To avoid repeating the failures of others, most individuals would use the success of others as a guide. Rimal [29] also pointed out that SCT was an important basis for predicting food handling processes. Intention and self-efficacy could predict food-handling behavior [30]. Bandura [23] also suggested that personal past performance affected self-efficacy, and higher self-efficacy could also affect future performance. Behavioral improvements could be enhanced by offering continuous reminders during food handling [31].

Bandura [23] considered SCT the most suitable for explaining dynamic human behavior and regarded cognition and self-regulation as the primary framework of causality. The development of this model could cultivate the confidence of individuals to use certain abilities. However, it could also be affected by many internal factors, such as the level of self-efficacy, which would then result in different behaviors. Self-efficacy refers to a person's belief in their own capabilities and their degree of confidence in being able to achieve a specific outcome.

2.3 Ethical Sustainability Affects Personal Behavior

Waston and Meah [32] indicated that even if consumers had an environmental motivation to avoid food waste, the real motivation to reduce food waste was the "ethic of thrift". People avoided wasting food because it caused feelings of guilt that were then combined with environmental concerns [17]. Food represented the physical and symbolic connection between humans and nature [33]. A strong negative attitude towards food waste could be explained by our respect for nature, and this could hopefully be extended to the community [34].

Therefore, food waste could be perceived as a waste of precious resources, which would be unfair from an anthropocentric perspective [35]. Food waste has been considered immoral, so people need to be called upon to change their behaviors and attitudes concerning food waste [36, 37]. Therefore, food is a valuable resource, as it is not always readily available. A change in attitude could benefit both humans and nature by reducing the negative aspects of food production and improving the balance between humans and nature [38]. The “moral” rule for food waste behavior could be "don't waste food". With this in mind, our core ethical commitment was to avoid wasting food through more sensible consumption. Doing so could protect and preserve the world in which we live [39].

As indicated by the self-perception theory, previous behaviors affected future actions. The theory suggested that if individuals were encouraged to behave in environmentally conscious ways, they were more likely to make environmentally conscious decisions [40, 41]. This theory reinforced environmental self-identity, or an individual's perception of themselves as a person who is environmentally conscious [42]. In addition, many previous studies found that social norms and the behaviors of social groups had a positive influence on behavior [33,43].

Therefore, the following hypotheses were developed:

H1: Attitudes towards ethical sustainability affected professional competence in food-waste prevention.

H2: Attitudes towards ethical sustainability affected self-efficacy.

2.4 Relationships among professional competence, self-efficacy, and food-waste prevention

Food knowledge and skills, communication skills, business knowledge, and food concepts were considered the main professional abilities of kitchen staff [44]. To conform to social trends and achieve the following objectives, Hegarty and O'Mahony [45] developed the four major fields of culinary courses, including information; culinary arts; food and life sciences; and business. The basic skills needed for kitchen staff were food science, product preparation, finance, personal management, marketing, language, and computer skills [46]. Professional competence for food-waste prevention included two aspects: hygiene knowledge and culinary knowledge. Professional attitudes towards food-waste prevention included three dimensions, namely personal moral attitudes, attitudes towards food ingredients, and attitudes towards education and training. Skill competence comprised two dimensions: dish analysis and meal planning management [47]. In the restaurant and hospitality industry, both food preparers and consumers are responsible for food waste. Food waste has also been identified as a problem in schools, kindergartens, canteens, gas stations, bars, restaurants, canteens, and hotels [48].

The concept of self-efficacy first appeared in an article by Bandura [20]. It refers to an individual’s belief whether one can successfully complete a task. The emphasis was on ability, and it could promote individual actions in certain areas and affect future activities. This aspect was related to personal values. Bandura [18] also defined self-efficacy as “in order to deal with an impending situation, an individual makes a subjective judgment on whether he or she can successfully perform the required behavior. This judgment will decide how much effort he or she will put in when he or she personally faces difficulties, and how long it will last”. That is, self-efficacy could affect whether a person would persist in an action after setting goals. This was also related to whether a person had the motivation to overcome obstacles in the learning process and the learning strategies adopted when solving problems.

The theoretical basis of perceived behavioral control was originally derived from the self-efficacy theory, so perceived behavioral control and self-efficacy were conceptually similar [49]. Many studies have employed the theory of rational action (TRA) [50] or its extended theory of planned behavior (TPB) model [51] to understand kitchen workers' behavioral intentions towards food-waste prevention. The TRA theory suggested that previous attitude (i.e., approval or disapproval of a certain behavior) and existing subjective norms (i.e., whether the implementation of a certain behavior was approved by others) were two important influences on a person's behavioral intentions to implement a certain behavior [52]. Based on these previous studies, the current study aimed to understand whether ethical sustainability and professional competence in food-waste prevention were important indicators of behavioral intentions among kitchen staff. According to TRA and TPB, a person’s attitude indicated the favorable or unfavorable evaluation of a certain behavior, so it could predict behavioral intentions [51]. Therefore, the following hypotheses were developed:

H3: Professional competence could positively influence self-efficacy in food-waste prevention.

H4: Professional competence in food-waste prevention could positively influence the sustainable behavioral intentions during food preparation.

H5: Self-efficacy could positively influence sustainable behavioral intentions during food preparation.

H6: Sustainable behavioral intentions during food preparation could positively influence the promotion of sustainable preparation.

2.5 Mediating role of professional competence in food-waste prevention and self-efficacy

Based on the iceberg model proposed by Spencer and Spencer [53], professional competence could be classified as five basic factors: motivation, traits, self-concept, knowledge, and skills. Knowledge and skills were explicit traits: Both were easily affected by the external environment, both were trainable, and both could be enhanced through education and learning. Implicit traits, such as self-concept, traits, and motivation, could not be easily explored and developed and, therefore, would need to be inherent in the individual. The capacity for efficient food handling was the most important factor in preventing food waste [54]. Self-efficacy was important during food preparation and was also linked to preventing food waste [55].

There is a considerable relationship between the self-efficacy of food handling competence and general self-efficacy. Therefore, if you can enhance your personal self-efficacy, you can also enhance your competence in food handling. It was necessary to improve professional knowledge and skills, as it could improve self-efficacy and food handling behavior [56, 57, 58]. In addition, through food handling and cooking classes, self-efficacy could be enhanced, and behavioral ability could be improved [59]. Therefore, the following hypotheses were developed:

H7: Professional competence in food-waste prevention mediated the attitudes toward ethical sustainability and the sustainable behavioral intentions during food preparation.

H8: Self-efficacy mediated the professional competence in food-waste prevention and the sustainable behavioral intentions during food preparation.

  1. Methodology

3.1 Data Collection

A pilot study was conducted to ensure the reliability of the measurements. The first part of the questionnaire included questions related to all the variables of this study. The second part obtained the demographics of the participants, including gender, age, education, job title, work experience, and restaurant type.

To ensure the face validity of the questionnaire, we invited three professors to evaluate the validity of the predicted measurements. One hundred pilot-study questionnaires were returned with a valid return rate of 100%. The research process included item and factor analyses to determine the final questionnaire. All items in the questionnaire had good reliability (Cronbach's a > 0.70). In addition, according to the feedback of the participants, some items were revised.

This study included employees in Taiwan’s catering industry (including hotels, restaurants, and group-meal companies) as the research subjects. The convenience sampling and purposive sampling methods were adopted, and assistance was sought, through telephone interviews and e-mails, from individual practitioners in the catering industry who met the sampling criteria. From May to August 2022, questionnaires were sent and returned by mail. Out of 500 questionnaires, 415 valid questionnaires were returned, which yielded a 90.2% valid response rate.

3.2 Measures

  A questionnaire survey method was adopted for this study. All scales used to measure the constructs of this study were adopted from previous studies. The questionnaires used were divided into 6 parts: part 1, ethical sustainability, including 3 items; part 2, professional competence in food-waste prevention scale, including 29 items for 7 dimensions; part 3, self-efficacy, including 5 items; part 4, sustainable food behavioral intentions during food preparation, including 3 items; part 5, sustainable food promotion, including 4 items; and part 6, personal demographic.

For items measuring ethical sustainability, a three-item scale was adapted from the work of Lindeman and Väänänen [60]. This scale was based on a 5-point Likert scale, ranging from 1 (“strongly disagree”) to 5 (“strongly agree”). The Cronbach's α value of this scale was 0.832. The scale of professional competence in food-waste prevention was according to the professional competence in food-waste prevention scale of Ko and Lu [47], with 29 questions of 7 dimensions, and a 5-point Likert scale ranging from 1 (“strongly disagree”) to 5 (“strongly agree”). The average mean of the questions in each dimension was taken as the average mean for analysis, and the total Cronbach's α value of the scale was 0.926. To measure self-efficacy, 5 items from Chen, Gully, and Eden [61] were adapted and rated by a 5-point Likert scale, ranging from 1 (“strongly disagree”) to 5 (“strongly agree”). Finally, sustainable behavioral intentions during food preparation were measured using 10 items from Fang, Wang, and Hsu [62], and there were 3 questions. The Cronbach's α value of this scale was 0.907. The sustainable preparation promotion was adapted from the 4-item scale of Teng, Chih, and Wang [63]. The Cronbach's α value of this scale was 0.850. These items were rated by a 5-point Likert scale, ranging from 1 (“strongly disagree”) to 5 (“strongly agree”).

3.3 Data Analysis

The model comprised two components, a measurement model and a causal structural model. Before data analysis, we performed structural equation modeling (SEM) using the AMOS 21.0 application software. Whether the observed variables reflected the hypothesized latent variables and whether the measurement model adequately fit the data was performed with a confirmatory factor analysis (CFA). Path analysis was used in the study to evaluate the fit of the structural model and the hypothesized relationships among the latent variables [64]. As suggested by Anderson and Gerbing [64], we started with a CFA to determine whether all indicator variables adequately reflected their underlying structure and that the measurement model was acceptable. The fit index, comparative fit index (CFI), and the root-mean-square error of approximation (RMSEA) were used to estimate the overall model fit.

  1. Results

The subjects were mostly male in an age range of 21-40 years (70.1%). The education level was mostly high school and university. Non-management positions accounted for 71.8%, working experience of 1-4 years accounted for 28.2%, and 4-12 years accounted for 35.6%. The participants worked in hotel restaurants, independent restaurants, and restaurant chains (including central kitchens) (Table 1).

Table 1  Sociodemographic details of subjects

Background variables

Category

Number of persons

Percentage (%)

Gender

Male

217

61.8

Female

134

38.2

Age

20 years old or younger

8

2.3

21-30 years old

139

39.6

31-40 years old

107

30.5

41-50 years old

65

18.5

51-60 years old

27

7.7

61 years old or older

5

1.4

Educational level

Junior high school or below

14

4

High school ∕ vocational high school

130

37

Junior college

67

19.1

University

130

37

Graduate school

10

2.8

Job position

Management

99

28.2

Non-management

252

71.8

Working years

1 year or less

24

6.8

More than 1 year to 4 years (inclusive)

99

28.2

More than 4 year to 8 years (inclusive)

64

18.2

More than 8 year to 12 years (inclusive)

61

17.4

More than 12 year to 16 years (inclusive)

34

9.7

More than 16 year to 20 years (inclusive)

29

8.3

More than 20 years 

40

11.4

Employment

Restaurant in Hotel

120

34.2

Individual Restaurant

136

38.7

Group meal/central kitchen

82

23.4

Others

13

3.7

4.1 Measurement Model

The model fit indices then indicated an acceptable overall measurement model: x 2/df = 2.77; GFI = 0.87; AGFI = 0.84; RMSEA = 0.071; SRMR = 0.043; CFI = 0.98; and NFI = 0.97 [65]. Table 2 shows that all items had statistically significant (p < 0.01) [66] factor loadings, ranging from 0.65 to 0.89. The composite reliability values for all the constructs ranged from 0.88 to 0.95, indicating that all constructs had good internal reliability [67]. Furthermore, the average extracted variance (AVE) for each construct exceeded 0.50, which established the convergent validity of the measurement model [68]. Table 3 shows that the facets between all constructs were significant (p < 0.01), ranging from 0.459 to 0.755. In addition, the square root of AVE for each construct was also higher than the cross-correlation among all variables, so this model had discriminant validity [68].

4.2 Structural Model

The structural models were x 2/df = 3.10; GFI = 0.86; AGFI = 0.83; CFI = 0.98; NFI = 0.97; RMSEA = 0.077; and SRMR = 0.067, which indicated the good fit of the structural model [65]. The path analysis showed that environmental sustainability positively affected professional competence in food-waste prevention (b = 0.51, p < 0.000) and self-efficacy (b = 0.14, p < 0.000); professional competence in food-waste prevention positively affected self-efficacy (b = 0.65, p < 0.000) and sustainable behavioral intentions during food preparation  (b = 0.43, p <0.000). Self-efficacy also positively affected sustainable behavioral intentions during food preparation (b = 0.41, p <0.000), and sustainable behavioral intentions during food preparation positively affected sustainable preparation promotion (b = 0.80, p <0.000). Therefore, the results supported H1, H2, H3, H4, H5, and H6 (Figure 2).

4.3 Mediating Effect

The mediating effect of the perceived value in hotel restaurants was tested by performing Preacher and Hayes [69] bootstrapping techniques with 95% confidence intervals, using 10,000 bootstraps to study the direct, indirect, and overall effects of the proposed model. Ethical sustainability had a significant overall impact (standardized coefficient = 0.328, p < 0.05) on sustainable preparation promotion (Table 4). Ethical sustainability had a positive significant direct effect (standardized coefficient = 0.140, p < 0.05) and indirect effect (standardized coefficient = 0.329, p < 0.01) on self-efficacy. These results indicated that professional competence in food-waste prevention played a partial mediating role in the relationships between ethical sustainability and self-efficacy and supported H7 of this study. Similarly, professional competence in food-waste prevention had a significant direct effect (standardized coefficient = 0.425, p < 0.01) and indirect effect (standardized coefficient = 0.329, p < 0.01) on sustainable behavioral intentions during food preparation. These results indicated that self-efficacy had a partial mediating role in the relationships between professional competence in food-waste prevention and sustainable behavioral intentions during food preparation. Therefore, H8 was supported.

Table 2 Measurement model estimated parameters

Items

Std. factor loadings

Cronbach’s α

CR

AVE

E. Ethical sustainability

.832

.836

.630

E1. I will choose to use the food produced through a method that does not harm the animals (for example, enough space for raising).

.77

E2. I can consider environmentally friendly ways to prepare food and try to avoid food waste.

.79

E3. I will consider ways that will not disrupt the ecological balance to avoid waste in the food preparation process.

.82

C. Professional competence in food-waste prevention a

.926

.930

.655

C1 Hygiene knowledge

.80

C2 Cooking knowledge

.85

C3 Personal moral attitude

.72

C4 Attitude towards ingredients

.81

C5 Education and training attitude

.83

C6 Dish analysis skills

.81

C7 Planning and management skills

.84

S. Self-efficacy

.907

.910

.670

S1. I can set a goal to avoid food waste, and I think I can get the desired result.

.76

S2. I will be able to successfully overcome several challenges to avoid food waste.

.83

S3. I believe that I can effectively perform several different tasks to avoid food waste.

.84

S4 Compared with others, I can complete most tasks well to avoid food waste.

.82

S5. Even if I encounter difficulties in food-waste prevention, I can try my best to achieve the goal.

.84

I. Sustainable behavioral intentions during food preparation

.879

.843

.696

I1. I am willing to take food preparation activities for a healthy and sustainable diet.

.78

I2. I will encourage others to participate in healthy and sustainable food preparation together.

.86

I3. I will adjust and change my food preparation habits to promote sustainable positive effects of food preparation on health and the environment.

.86

B. Sustainable preparation promotion behavior

.850

.856

.600

B1. I implement green procurement (e.g., purchase local food, support green diet).

.65

B2. I implement and supervise others to reduce food waste in preparation.

.77

B3. I participate in the action of a healthy and sustainable diet.

.89

B4. I support the sustainable food/snacks activities organized by the government.

.77

a Twenty-nine items measured for professional competence load onto seven factors.

Table 3 Relevance among various dimensions and the correlations among all the constructs

Variable

Average mean

Standard deviation

E1

C

S

I

B

E. Ethical sustainability

4.10

.58

.8182

C. Professional competence in food-waste prevention

4.39

.55

.459

.789

S. Self-efficacy

4.12

.51

.712

.497

.806

I. Sustainable behavioral intentions during food preparation

4.23

.61

.680

.537

.665

.844

B. Sustainable preparation promotion behavior

4.13

.61

.680

.519

.721

.755

.777

**p < 0.01; N =351

1E: Ethical sustainability; C: Professional competence in food-waste prevention; S: Self-efficacy; I: Sustainable behavioral intentions during food preparation; B: Sustainable preparation promotion

2The square root of the AVE for discriminant validity is italicized along the diagonal.

Figure 3. Path analysis of each construct for overall structure model

(E: Ethical sustainability; C: Professional competence in food-waste prevention; S: Self-efficacy; I: Sustainable behavioral intentions during food preparation; B: Sustainable preparation promotion)

Table 4 The total, indirect, and direct effects of the hypothetical model

Bootstrapping

Estimate

Bias-corrected

percentile 95% CI

Lower

Upper

Two-tailed

significance

Indirect effect

Ethical sustainability→Professional competence in food-waste prevention →Self-efficacy

.329

.224

.432

.007**

Professional competence in food-waste prevention →Self-efficacy→ Sustainable behavioral intentions during food preparation

.266

.181

.397

.004**

Direct effect

Ethical sustainability→Professional competence in food-waste prevention

.510

.385

.603

.019*

Ethical sustainability→Self-efficacy

.140

.032

.277

.021*

Professional competence in food-waste prevention →Self-efficacy

.645

.483

.740

.009**

Professional competence in food-waste prevention → Sustainable behavioral intentions during food preparation

.425

.250

.617

.006**

Self-efficacy→ Sustainable behavioral intentions during food preparation

.413

.245

.577

.018*

Sustainable behavioral intentions during food preparation→ Sustainable preparation promotion

.798

.721

.859

.009**

Total effect

Ethical sustainability→Sustainable preparation promotion

.328

.234

.420

.018*

  1. Conclusion and Discussion

5.1 Conclusion

According to the data analysis results, the dimensions of ethical sustainability, professional competence in food-waste prevention, self-efficacy, sustainable preparation intention, and sustainable preparation promotion were all found to be significantly and positively correlated. This study first analyzed the CFA among the various dimensions, and the results showed that this model had a good fit. All indices passed the standards, indicating a compliant fit. This model could interpret the impact of each dimension. Ethical sustainability positively affected professional competence in food-waste prevention and self-efficacy, professional competence in food-waste prevention positively affected self-efficacy and sustainable behavioral intentions during food preparation. Similarly, self-efficacy positively affected sustainable behavioral intentions during food preparation, and sustainable behavioral intentions during food preparation significantly and positively affected sustainable preparation promotion. Therefore, H1, H2, H3, H4, H5, and H6 all had empirical support. Moreover, professional competence in food-waste prevention was the factor mediating ethical sustainability to sustainable behavioral intentions during food preparation, while self-efficacy was the factor mediating professional competence in food-waste prevention to sustainable behavioral intentions during food preparation. Both H7 and H8 were supported.

5.2 Discussion

Ethical sustainability positively affected professional competence in food-waste prevention and self-efficacy, while professional competence in food-waste prevention was the mediating factor between ethical sustainability and sustainable behavioral intentions during food preparation. Edwards and Mercer [70] strongly suggested that lifestyle choices, such as diet and career choices, had an ethical basis. Waston and Meah [32] explained that the primary driver for lower food waste was an “ethic of thrift”, which was the ethical motivation to act appropriately and “be thrifty”. The ethical rule for food waste behavior was "don't waste food". Therefore, the core ethical commitment in the study was to avoid wasting food, thus making consumption more reasonable [39]. Therefore, kitchen staff must have a personal imperative to prevent food waste in order to establish relevant behaviors and build professional competence in that arena.

Based on social cognitive theory in the context of food waste, females were more cognitively advanced than males. Females more closely monitored reducing food waste, so they were more motivated [71]. This contributed to their identification of more motivational factors that could have had a positive effect on their behavior. The environment affected the individuals and, in turn, their behavior. If actual catering producers were more aware of food waste, it could help enhance their professional competence in food-waste prevention.

The research on food waste has shown that preventing food waste required different culinary skills and knowledge, so leftovers food could be re-purposed [72, 73]. During storage, leftovers were often misplaced or left in the refrigerator for too long and spoiled [14, 15, 17]. Therefore, professional competence in food preparation is important.

 Professional competence in food-waste prevention positively affected self-efficacy and sustainable behavioral intentions during food preparation. Self-efficacy was the mediating factor between professional competence in food-waste prevention and sustainable behavioral intentions during food preparation. Lin [74] defined knowledge-sharing as a culture of social interaction, and the composition of this culture included knowledge exchange, experiential inheritance, and mutual learning among employees because the growth of knowledge could increase self-efficacy. Ding and Huang [75] suggested that mutual assistance, cooperation, and knowledge-sharing were critical for obtaining a competitive advantage. Bandura [20] suggested that individuals with high self-efficacy could have more enthusiasm, broader vision, broader thinking, and thus, more courage to accept challenges, and these behaviors were predominantly guided by the successful experiences of others. Self-efficacy and behavioral intentions could, indeed, predict food handling behaviors [30].

 Self-efficacy in processing food was important and correlated with general self-efficacy. Therefore, relevant educational policies should support consumers in developing self-efficacy related to sustainable food handling. Previous research pointed to the need for self-efficacy to enhance knowledge and skills, yet building self-efficacy required the inclusion of food handling and cooking classes in school curricula based on real-world experiences and the social marketing of public policy [45, 56, 58,  64].

Self-efficacy could be conceptualized with different levels of linkages, ranging from general self-efficacy (a person's ability to exercise self-control in different situations) to domain-specific self-efficacy (related to a person's ability to control behavior in specific settings (such as work and home)) to task-related self-efficacy (such as confidence in being able to correctly sort leftover waste) [76]. Domain- and task-related self-efficacy was positively correlated with general self-efficacy [77]. We explored the different hierarchies of self-efficacy in food waste in order to provide insight into the root causes of food waste. Therefore, we investigated the causes of household food waste by focusing on the effect of self-efficacy on consumer food waste [76]. Therefore, enhancing professional competence in food-waste prevention could improve self-efficacy and help promote more sustainable practices around food preparation and surplus food.  

5.3 Management Implications

This study shed light the factors influencing food waste prevention in the catering industry and could help the government and catering stakeholders to made specific recommendations for promoting food-waste prevention. The results showed that the ethical sustainability attitudes of kitchen staff affected professional competence in food-waste prevention and self-efficacy. Ethical sustainability among kitchen staff was related to their individual experiences and work/education environments. Therefore, the awareness and education provided by companies and governments concerning food-waste prevention could indirectly affect the concept of ethical sustainability among kitchen staff. Influencing the knowledge, attitudes, and skills of professional competence in food-waste prevention, and continuously promoting better strategies for surplus food, are important factors for governmental and corporate sustainability promotion. Self-efficacy indicates the individual believes that they can successfully perform a certain task or challenge, and they have the self-confidence to complete the task successfully [18, 23]. We need more consumers who are aware of and committed to food-waste prevention, and we need strong policymakers with the right strategies to promote food-waste prevention systematically. In addition, the cultivation of professional competence in food-waste prevention could affect self-efficacy and could have a significant impact on behavioral intentions through self-efficacy. For example, a company could regularly hold competitions for the creative preparation of surplus food in order to encourage self-reflection in catering practitioners [78]. Competitions or mutual internal audits of surplus food and food-waste prevention among catering practitioners could be helpful to understand limitations and share effective approaches, which could then further improve the reduction of food waste company-wide.

5.4 Practical Implications

It has been shown that the greater the professional competence of employees, the greater the increase in self-efficacy. Sustainability issues have become an important consideration in the catering industry in recent years. The responsibility and prevention of food waste are not only related to consumers but, rather, are closely related to the policies and factors that govern the entire supply chain. It is an encouraging sign that food-waste prevention is now a major socio-economic issue globally.

Implicit traits could be adjusted through a longer period of education and training, psychological counseling, or accumulated practical training [37]. According to our results, the level of performance depends on the level of self-efficacy and professional competence. More time must be spent on the psychological development of professional competence before effectiveness can be achieved. Related courses such as sustainability ethics, workplace ethics, and professional ethics are also indispensable components. In these courses, such topics can be discussed via case studies, peer experience sharing, and industry experience seminars for exploring individual perceptions that contribute to the improvement of self-efficacy and overall behavior motivation.

Kitchen staff are crucial for food preparation and management in the catering industry. The knowledge–attitude–behavior model [79] suggested that increased environmental knowledge could enhance environmental awareness and concern, leading to more environmentally conscious behaviors. This knowledge also generated positive thoughts, concerns, and practices [80]. In other words, the environment changed the person, and only personal changes could lead to changes in attitudes towards surplus food and more sustainable efforts when processing ingredients and preparing food. These could, then, aid the catering industry in food-waste prevention, sustainable ethics, the sustainable use of ingredients, sustainable cooking, and environmental protection.

5.5 Research Limitations and Future Studies

There were some limitations that should be considered. First, we primarily focused on those working in hotels and restaurants, as well as those working for group-meal providers, as a whole. To understand the differences in their environments, they should be considered separately. In the future, chain restaurants and central catering factories may be considered as the subjects, and different results may be obtained since other products in these catering sectors have been more standardized.

Due to the survey method used to collect data at a specific point in time, the results may be somewhat biased because the research instrument was a self-report survey [81]. Therefore, future studies should design more rigorous research procedures, preferably collecting data on predictor and outcome variables from different respondents. Finally, this study involved quantitative research, which could be supplemented by qualitative research in the future to understand issues associated with its implementation. We could have collected more factors affecting food waste in order to explore the actual situation of sustainable preparation behavior, such as the types of service and the ranks of kitchen staff, as well as to discuss the factors regulating sustainable preparation behavior.

References

  1. Amicarelli, V., Lagioia, G., and Bux, C. (2021). Global warming potential of food waste through the life cycle assessment: An analytical review. Environmental Impact Assessment Review, 91(11), 106677.
  2. Food and Agriculture Organization (FAO) (2022). Food Losses and Food Waste. Available online: http://www.fao.org/food-loss-and-food-waste/flw-data) (accessed on 10 Oct, 2022).
  3. Tarasuk, V., & Eakin, J. (2005). Food Assistance Through "Surplus" Food: Insights from an Ethnographic Study of Food Bank Work. Agriculture and Human Values, 177-186.
  4. WRAP (2022). Food Surplus and Waste in the UK Key Facts. Available online: https://wrap.org.uk/resources/report/food-surplus-and-waste-uk-key-facts (accessed on 20 Aug, 2022).
  5. Jungowska, J., Kulczy´ nski,B., Sidor, A., Gramza-MichaÅ‚owska, A. (2021). Assessment of Factors Affecting the Amount of Food Waste in Households Run by Polish Women Aware of Well-Being. Sustainability, 13, 976. https://doi.org/ 10.3390/su13020976
  6. Foley, J. A., Ramankutty, N., Brauman, K. A., Cassidy, E. S., Gerber, J. S., Johnston, M., & Zaks, D. P. M. (2011). Solutions for a cultivated planet. Nature, 478(7369), 337–342.
  7. Willett, W., Rockström, J., Loken, B., Springmann, M., Lang, T., Vermeulen, S., Garnett, T., Tilman, D.,Declerck, F., Wood, A., Jonell, M., Clark, M., Gordon, L., Fanzo, J., Hawkes, C., Zurayk, R., Rivera, J., Vries, W., Sibanda, L., & Murray, C. (2019). Food in the Anthropocene: the EAT–Lancet Commission on healthy diets from sustainable food systems. The Lancet. 393.(10170). 447-492.
  8. Stenmarck, Â., Jensen, C., Quested, T., Moates, G., Buksti, M., Cseh, B., & Scherhaufer, S. (2016). Estimates of European food waste levels. IVL Swedish Environmental Research Institute.
  9. Stirnimann, A. and Zizka, L. (2022). Waste not, want not: Managerial attitudes towards mitigating food waste in the Swiss-German restaurant industry. Journal of Foodservice Business Research, 25(3), 302–328
  10. McAdams, B., Massow, M., Gallant, M., and Hayhoe, M.A. (2019). A cross industry evaluation of food waste in restaurants. Journal of Foodservice Business Research, 22(5), 449–466.https://doi.org/10.1080/15378020.2019.1637220
  11. Ganglbauer, E., Fitzpatrick, G., Comber, R. (2013). Negotiating food waste: using a practice lens to inform design. ACM Trans. Computer Human Interaction, 20, 1-25.
  12. Graham-Rowe, E., Jessop, D.C., and Sparks, P. (2014). Identifying motivations and barriers to minimizing household food waste. Resources, Conservation and Recycling, 84, 15-23.
  13. Wang, Y.F. (2016). Improving Culinary Education by Examining the Green Culinary Behaviors of Hospitality College Students. Journal of Hospitality & Tourism Education, 28(1), 1-9, DOI: 10.1080/10963758.2015.1127167
  14. Farr-Wharton, G., Foth, M., Choi, J.H.J. (2014). Identifying factors that promote consumer behaviours causing expired domestic food waste. Journal of Consumer Behavior,13, 393-402.
  15. Waitt, G., Phillips, C. (2016). Food waste and domestic refrigeration: a visceral and material approach. Social & Cultural Geography, 17, 359-379.
  16. Secondi, L., Principato, L., and Laureti, T. (2015). Household food waste behaviour in EU- 27 countries: a multilevel analysis. Food Policy, 56, 25-40.
  17. Blichfeldt, B.S., Mikkelsen, M., and Gram, M.(2015). When it stops being food: The edibility, ideology, procrastination, objectification and internalization of household food waste. Food, Culture & Society, 18, 89-105.
  18. Bandura, A. (1982). Self-efficacy mechanism in human agency. American Psychologist, 37, 122-147.
  19. Wood, R., & Bandura, A. (1989). Impact of conceptions of ability on self-regulatory mechanisms and complex decision making. Journal of Personality and Social Psychology, 56,407-415.
  20. Bandura, A. (1977). Self-Efficacy: Toward a Unifying Theory of Behavioral Change. Psychological Review, 84(2), 191-215.
  21. Gist, M. E., & Mitchell, T. R. (1992). Self-efficacy: A theoretical analysis of its determinants and malleability. Academy of Management Review, 17, 183-211.
  22. Stajkovic, A. D., & Luthans, F. (1998). Self-efficacy and work-related performance: A metaanalysis. Psychological Bulletin, 124, 240-261.
  23. Bandura, A. (1986). Social foundations of thought and action. Englewood Cliffs, NJ: Prentice-Hall.
  24. Garrone, P., Melacini, M., & Perego, A. (2013). Feed the Hungry: The Potential of Surplus Food Recovery. Milan, Italy: Edizioni AngeloGuerini e Associati SpA.
  25. Parfitt, J., Barthel, M., & Macnaughton, S. (2010). Food waste within food supply chains: Quantification and potential for change to 2050. Philosophical Transactions of the Royal Society B, 365, 3065–3081.
  26. Göbel, C., Langen, N., Blumentha, A., Teitscheid, P., and Ritter, G. (2015). Cutting Food Waste through Cooperation along the food Supply Chain. Sustainability, 7, 1431-1438.
  27. Hawkes, C., & Webster, J. (2000). Too much and too little? debates on surplus food redistribution. London: Sustain.
  28. Bandura, A. (1977). Social learning theory. Englewood Cliff, NJ: Prentice Hall.
  29. Rimal, R.N. (2000). Closing the knowledge-behavior gap in health promotion: The mediating role of self-efficacy. Health Communication, 12(3), 219–237.
  30. Bearth, A., Cousin, M. -E., & Siegrist, M. (2014). Investigating novice cooks' behavior change: Avoiding cross-contamination. Food Control, 40, 26–31.
  31. Mullan, B., Allom, V., Fayn, K., and Johnston, I. (2014). Building habit strength: A pilot intervention designed to improve food-safety behavior. Food Research International, 66, 274-278.
  32. Watson, M., & Meah, A. (2012). Food, waste and safety: Negotiating conflicting social anxieties into the practices of domestic provisioning. The Sociological Review, 60(S2), 102–120.
  33. Goldstein, N. J., Cialdini, R. B., & Griskevicius, V. (2008). A room with a viewpoint: Using norms to motivate environmental conservation in hotels. Journal of Consumer Research, 35, 472-482. doi:10.1086/586910
  34. Glendinning, C. (1994). My name is Chellis and I’m in recovery from Western civilization. Gabriola Island: New Catalyst Books.
  35. Abram, D. (1996) The spell of the sensuous. New York: Vintage Books.
  36. MacMillan, T. (2009) What is wrong with waste? Food Ethics, 4 (3), p. 4.
  37. Stancu, V., Haugaard, P. and Lähteenmäki, L. (2016) Determinants of consumer food waste behaviour: two routes to food waste. Appetite, 96, 7-17.
  38. Gjerris, M., & Gaiani, S. (2013). Household food waste in Nordic countries:Estimations and ethical implications. Nordic Journal of Applied Ethics, 7 (1),6-23.
  39. Lehtokunnas, T., Mattila, M., Narvanen, E., and Mesiranta, N. (2020). Towards a circular economy in food consumption: Food waste reduction practices as ethical work. Journal of Consumer Culture, https://doi.org/10.1177/1469540520926252
  40. Lacasse, K. (2016). Don’t be satisfied, identify! Strengthening positive spillover by connecting pro-environmental behaviors to an “environmentalist” label. Journal of Environmental Psychology, 48, 149-158. doi:10.1016/j.jenvp.2016.09.006
  41. van der Werff, E., Steg, L., & Keizer, K. (2014). Follow the signal: When past proenvironmental actions signal who you are. Journal of Environmental Psychology, 40, 273-282. doi:10.1016/j.jenvp.2014.07.004
  42. Whitmarsh, L., & O’Neill, S. (2010). Green identity, green living? The role of pro-environmental self-identity in determining consistency across diverse proenvironmental behaviours. Journal of Environmental Psychology, 30, 305-314. doi:10.1016/j.jenvp.2010.01.003
  43. Cialdini, R. B., Kallgren, C. A., & Reno, R. R. (1991). A focus theory of normative conduct: A theoretical refinement and reevaluation of the role of norms in human behavior. Advances in Experimental Social Psychology, 24, 201-234. doi:10.1016/S0065-2601(08)60330-5
  44. Harrington, R.J., Mandabach, K.H., VanLeeuwen, D., and Thibodeaux, W. (2005). A multi-lens framework explaining structural differences across foodservice and culinary education. International Journal of Hospitality Management, 24, 198-218.
  45. Hegarty, J. A., & O’Mahony, G. B. (2001). Gastronomy: A phenomenon of cultural expressionism and an aesthetic for living. International Journal of Hospitality Management, 20, 3-13.
  46. Agut, S., Grau, R., and Peiro, J.M. (2003). Competency needs among managers from Spanish hotels and restaurants and their training demands. International Journal of Hospitality Management, 22, 281-295.
  47. Ko,W.H. and Lu, M.Y. (2021). Developing a professional competence scale for kitchen staff: Food value and availability for surplus food. International Journal of Hospitality Management, 95, 102926.
  48. Tekin, Ö.A.; Ilyasov, A. (2017). The Food Waste in Five-Star Hotels: A Study on Turkish Guests’ Attitudes. Journal of Tourism and Gastronomy Studies, 5, 13–31.
  49. Ajzen, I. (2002) Perceived behavioral control, self-efficacy, locus of control, and the theory of planned behavior. Journal of Applied Social Psychology, 32(4), 665-683.
  50. Ajzen, I. and Fishbein, M. (1980). Understanding Attitudes and Predicting Social Behavior, Prentice-Hall, Englewood Cliffs, NJ.
  51. Ajzen, I. (1991). The theory of planned behavior. Organizational Behavior and Human Decision Processes, 50(2), 179-211.
  52. Chan, E.S., Hon, A.H., Chan, W. and Okumus, F. (2014). What drives employees’ intentions to implement green practices in hotels? the role of knowledge, awareness, concern and ecological behavior. International Journal of Hospitality Management, 40, 20-28.
  53. Spencer, L. M., & Spencer, S. M. (1993). Competence at work- Models for Superior Performance. NY: John Wiley & Sons.
  54. Romani, S., Grappi, S., Bagozzi, R.P. and Barone, A.M. (2018) Domestic food practices. A study of food management behaviors and the role of food preparation planning in reducing waste. Appetite, 121, 215-227.
  55. Lavelle, F., McGowan, L., Spence, M., Caraher, M., Raats, M.M., Hollywood, L., McDowell, D., McCloat, A., Mooney, E. and Dean, M. (2016) Barriers and facilitators to cooking from 'scratch' using basic or raw ingredients. A qualitative interview study. Appetite, 107, 383-391.
  56. Thyberg, K.L. and Tonjes, D.J. (2016) Drivers of food waste and their implications for sustainable policy development. Resources, Conservation and Recycling, 106, 110-123.
  57. Hebrok, M. and Boks, C. (2017) Household food waste. Drivers and potential intervention points for design—an extensive review. Journal of Cleaner Production, 151, 380-392.
  58. Schanes, K., Dobernig, K. and Gözet, B. (2018) Food waste matters—a systematic review of household food waste practices and their policy implications. Journal of Cleaner Production, 182, 978-991.
  59. Andreasen, A.R. (2002) Marketing social marketing in the social change marketplace. Journal of Public Policy & Marketing, 21(1), 3-13.
  60. Lindeman, M. & Väänänen, M. (2000). Measurement of ethical food choice motives. Appetite, 34(1),55-59.
  61. Chen, G., Gully, S.M., Eden, D. (2001).Validation of a New General Self-Efficacy Scale. Organizational Research Methods, 4(1), 62-83.
  62. Fang, W.T., Ng, E., Wang, C.M., and Hsu, M.L. (2017). Normative Beliefs, Attitudes, and Social Norms: People Reduce Waste as an Index of Social Relationships When Spending Leisure Time. Sustainability, 9, 1696.
  63. Teng, C.C., Chih, C. and Wang, Y.C. (2020). Decisional Factors Driving Household Food Waste Prevention: Evidence from Taiwanese Families. Sustainability, 12, 6666; doi:10.3390/su12166666.
  64. Anderson, J.C. and Gerbing, D.W. (1988). Structural equation modeling in practice: a review and recommended two-step approach. Psychological Bulletin, 103(3), 411-423.
  65. Hair, J.F., Anderson, R.E., Tatham, R.L. and Black, W.C. (2010). Multivariate Data Analysis, 7th ed., Prentice Hall, Englewood Cliffs, NJ.
  66. Hu, L.T. and Bentler, P.M. (1999). Cutoff criteria for fit indexes in covariance structure analysis: conventional criteria versus new alternatives. Structural Equation Modeling: A Multidisciplinary Journal, 6(1), 1-55.
  67. Jöreskog, K.G. and Sörbom, D. (1989), LISREL 7: A Guide to the Program and Applications, SPSS, Chicago, IL.
  68. Fornell, C. and Larcker, D.F. (1981). Evaluating structural equation models with unobservable variables and measurement error. Journal of Marketing Research, 18(1), 39-50.
  69. Preacher, K.J. and Hayes, A.F. (2008). Asymptotic and resampling strategies for assessing and comparing indirect effects in multiple mediator models. Behavior Research Methods, 40, 879-891.
  70. Edwards, F. & Mercer, D. (2007). Gleaning from Gluttony: an Australian youth subculture confronts the ethics of waste. Australian Geographer, 38(3), 279-296.
  71. Goodman-Smith, F., Mirosa, R., and Mirosa, M. (2020). Understanding the E_ect of Dining and Motivational Factors on Out-Of-Home Consumer Food Waste. Sustainability, 12, 6507.
  72. Cappellini, B. (2009). The sacrifice of re-use: The travels of leftovers and family relations. Journal of Consumer Behavior, 8, 365-375.
  73. Southerton,D. & Yates,L. (2015). Exploring food waste through the lens of social practice theories: Some reflections on eating as compound practice. In: Ekstrom KM (ed.) Waste Management and Sustainable Consumption: Reflections on Consumer Waste. London; Chicago, IL: Routledge, 133-149.
  74. Lin, H. F. (2007). Effects of extrinsic and intrinsic motivation on employee knowledge sharing intentions. Journal of Information Science, 33(2), 135-149.
  75. Ding, X. H., & Huang, R. H. (2010). Effects of knowledge spillover on interorganizational resource sharing decision in collaborative knowledge creation. European Journal of Operational Research, 201, 949-959.
  76. Grether, T., Sowislo, J.F. and Wiese, B.S. (2018) Top-down or bottom-up? Prospective relations between general and domain-specific self-efficacy beliefs during a work-family transition. Personality and Individual Differences,121, 131-139.
  77. Luszczynska, A., Scholz, U. and Schwarzer, R. (2005) The general self-efficacy scale: multicultural validation studies. The Journal of Psychology, 139(5), 439-457.
  78. Saks, A. M. (1995). Longitudinal field investigation of the moderating and mediating effects of self-efficacy on the relationship between training and newcomer adjustment. Journal of Applied Psychology, 80(2), 211–225. https://doi.org/10.1037/0021-9010.80.2.211
  79. Kollmuss, A. and Agyeman, J. (2002). Mind the gap: why do people act environmentally and what are the barriers to pro-environmental behavior?. Environmental Education Research, 8(3), 239-260.
  80. Ruiz-Molina, M.E. and Gil-Saura, I. (2008). Perceived value, customer attitude and loyalty in retailing. Journal of Retail & Leisure Property, 7(4), 305-314.
  81. Podsakoff, P.M., MacKenzie, S.B., Lee, J.Y. and Podsakoff, N.P. (2003). Common method biases in behavioral research: a critical review of the literature and recommended remedies. Journal of Applied Psychology, 88(5), 879-903.

Reviewer 2 Report

Comments and Suggestions for Authors

Dear authors,
I regret that this version of the work cannot be revised. It must be rechecked by the editors.

Kind regards

Author Response

To Editor-in-Chief and reviewers:

Thank you for your comments and suggestions. I appreciate the time and effort that you and the reviewers have dedicated to providing your valuable feedback on my manuscript. I had major revised in this paper as final attached. Please see our comments below.

The following are Reviewer comments:

Overall

  1. The manuscript needs English editing.
  2. The references are not according to MDPI guidelines.

Ans: I had made major revise in all sections.

I had revised paper due to English language editing by MDPI. Please see the certification as below, and I also to check MDPI guideline to check all paper. Please see the revised paper. And, I also change title as “Sustainable Preparation Behavior for Kitchen Staff in order to Limit Food Waste”.

The following paper are major revised to the Reviewers’ 2 comments.

Sustainable Preparation Behavior for Kitchen Staff in order to Limit Food Waste

Abstract

The concepts of culinary sustainability and avoiding the waste of surplus food have become important sustainability trends today. How to integrate the handling of surplus food in the catering industry is a topic of concern in the industry. Kitchen staff are the vital soul of any restaurant, and we intend to discuss how kitchen staff actually behave and to explore factors that influence their behaviors in order to develop an implementation model for food-waste prevention. Therefore, this study explored a model of ethical sustainability, professional competence, self-efficacy, sustainable food preparation objectives, and sustainable food promotion and behavior focused on limiting food waste.  This study used a questionnaire and surveyed employees who had been employed for more than 6 months. From May to August 2022, 500 questionnaires were distributed; 415 valid questionnaires were retrieved, yielding a 90.2% recovery rate. According to the structural equation modeling analysis between the dimensions, ethical sustainability should have had a positive influence on professional competence in food-waste prevention and self-efficacy. Professional competence in food-waste prevention affected self-efficacy and behavioral intentions during food preparation; self-efficacy also significantly affected behavioral intentions towards sustainable food preparation. Similarly, behavioral intention had a positive influence on promoting sustainable behaviors. This model was a good fit for the acquired data. Professional competence in food-waste prevention was found to be the mediating factor between ethical sustainability and the behavioral intentions towards sustainable food preparation, and self-efficacy was the mediating factor between the professional competence in food-waste prevention and the behavioral intentions towards sustainable food preparation.

Keywords: surplus food, sustainable preparation behavior, food waste, kitchen staff, ethical sustainability, self- efficacy

  1. Introduction

Food is an important daily necessity. With advancements in the global economy, equipment, and food production efficiency, an abundant supply of food can provide for more people. From production to consumption, food is often discarded by consumers even though it is safe to eat because they have bought too much or have insufficient knowledge about the shelf life of food. It is estimated that one-third (1.3 billion tons) of the food produced globally for consumption is wasted every year [1,2]. The problem of food waste has been included in the World Health Organizations’ sustainable development goals (SDG 17) in order to reduce global human food waste by half and to reduce food loss by 2030. Food service and family experiences have been shown to be an appropriate way to address national and international sustainable goals [1].

The problem of surplus food management has also attracted significant attention from academics and other concerned practitioners because the normal dietary sources of a population are often interrupted due to limited resources, and experts believe that it is possible to reduce food insecurity via surplus food management. In addition to the commercial economic losses due to food production and waste, the social and environmental losses have also been significant, especially when foods that were still edible were not able to be distributed and sold, for various reasons, and were eventually discarded [3].

The Food and Agriculture Organization (FAO) of the United Nations has estimated that the annual global food waste across the entire food chain was USD 2.6 trillion. The Waste and Resources Action Plan estimated that food waste in the UK totaled around 10 million tons per year, 70% of which was avoidable [4]. Food waste is an urgent and serious issue in Western countries. The increase in the amount of food waste directly affects environmental changes, including greenhouse gas emissions and packaging pollution. In Poland, 9.2 million tons of grain have been lost every year, 53% of which was produced by consumers. In order to reduce consumer food waste, it was necessary to understand the behavioral factors [5]. The key problem of food waste has been primarily due to the supply chain, from production, preparation, and processing to transportation and consumption, all of which require considerable costs and resources. Therefore, practitioners and consumers should pay close attention to this issue from both the supply and the demand side. Reducing food waste is an important factor for sustainable development. The predicted population growth will require an adequate food system that includes the sustainable development of natural ecological resources [6,7]. Food waste occurs at all stages of the food supply chain, and the further down the supply chain, the greater the negative environmental, economic, and social impacts of food waste [8].

In restaurants, government intervention and enhanced waste disposal collection are necessary. Research has shown that changing consumer behavior, combined with corporate social responsibility, mitigated food waste [9]. The results of the study showed that many restaurants have prioritized reducing food waste. However, most of the final factors affecting food waste in restaurants were quality and specification issues during produ711ction. Generally, food waste was the worst in casual restaurants, and large meal portions resulted in more food waste [10].

The chefs are the souls of a restaurant. In addition to making good dishes, a good chef can also improve the product value while maintaining their social responsibility. Food oversupply has also been related to consumers' perceived concept of preservation. Consumers may purchase more than they need in order to save time [11, 12]. Among these food behaviors, discarding reusable ingredients as kitchen waste was the most severe (28.8% of ingredients were considered kitchen waste and discarded) [13]. Storing and sorting food in an orderly manner (for example, stacking old and new food, or sorting by frequency of use), and reordering regularly could reduce food waste [14, 15]. According to Secondi, Principato, and Laureti [16], one of the best courses of action to avoid wasting food was to estimate portion sizes correctly. Misplaced, forgotten, or expired leftovers were an issue when storing food [17].

Social cognitive theory (SCT) explains human behavior based on the dynamic interactions of individual experiences, the actions of others, and environment influences [18]. Self-efficacy is the belief in one's ability to change motivations, cognitive resources, and courses of action in order to meet the needs of a particular situation [19-22]. Research has found that self-efficacy predicted several job-related outcomes, such as work attitudes, training proficiency, and job performance. Bandura [23] pointed out that self-efficacy was the result of interactions with the external environment, personal professional competency, and learning effectiveness, which could then motivate behavior. Kitchen staff play a key role in food production from the kitchen to the table, and a sustainability-focused food revolution should be initiated for reducing carbon emissions and mitigating food waste to preserve the planet’s ecosystem.

Therefore, the attitudes of kitchen staff to avoid food waste could affect how they handle food, and sustainable cooking could provide substantial results in the future for environmental and food sustainability. This study explored the perspectives of kitchen staff towards limiting food waste by evaluating the behavioral patterns such as ethical sustainability, professional competence in food-waste prevention, self-efficacy, behavioral intentions towards sustainable food preparation, and sustainable food promotion from a practical perspective. This research could help promote sustainable cooking and food preparation, from the individual to the restaurant, and then to the catering industry. By encouraging consistent sustainability in the catering industry, we can more effectively reduce food waste.

  1. Theoretical background and hypotheses development

2.1 Surplus food and food waste

The conceptual model of availability, surplus, recoverability, and waste (ASRW) provided a clear definition of surplus food and distinguished it from food waste by introducing the concepts of availability, surplus, recoverability, and waste. According to this model, available food was defined as "all food in the food supply chain", which consisted of three parts: "human consumption", "surplus food", and "food scraps”. There has been no universally accepted definition of food waste and surplus food because there have been different interpretations from different perspectives [24].

According to the FAO, food waste consisted of three distinct problems. Food loss (FL) was the reduction in weight or nutritional value of food originally intended for human consumption. These losses were primarily due to processes in the supply chain, such as poor management, errors, and irregular processes (such as improper farming and harvesting), resulting in reduced edible weight. Food waste (FW) was the discarding of edible food, whether expired or spoiled, due to reasons such as oversupply, lack of a sales strategy, or excessive or unplanned purchases by individual consumers. It was also the waste of edible food due to improper distribution, storage, transportation, and preparation, both in the home and in restaurants [25].

Moreover, food waste occurs at all stages of the food chain: production, distribution, retail, and consumption processes [26]. Types of waste at all stages of the food chain are shown in Figure 1. Wasted food is any food that is lost due to spoilage or being discarded. "Waste food" as opposed to "food waste" refers to food that has not been used for its intended purpose but could be further used as a valuable resource [2].

Figure 1. Food waste styles in all stages of the food chain

According to scholars [27], surplus food could be divided into four categories: (1) Unprofitable crops refer to the market phenomenon after the harvest of agricultural products, and the surplus of grain sales is generated due to a stable sale price. When production is much higher than demand, it can lead to an oversupply in the market. (2) Non-perishable processed foods are foods with a long shelf life (such as dry goods, canned foods, etc.), damaged packaging, incorrect packaging, promotions near the expiration date, etc. (3) Perishable fresh foods are fresh fruits and vegetables, bread, dairy products, refrigerated ready-to-eat foods, etc.; products that are not sold because they are close to their expiration date but are still safe to eat; and foods that agricultural producers and importers may not be able to sell because the fruit or vegetables are irregular in shape and size. (4) Perishable foods include ready-to-eat foods such as sandwiches, cakes, and pastries [27].

2.2 Social Cognitive Theory

Social cognitive theory (SCT) was first proposed by Bandura, who combined the concepts of behaviorism and social learning [18]. Behavior is generated by the interaction between the individual and the environment, rather than determined by any single aspect. These factors include environmental impacts (the overall social environment, organizational policies, and culture); individual cognition and personal factors (personal motivation, attitude); and behavior (intention and willingness), all of which interact. Figure 2 shows the dynamic formation of the person, the environment, and behavior in SCT, where personal behavior is affected by the environment and influenced by personal subjective cognitions. The psychological process of self-introspection is achieved through the inner driving force of the individual, and the person feels and reacts to their external environment due to their inner thoughts, rather than being stimulated by the environment alone. Therefore, this research hypothesized that environmental factors could influence personal factors, and personal factors could influence behavioral factors.

 Figure 2. Interactive model of social cognitive theory

Bandura [28] suggested that persons with high self-efficacy could have more enthusiasm, broader vision, broader thinking, and, therefore, courage to accept challenges. To avoid repeating the failures of others, most individuals would use the success of others as a guide. Rimal [29] also pointed out that SCT was an important basis for predicting food handling processes. Intention and self-efficacy could predict food-handling behavior [30]. Bandura [23] also suggested that personal past performance affected self-efficacy, and higher self-efficacy could also affect future performance. Behavioral improvements could be enhanced by offering continuous reminders during food handling [31].

Bandura [23] considered SCT the most suitable for explaining dynamic human behavior and regarded cognition and self-regulation as the primary framework of causality. The development of this model could cultivate the confidence of individuals to use certain abilities. However, it could also be affected by many internal factors, such as the level of self-efficacy, which would then result in different behaviors. Self-efficacy refers to a person's belief in their own capabilities and their degree of confidence in being able to achieve a specific outcome.

2.3 Ethical Sustainability Affects Personal Behavior

Waston and Meah [32] indicated that even if consumers had an environmental motivation to avoid food waste, the real motivation to reduce food waste was the "ethic of thrift". People avoided wasting food because it caused feelings of guilt that were then combined with environmental concerns [17]. Food represented the physical and symbolic connection between humans and nature [33]. A strong negative attitude towards food waste could be explained by our respect for nature, and this could hopefully be extended to the community [34].

Therefore, food waste could be perceived as a waste of precious resources, which would be unfair from an anthropocentric perspective [35]. Food waste has been considered immoral, so people need to be called upon to change their behaviors and attitudes concerning food waste [36, 37]. Therefore, food is a valuable resource, as it is not always readily available. A change in attitude could benefit both humans and nature by reducing the negative aspects of food production and improving the balance between humans and nature [38]. The “moral” rule for food waste behavior could be "don't waste food". With this in mind, our core ethical commitment was to avoid wasting food through more sensible consumption. Doing so could protect and preserve the world in which we live [39].

As indicated by the self-perception theory, previous behaviors affected future actions. The theory suggested that if individuals were encouraged to behave in environmentally conscious ways, they were more likely to make environmentally conscious decisions [40, 41]. This theory reinforced environmental self-identity, or an individual's perception of themselves as a person who is environmentally conscious [42]. In addition, many previous studies found that social norms and the behaviors of social groups had a positive influence on behavior [33,43].

Therefore, the following hypotheses were developed:

H1: Attitudes towards ethical sustainability affected professional competence in food-waste prevention.

H2: Attitudes towards ethical sustainability affected self-efficacy.

2.4 Relationships among professional competence, self-efficacy, and food-waste prevention

Food knowledge and skills, communication skills, business knowledge, and food concepts were considered the main professional abilities of kitchen staff [44]. To conform to social trends and achieve the following objectives, Hegarty and O'Mahony [45] developed the four major fields of culinary courses, including information; culinary arts; food and life sciences; and business. The basic skills needed for kitchen staff were food science, product preparation, finance, personal management, marketing, language, and computer skills [46]. Professional competence for food-waste prevention included two aspects: hygiene knowledge and culinary knowledge. Professional attitudes towards food-waste prevention included three dimensions, namely personal moral attitudes, attitudes towards food ingredients, and attitudes towards education and training. Skill competence comprised two dimensions: dish analysis and meal planning management [47]. In the restaurant and hospitality industry, both food preparers and consumers are responsible for food waste. Food waste has also been identified as a problem in schools, kindergartens, canteens, gas stations, bars, restaurants, canteens, and hotels [48].

The concept of self-efficacy first appeared in an article by Bandura [20]. It refers to an individual’s belief whether one can successfully complete a task. The emphasis was on ability, and it could promote individual actions in certain areas and affect future activities. This aspect was related to personal values. Bandura [18] also defined self-efficacy as “in order to deal with an impending situation, an individual makes a subjective judgment on whether he or she can successfully perform the required behavior. This judgment will decide how much effort he or she will put in when he or she personally faces difficulties, and how long it will last”. That is, self-efficacy could affect whether a person would persist in an action after setting goals. This was also related to whether a person had the motivation to overcome obstacles in the learning process and the learning strategies adopted when solving problems.

The theoretical basis of perceived behavioral control was originally derived from the self-efficacy theory, so perceived behavioral control and self-efficacy were conceptually similar [49]. Many studies have employed the theory of rational action (TRA) [50] or its extended theory of planned behavior (TPB) model [51] to understand kitchen workers' behavioral intentions towards food-waste prevention. The TRA theory suggested that previous attitude (i.e., approval or disapproval of a certain behavior) and existing subjective norms (i.e., whether the implementation of a certain behavior was approved by others) were two important influences on a person's behavioral intentions to implement a certain behavior [52]. Based on these previous studies, the current study aimed to understand whether ethical sustainability and professional competence in food-waste prevention were important indicators of behavioral intentions among kitchen staff. According to TRA and TPB, a person’s attitude indicated the favorable or unfavorable evaluation of a certain behavior, so it could predict behavioral intentions [51]. Therefore, the following hypotheses were developed:

H3: Professional competence could positively influence self-efficacy in food-waste prevention.

H4: Professional competence in food-waste prevention could positively influence the sustainable behavioral intentions during food preparation.

H5: Self-efficacy could positively influence sustainable behavioral intentions during food preparation.

H6: Sustainable behavioral intentions during food preparation could positively influence the promotion of sustainable preparation.

2.5 Mediating role of professional competence in food-waste prevention and self-efficacy

Based on the iceberg model proposed by Spencer and Spencer [53], professional competence could be classified as five basic factors: motivation, traits, self-concept, knowledge, and skills. Knowledge and skills were explicit traits: Both were easily affected by the external environment, both were trainable, and both could be enhanced through education and learning. Implicit traits, such as self-concept, traits, and motivation, could not be easily explored and developed and, therefore, would need to be inherent in the individual. The capacity for efficient food handling was the most important factor in preventing food waste [54]. Self-efficacy was important during food preparation and was also linked to preventing food waste [55].

There is a considerable relationship between the self-efficacy of food handling competence and general self-efficacy. Therefore, if you can enhance your personal self-efficacy, you can also enhance your competence in food handling. It was necessary to improve professional knowledge and skills, as it could improve self-efficacy and food handling behavior [56, 57, 58]. In addition, through food handling and cooking classes, self-efficacy could be enhanced, and behavioral ability could be improved [59]. Therefore, the following hypotheses were developed:

H7: Professional competence in food-waste prevention mediated the attitudes toward ethical sustainability and the sustainable behavioral intentions during food preparation.

H8: Self-efficacy mediated the professional competence in food-waste prevention and the sustainable behavioral intentions during food preparation.

  1. Methodology

3.1 Data Collection

A pilot study was conducted to ensure the reliability of the measurements. The first part of the questionnaire included questions related to all the variables of this study. The second part obtained the demographics of the participants, including gender, age, education, job title, work experience, and restaurant type.

To ensure the face validity of the questionnaire, we invited three professors to evaluate the validity of the predicted measurements. One hundred pilot-study questionnaires were returned with a valid return rate of 100%. The research process included item and factor analyses to determine the final questionnaire. All items in the questionnaire had good reliability (Cronbach's a > 0.70). In addition, according to the feedback of the participants, some items were revised.

This study included employees in Taiwan’s catering industry (including hotels, restaurants, and group-meal companies) as the research subjects. The convenience sampling and purposive sampling methods were adopted, and assistance was sought, through telephone interviews and e-mails, from individual practitioners in the catering industry who met the sampling criteria. From May to August 2022, questionnaires were sent and returned by mail. Out of 500 questionnaires, 415 valid questionnaires were returned, which yielded a 90.2% valid response rate.

3.2 Measures

  A questionnaire survey method was adopted for this study. All scales used to measure the constructs of this study were adopted from previous studies. The questionnaires used were divided into 6 parts: part 1, ethical sustainability, including 3 items; part 2, professional competence in food-waste prevention scale, including 29 items for 7 dimensions; part 3, self-efficacy, including 5 items; part 4, sustainable food behavioral intentions during food preparation, including 3 items; part 5, sustainable food promotion, including 4 items; and part 6, personal demographic.

For items measuring ethical sustainability, a three-item scale was adapted from the work of Lindeman and Väänänen [60]. This scale was based on a 5-point Likert scale, ranging from 1 (“strongly disagree”) to 5 (“strongly agree”). The Cronbach's α value of this scale was 0.832. The scale of professional competence in food-waste prevention was according to the professional competence in food-waste prevention scale of Ko and Lu [47], with 29 questions of 7 dimensions, and a 5-point Likert scale ranging from 1 (“strongly disagree”) to 5 (“strongly agree”). The average mean of the questions in each dimension was taken as the average mean for analysis, and the total Cronbach's α value of the scale was 0.926. To measure self-efficacy, 5 items from Chen, Gully, and Eden [61] were adapted and rated by a 5-point Likert scale, ranging from 1 (“strongly disagree”) to 5 (“strongly agree”). Finally, sustainable behavioral intentions during food preparation were measured using 10 items from Fang, Wang, and Hsu [62], and there were 3 questions. The Cronbach's α value of this scale was 0.907. The sustainable preparation promotion was adapted from the 4-item scale of Teng, Chih, and Wang [63]. The Cronbach's α value of this scale was 0.850. These items were rated by a 5-point Likert scale, ranging from 1 (“strongly disagree”) to 5 (“strongly agree”).

3.3 Data Analysis

The model comprised two components, a measurement model and a causal structural model. Before data analysis, we performed structural equation modeling (SEM) using the AMOS 21.0 application software. Whether the observed variables reflected the hypothesized latent variables and whether the measurement model adequately fit the data was performed with a confirmatory factor analysis (CFA). Path analysis was used in the study to evaluate the fit of the structural model and the hypothesized relationships among the latent variables [64]. As suggested by Anderson and Gerbing [64], we started with a CFA to determine whether all indicator variables adequately reflected their underlying structure and that the measurement model was acceptable. The fit index, comparative fit index (CFI), and the root-mean-square error of approximation (RMSEA) were used to estimate the overall model fit.

  1. Results

The subjects were mostly male in an age range of 21-40 years (70.1%). The education level was mostly high school and university. Non-management positions accounted for 71.8%, working experience of 1-4 years accounted for 28.2%, and 4-12 years accounted for 35.6%. The participants worked in hotel restaurants, independent restaurants, and restaurant chains (including central kitchens) (Table 1).

Table 1  Sociodemographic details of subjects

Background variables

Category

Number of persons

Percentage (%)

Gender

Male

217

61.8

Female

134

38.2

Age

20 years old or younger

8

2.3

21-30 years old

139

39.6

31-40 years old

107

30.5

41-50 years old

65

18.5

51-60 years old

27

7.7

61 years old or older

5

1.4

Educational level

Junior high school or below

14

4

High school ∕ vocational high school

130

37

Junior college

67

19.1

University

130

37

Graduate school

10

2.8

Job position

Management

99

28.2

Non-management

252

71.8

Working years

1 year or less

24

6.8

More than 1 year to 4 years (inclusive)

99

28.2

More than 4 year to 8 years (inclusive)

64

18.2

More than 8 year to 12 years (inclusive)

61

17.4

More than 12 year to 16 years (inclusive)

34

9.7

More than 16 year to 20 years (inclusive)

29

8.3

More than 20 years 

40

11.4

Employment

Restaurant in Hotel

120

34.2

Individual Restaurant

136

38.7

Group meal/central kitchen

82

23.4

Others

13

3.7

4.1 Measurement Model

The model fit indices then indicated an acceptable overall measurement model: x 2/df = 2.77; GFI = 0.87; AGFI = 0.84; RMSEA = 0.071; SRMR = 0.043; CFI = 0.98; and NFI = 0.97 [65]. Table 2 shows that all items had statistically significant (p < 0.01) [66] factor loadings, ranging from 0.65 to 0.89. The composite reliability values for all the constructs ranged from 0.88 to 0.95, indicating that all constructs had good internal reliability [67]. Furthermore, the average extracted variance (AVE) for each construct exceeded 0.50, which established the convergent validity of the measurement model [68]. Table 3 shows that the facets between all constructs were significant (p < 0.01), ranging from 0.459 to 0.755. In addition, the square root of AVE for each construct was also higher than the cross-correlation among all variables, so this model had discriminant validity [68].

4.2 Structural Model

The structural models were x 2/df = 3.10; GFI = 0.86; AGFI = 0.83; CFI = 0.98; NFI = 0.97; RMSEA = 0.077; and SRMR = 0.067, which indicated the good fit of the structural model [65]. The path analysis showed that environmental sustainability positively affected professional competence in food-waste prevention (b = 0.51, p < 0.000) and self-efficacy (b = 0.14, p < 0.000); professional competence in food-waste prevention positively affected self-efficacy (b = 0.65, p < 0.000) and sustainable behavioral intentions during food preparation  (b = 0.43, p <0.000). Self-efficacy also positively affected sustainable behavioral intentions during food preparation (b = 0.41, p <0.000), and sustainable behavioral intentions during food preparation positively affected sustainable preparation promotion (b = 0.80, p <0.000). Therefore, the results supported H1, H2, H3, H4, H5, and H6 (Figure 2).

4.3 Mediating Effect

The mediating effect of the perceived value in hotel restaurants was tested by performing Preacher and Hayes [69] bootstrapping techniques with 95% confidence intervals, using 10,000 bootstraps to study the direct, indirect, and overall effects of the proposed model. Ethical sustainability had a significant overall impact (standardized coefficient = 0.328, p < 0.05) on sustainable preparation promotion (Table 4). Ethical sustainability had a positive significant direct effect (standardized coefficient = 0.140, p < 0.05) and indirect effect (standardized coefficient = 0.329, p < 0.01) on self-efficacy. These results indicated that professional competence in food-waste prevention played a partial mediating role in the relationships between ethical sustainability and self-efficacy and supported H7 of this study. Similarly, professional competence in food-waste prevention had a significant direct effect (standardized coefficient = 0.425, p < 0.01) and indirect effect (standardized coefficient = 0.329, p < 0.01) on sustainable behavioral intentions during food preparation. These results indicated that self-efficacy had a partial mediating role in the relationships between professional competence in food-waste prevention and sustainable behavioral intentions during food preparation. Therefore, H8 was supported.

Table 2 Measurement model estimated parameters

Items

Std. factor loadings

Cronbach’s α

CR

AVE

E. Ethical sustainability

.832

.836

.630

E1. I will choose to use the food produced through a method that does not harm the animals (for example, enough space for raising).

.77

E2. I can consider environmentally friendly ways to prepare food and try to avoid food waste.

.79

E3. I will consider ways that will not disrupt the ecological balance to avoid waste in the food preparation process.

.82

C. Professional competence in food-waste prevention a

.926

.930

.655

C1 Hygiene knowledge

.80

C2 Cooking knowledge

.85

C3 Personal moral attitude

.72

C4 Attitude towards ingredients

.81

C5 Education and training attitude

.83

C6 Dish analysis skills

.81

C7 Planning and management skills

.84

S. Self-efficacy

.907

.910

.670

S1. I can set a goal to avoid food waste, and I think I can get the desired result.

.76

S2. I will be able to successfully overcome several challenges to avoid food waste.

.83

S3. I believe that I can effectively perform several different tasks to avoid food waste.

.84

S4 Compared with others, I can complete most tasks well to avoid food waste.

.82

S5. Even if I encounter difficulties in food-waste prevention, I can try my best to achieve the goal.

.84

I. Sustainable behavioral intentions during food preparation

.879

.843

.696

I1. I am willing to take food preparation activities for a healthy and sustainable diet.

.78

I2. I will encourage others to participate in healthy and sustainable food preparation together.

.86

I3. I will adjust and change my food preparation habits to promote sustainable positive effects of food preparation on health and the environment.

.86

B. Sustainable preparation promotion behavior

.850

.856

.600

B1. I implement green procurement (e.g., purchase local food, support green diet).

.65

B2. I implement and supervise others to reduce food waste in preparation.

.77

B3. I participate in the action of a healthy and sustainable diet.

.89

B4. I support the sustainable food/snacks activities organized by the government.

.77

a Twenty-nine items measured for professional competence load onto seven factors.

Table 3 Relevance among various dimensions and the correlations among all the constructs

Variable

Average mean

Standard deviation

E1

C

S

I

B

E. Ethical sustainability

4.10

.58

.8182

C. Professional competence in food-waste prevention

4.39

.55

.459

.789

S. Self-efficacy

4.12

.51

.712

.497

.806

I. Sustainable behavioral intentions during food preparation

4.23

.61

.680

.537

.665

.844

B. Sustainable preparation promotion behavior

4.13

.61

.680

.519

.721

.755

.777

**p < 0.01; N =351

1E: Ethical sustainability; C: Professional competence in food-waste prevention; S: Self-efficacy; I: Sustainable behavioral intentions during food preparation; B: Sustainable preparation promotion

2The square root of the AVE for discriminant validity is italicized along the diagonal.

Figure 3. Path analysis of each construct for overall structure model

(E: Ethical sustainability; C: Professional competence in food-waste prevention; S: Self-efficacy; I: Sustainable behavioral intentions during food preparation; B: Sustainable preparation promotion)

Table 4 The total, indirect, and direct effects of the hypothetical model

Bootstrapping

Estimate

Bias-corrected

percentile 95% CI

Lower

Upper

Two-tailed

significance

Indirect effect

Ethical sustainability→Professional competence in food-waste prevention →Self-efficacy

.329

.224

.432

.007**

Professional competence in food-waste prevention →Self-efficacy→ Sustainable behavioral intentions during food preparation

.266

.181

.397

.004**

Direct effect

Ethical sustainability→Professional competence in food-waste prevention

.510

.385

.603

.019*

Ethical sustainability→Self-efficacy

.140

.032

.277

.021*

Professional competence in food-waste prevention →Self-efficacy

.645

.483

.740

.009**

Professional competence in food-waste prevention → Sustainable behavioral intentions during food preparation

.425

.250

.617

.006**

Self-efficacy→ Sustainable behavioral intentions during food preparation

.413

.245

.577

.018*

Sustainable behavioral intentions during food preparation→ Sustainable preparation promotion

.798

.721

.859

.009**

Total effect

Ethical sustainability→Sustainable preparation promotion

.328

.234

.420

.018*

  1. Conclusion and Discussion

5.1 Conclusion

According to the data analysis results, the dimensions of ethical sustainability, professional competence in food-waste prevention, self-efficacy, sustainable preparation intention, and sustainable preparation promotion were all found to be significantly and positively correlated. This study first analyzed the CFA among the various dimensions, and the results showed that this model had a good fit. All indices passed the standards, indicating a compliant fit. This model could interpret the impact of each dimension. Ethical sustainability positively affected professional competence in food-waste prevention and self-efficacy, professional competence in food-waste prevention positively affected self-efficacy and sustainable behavioral intentions during food preparation. Similarly, self-efficacy positively affected sustainable behavioral intentions during food preparation, and sustainable behavioral intentions during food preparation significantly and positively affected sustainable preparation promotion. Therefore, H1, H2, H3, H4, H5, and H6 all had empirical support. Moreover, professional competence in food-waste prevention was the factor mediating ethical sustainability to sustainable behavioral intentions during food preparation, while self-efficacy was the factor mediating professional competence in food-waste prevention to sustainable behavioral intentions during food preparation. Both H7 and H8 were supported.

5.2 Discussion

Ethical sustainability positively affected professional competence in food-waste prevention and self-efficacy, while professional competence in food-waste prevention was the mediating factor between ethical sustainability and sustainable behavioral intentions during food preparation. Edwards and Mercer [70] strongly suggested that lifestyle choices, such as diet and career choices, had an ethical basis. Waston and Meah [32] explained that the primary driver for lower food waste was an “ethic of thrift”, which was the ethical motivation to act appropriately and “be thrifty”. The ethical rule for food waste behavior was "don't waste food". Therefore, the core ethical commitment in the study was to avoid wasting food, thus making consumption more reasonable [39]. Therefore, kitchen staff must have a personal imperative to prevent food waste in order to establish relevant behaviors and build professional competence in that arena.

Based on social cognitive theory in the context of food waste, females were more cognitively advanced than males. Females more closely monitored reducing food waste, so they were more motivated [71]. This contributed to their identification of more motivational factors that could have had a positive effect on their behavior. The environment affected the individuals and, in turn, their behavior. If actual catering producers were more aware of food waste, it could help enhance their professional competence in food-waste prevention.

The research on food waste has shown that preventing food waste required different culinary skills and knowledge, so leftovers food could be re-purposed [72, 73]. During storage, leftovers were often misplaced or left in the refrigerator for too long and spoiled [14, 15, 17]. Therefore, professional competence in food preparation is important.

 Professional competence in food-waste prevention positively affected self-efficacy and sustainable behavioral intentions during food preparation. Self-efficacy was the mediating factor between professional competence in food-waste prevention and sustainable behavioral intentions during food preparation. Lin [74] defined knowledge-sharing as a culture of social interaction, and the composition of this culture included knowledge exchange, experiential inheritance, and mutual learning among employees because the growth of knowledge could increase self-efficacy. Ding and Huang [75] suggested that mutual assistance, cooperation, and knowledge-sharing were critical for obtaining a competitive advantage. Bandura [20] suggested that individuals with high self-efficacy could have more enthusiasm, broader vision, broader thinking, and thus, more courage to accept challenges, and these behaviors were predominantly guided by the successful experiences of others. Self-efficacy and behavioral intentions could, indeed, predict food handling behaviors [30].

 Self-efficacy in processing food was important and correlated with general self-efficacy. Therefore, relevant educational policies should support consumers in developing self-efficacy related to sustainable food handling. Previous research pointed to the need for self-efficacy to enhance knowledge and skills, yet building self-efficacy required the inclusion of food handling and cooking classes in school curricula based on real-world experiences and the social marketing of public policy [45, 56, 58,  64].

Self-efficacy could be conceptualized with different levels of linkages, ranging from general self-efficacy (a person's ability to exercise self-control in different situations) to domain-specific self-efficacy (related to a person's ability to control behavior in specific settings (such as work and home)) to task-related self-efficacy (such as confidence in being able to correctly sort leftover waste) [76]. Domain- and task-related self-efficacy was positively correlated with general self-efficacy [77]. We explored the different hierarchies of self-efficacy in food waste in order to provide insight into the root causes of food waste. Therefore, we investigated the causes of household food waste by focusing on the effect of self-efficacy on consumer food waste [76]. Therefore, enhancing professional competence in food-waste prevention could improve self-efficacy and help promote more sustainable practices around food preparation and surplus food.  

5.3 Management Implications

This study shed light the factors influencing food waste prevention in the catering industry and could help the government and catering stakeholders to made specific recommendations for promoting food-waste prevention. The results showed that the ethical sustainability attitudes of kitchen staff affected professional competence in food-waste prevention and self-efficacy. Ethical sustainability among kitchen staff was related to their individual experiences and work/education environments. Therefore, the awareness and education provided by companies and governments concerning food-waste prevention could indirectly affect the concept of ethical sustainability among kitchen staff. Influencing the knowledge, attitudes, and skills of professional competence in food-waste prevention, and continuously promoting better strategies for surplus food, are important factors for governmental and corporate sustainability promotion. Self-efficacy indicates the individual believes that they can successfully perform a certain task or challenge, and they have the self-confidence to complete the task successfully [18, 23]. We need more consumers who are aware of and committed to food-waste prevention, and we need strong policymakers with the right strategies to promote food-waste prevention systematically. In addition, the cultivation of professional competence in food-waste prevention could affect self-efficacy and could have a significant impact on behavioral intentions through self-efficacy. For example, a company could regularly hold competitions for the creative preparation of surplus food in order to encourage self-reflection in catering practitioners [78]. Competitions or mutual internal audits of surplus food and food-waste prevention among catering practitioners could be helpful to understand limitations and share effective approaches, which could then further improve the reduction of food waste company-wide.

5.4 Practical Implications

It has been shown that the greater the professional competence of employees, the greater the increase in self-efficacy. Sustainability issues have become an important consideration in the catering industry in recent years. The responsibility and prevention of food waste are not only related to consumers but, rather, are closely related to the policies and factors that govern the entire supply chain. It is an encouraging sign that food-waste prevention is now a major socio-economic issue globally.

Implicit traits could be adjusted through a longer period of education and training, psychological counseling, or accumulated practical training [37]. According to our results, the level of performance depends on the level of self-efficacy and professional competence. More time must be spent on the psychological development of professional competence before effectiveness can be achieved. Related courses such as sustainability ethics, workplace ethics, and professional ethics are also indispensable components. In these courses, such topics can be discussed via case studies, peer experience sharing, and industry experience seminars for exploring individual perceptions that contribute to the improvement of self-efficacy and overall behavior motivation.

Kitchen staff are crucial for food preparation and management in the catering industry. The knowledge–attitude–behavior model [79] suggested that increased environmental knowledge could enhance environmental awareness and concern, leading to more environmentally conscious behaviors. This knowledge also generated positive thoughts, concerns, and practices [80]. In other words, the environment changed the person, and only personal changes could lead to changes in attitudes towards surplus food and more sustainable efforts when processing ingredients and preparing food. These could, then, aid the catering industry in food-waste prevention, sustainable ethics, the sustainable use of ingredients, sustainable cooking, and environmental protection.

5.5 Research Limitations and Future Studies

There were some limitations that should be considered. First, we primarily focused on those working in hotels and restaurants, as well as those working for group-meal providers, as a whole. To understand the differences in their environments, they should be considered separately. In the future, chain restaurants and central catering factories may be considered as the subjects, and different results may be obtained since other products in these catering sectors have been more standardized.

Due to the survey method used to collect data at a specific point in time, the results may be somewhat biased because the research instrument was a self-report survey [81]. Therefore, future studies should design more rigorous research procedures, preferably collecting data on predictor and outcome variables from different respondents. Finally, this study involved quantitative research, which could be supplemented by qualitative research in the future to understand issues associated with its implementation. We could have collected more factors affecting food waste in order to explore the actual situation of sustainable preparation behavior, such as the types of service and the ranks of kitchen staff, as well as to discuss the factors regulating sustainable preparation behavior.

References

  1. Amicarelli, V., Lagioia, G., and Bux, C. (2021). Global warming potential of food waste through the life cycle assessment: An analytical review. Environmental Impact Assessment Review, 91(11), 106677.
  2. Food and Agriculture Organization (FAO) (2022). Food Losses and Food Waste. Available online: http://www.fao.org/food-loss-and-food-waste/flw-data) (accessed on 10 Oct, 2022).
  3. Tarasuk, V., & Eakin, J. (2005). Food Assistance Through "Surplus" Food: Insights from an Ethnographic Study of Food Bank Work. Agriculture and Human Values, 177-186.
  4. WRAP (2022). Food Surplus and Waste in the UK Key Facts. Available online: https://wrap.org.uk/resources/report/food-surplus-and-waste-uk-key-facts (accessed on 20 Aug, 2022).
  5. Jungowska, J., Kulczy´ nski,B., Sidor, A., Gramza-MichaÅ‚owska, A. (2021). Assessment of Factors Affecting the Amount of Food Waste in Households Run by Polish Women Aware of Well-Being. Sustainability, 13, 976. https://doi.org/ 10.3390/su13020976
  6. Foley, J. A., Ramankutty, N., Brauman, K. A., Cassidy, E. S., Gerber, J. S., Johnston, M., & Zaks, D. P. M. (2011). Solutions for a cultivated planet. Nature, 478(7369), 337–342.
  7. Willett, W., Rockström, J., Loken, B., Springmann, M., Lang, T., Vermeulen, S., Garnett, T., Tilman, D.,Declerck, F., Wood, A., Jonell, M., Clark, M., Gordon, L., Fanzo, J., Hawkes, C., Zurayk, R., Rivera, J., Vries, W., Sibanda, L., & Murray, C. (2019). Food in the Anthropocene: the EAT–Lancet Commission on healthy diets from sustainable food systems. The Lancet. 393.(10170). 447-492.
  8. Stenmarck, Â., Jensen, C., Quested, T., Moates, G., Buksti, M., Cseh, B., & Scherhaufer, S. (2016). Estimates of European food waste levels. IVL Swedish Environmental Research Institute.
  9. Stirnimann, A. and Zizka, L. (2022). Waste not, want not: Managerial attitudes towards mitigating food waste in the Swiss-German restaurant industry. Journal of Foodservice Business Research, 25(3), 302–328
  10. McAdams, B., Massow, M., Gallant, M., and Hayhoe, M.A. (2019). A cross industry evaluation of food waste in restaurants. Journal of Foodservice Business Research, 22(5), 449–466.https://doi.org/10.1080/15378020.2019.1637220
  11. Ganglbauer, E., Fitzpatrick, G., Comber, R. (2013). Negotiating food waste: using a practice lens to inform design. ACM Trans. Computer Human Interaction, 20, 1-25.
  12. Graham-Rowe, E., Jessop, D.C., and Sparks, P. (2014). Identifying motivations and barriers to minimizing household food waste. Resources, Conservation and Recycling, 84, 15-23.
  13. Wang, Y.F. (2016). Improving Culinary Education by Examining the Green Culinary Behaviors of Hospitality College Students. Journal of Hospitality & Tourism Education, 28(1), 1-9, DOI: 10.1080/10963758.2015.1127167
  14. Farr-Wharton, G., Foth, M., Choi, J.H.J. (2014). Identifying factors that promote consumer behaviours causing expired domestic food waste. Journal of Consumer Behavior,13, 393-402.
  15. Waitt, G., Phillips, C. (2016). Food waste and domestic refrigeration: a visceral and material approach. Social & Cultural Geography, 17, 359-379.
  16. Secondi, L., Principato, L., and Laureti, T. (2015). Household food waste behaviour in EU- 27 countries: a multilevel analysis. Food Policy, 56, 25-40.
  17. Blichfeldt, B.S., Mikkelsen, M., and Gram, M.(2015). When it stops being food: The edibility, ideology, procrastination, objectification and internalization of household food waste. Food, Culture & Society, 18, 89-105.
  18. Bandura, A. (1982). Self-efficacy mechanism in human agency. American Psychologist, 37, 122-147.
  19. Wood, R., & Bandura, A. (1989). Impact of conceptions of ability on self-regulatory mechanisms and complex decision making. Journal of Personality and Social Psychology, 56,407-415.
  20. Bandura, A. (1977). Self-Efficacy: Toward a Unifying Theory of Behavioral Change. Psychological Review, 84(2), 191-215.
  21. Gist, M. E., & Mitchell, T. R. (1992). Self-efficacy: A theoretical analysis of its determinants and malleability. Academy of Management Review, 17, 183-211.
  22. Stajkovic, A. D., & Luthans, F. (1998). Self-efficacy and work-related performance: A metaanalysis. Psychological Bulletin, 124, 240-261.
  23. Bandura, A. (1986). Social foundations of thought and action. Englewood Cliffs, NJ: Prentice-Hall.
  24. Garrone, P., Melacini, M., & Perego, A. (2013). Feed the Hungry: The Potential of Surplus Food Recovery. Milan, Italy: Edizioni AngeloGuerini e Associati SpA.
  25. Parfitt, J., Barthel, M., & Macnaughton, S. (2010). Food waste within food supply chains: Quantification and potential for change to 2050. Philosophical Transactions of the Royal Society B, 365, 3065–3081.
  26. Göbel, C., Langen, N., Blumentha, A., Teitscheid, P., and Ritter, G. (2015). Cutting Food Waste through Cooperation along the food Supply Chain. Sustainability, 7, 1431-1438.
  27. Hawkes, C., & Webster, J. (2000). Too much and too little? debates on surplus food redistribution. London: Sustain.
  28. Bandura, A. (1977). Social learning theory. Englewood Cliff, NJ: Prentice Hall.
  29. Rimal, R.N. (2000). Closing the knowledge-behavior gap in health promotion: The mediating role of self-efficacy. Health Communication, 12(3), 219–237.
  30. Bearth, A., Cousin, M. -E., & Siegrist, M. (2014). Investigating novice cooks' behavior change: Avoiding cross-contamination. Food Control, 40, 26–31.
  31. Mullan, B., Allom, V., Fayn, K., and Johnston, I. (2014). Building habit strength: A pilot intervention designed to improve food-safety behavior. Food Research International, 66, 274-278.
  32. Watson, M., & Meah, A. (2012). Food, waste and safety: Negotiating conflicting social anxieties into the practices of domestic provisioning. The Sociological Review, 60(S2), 102–120.
  33. Goldstein, N. J., Cialdini, R. B., & Griskevicius, V. (2008). A room with a viewpoint: Using norms to motivate environmental conservation in hotels. Journal of Consumer Research, 35, 472-482. doi:10.1086/586910
  34. Glendinning, C. (1994). My name is Chellis and I’m in recovery from Western civilization. Gabriola Island: New Catalyst Books.
  35. Abram, D. (1996) The spell of the sensuous. New York: Vintage Books.
  36. MacMillan, T. (2009) What is wrong with waste? Food Ethics, 4 (3), p. 4.
  37. Stancu, V., Haugaard, P. and Lähteenmäki, L. (2016) Determinants of consumer food waste behaviour: two routes to food waste. Appetite, 96, 7-17.
  38. Gjerris, M., & Gaiani, S. (2013). Household food waste in Nordic countries:Estimations and ethical implications. Nordic Journal of Applied Ethics, 7 (1),6-23.
  39. Lehtokunnas, T., Mattila, M., Narvanen, E., and Mesiranta, N. (2020). Towards a circular economy in food consumption: Food waste reduction practices as ethical work. Journal of Consumer Culture, https://doi.org/10.1177/1469540520926252
  40. Lacasse, K. (2016). Don’t be satisfied, identify! Strengthening positive spillover by connecting pro-environmental behaviors to an “environmentalist” label. Journal of Environmental Psychology, 48, 149-158. doi:10.1016/j.jenvp.2016.09.006
  41. van der Werff, E., Steg, L., & Keizer, K. (2014). Follow the signal: When past proenvironmental actions signal who you are. Journal of Environmental Psychology, 40, 273-282. doi:10.1016/j.jenvp.2014.07.004
  42. Whitmarsh, L., & O’Neill, S. (2010). Green identity, green living? The role of pro-environmental self-identity in determining consistency across diverse proenvironmental behaviours. Journal of Environmental Psychology, 30, 305-314. doi:10.1016/j.jenvp.2010.01.003
  43. Cialdini, R. B., Kallgren, C. A., & Reno, R. R. (1991). A focus theory of normative conduct: A theoretical refinement and reevaluation of the role of norms in human behavior. Advances in Experimental Social Psychology, 24, 201-234. doi:10.1016/S0065-2601(08)60330-5
  44. Harrington, R.J., Mandabach, K.H., VanLeeuwen, D., and Thibodeaux, W. (2005). A multi-lens framework explaining structural differences across foodservice and culinary education. International Journal of Hospitality Management, 24, 198-218.
  45. Hegarty, J. A., & O’Mahony, G. B. (2001). Gastronomy: A phenomenon of cultural expressionism and an aesthetic for living. International Journal of Hospitality Management, 20, 3-13.
  46. Agut, S., Grau, R., and Peiro, J.M. (2003). Competency needs among managers from Spanish hotels and restaurants and their training demands. International Journal of Hospitality Management, 22, 281-295.
  47. Ko,W.H. and Lu, M.Y. (2021). Developing a professional competence scale for kitchen staff: Food value and availability for surplus food. International Journal of Hospitality Management, 95, 102926.
  48. Tekin, Ö.A.; Ilyasov, A. (2017). The Food Waste in Five-Star Hotels: A Study on Turkish Guests’ Attitudes. Journal of Tourism and Gastronomy Studies, 5, 13–31.
  49. Ajzen, I. (2002) Perceived behavioral control, self-efficacy, locus of control, and the theory of planned behavior. Journal of Applied Social Psychology, 32(4), 665-683.
  50. Ajzen, I. and Fishbein, M. (1980). Understanding Attitudes and Predicting Social Behavior, Prentice-Hall, Englewood Cliffs, NJ.
  51. Ajzen, I. (1991). The theory of planned behavior. Organizational Behavior and Human Decision Processes, 50(2), 179-211.
  52. Chan, E.S., Hon, A.H., Chan, W. and Okumus, F. (2014). What drives employees’ intentions to implement green practices in hotels? the role of knowledge, awareness, concern and ecological behavior. International Journal of Hospitality Management, 40, 20-28.
  53. Spencer, L. M., & Spencer, S. M. (1993). Competence at work- Models for Superior Performance. NY: John Wiley & Sons.
  54. Romani, S., Grappi, S., Bagozzi, R.P. and Barone, A.M. (2018) Domestic food practices. A study of food management behaviors and the role of food preparation planning in reducing waste. Appetite, 121, 215-227.
  55. Lavelle, F., McGowan, L., Spence, M., Caraher, M., Raats, M.M., Hollywood, L., McDowell, D., McCloat, A., Mooney, E. and Dean, M. (2016) Barriers and facilitators to cooking from 'scratch' using basic or raw ingredients. A qualitative interview study. Appetite, 107, 383-391.
  56. Thyberg, K.L. and Tonjes, D.J. (2016) Drivers of food waste and their implications for sustainable policy development. Resources, Conservation and Recycling, 106, 110-123.
  57. Hebrok, M. and Boks, C. (2017) Household food waste. Drivers and potential intervention points for design—an extensive review. Journal of Cleaner Production, 151, 380-392.
  58. Schanes, K., Dobernig, K. and Gözet, B. (2018) Food waste matters—a systematic review of household food waste practices and their policy implications. Journal of Cleaner Production, 182, 978-991.
  59. Andreasen, A.R. (2002) Marketing social marketing in the social change marketplace. Journal of Public Policy & Marketing, 21(1), 3-13.
  60. Lindeman, M. & Väänänen, M. (2000). Measurement of ethical food choice motives. Appetite, 34(1),55-59.
  61. Chen, G., Gully, S.M., Eden, D. (2001).Validation of a New General Self-Efficacy Scale. Organizational Research Methods, 4(1), 62-83.
  62. Fang, W.T., Ng, E., Wang, C.M., and Hsu, M.L. (2017). Normative Beliefs, Attitudes, and Social Norms: People Reduce Waste as an Index of Social Relationships When Spending Leisure Time. Sustainability, 9, 1696.
  63. Teng, C.C., Chih, C. and Wang, Y.C. (2020). Decisional Factors Driving Household Food Waste Prevention: Evidence from Taiwanese Families. Sustainability, 12, 6666; doi:10.3390/su12166666.
  64. Anderson, J.C. and Gerbing, D.W. (1988). Structural equation modeling in practice: a review and recommended two-step approach. Psychological Bulletin, 103(3), 411-423.
  65. Hair, J.F., Anderson, R.E., Tatham, R.L. and Black, W.C. (2010). Multivariate Data Analysis, 7th ed., Prentice Hall, Englewood Cliffs, NJ.
  66. Hu, L.T. and Bentler, P.M. (1999). Cutoff criteria for fit indexes in covariance structure analysis: conventional criteria versus new alternatives. Structural Equation Modeling: A Multidisciplinary Journal, 6(1), 1-55.
  67. Jöreskog, K.G. and Sörbom, D. (1989), LISREL 7: A Guide to the Program and Applications, SPSS, Chicago, IL.
  68. Fornell, C. and Larcker, D.F. (1981). Evaluating structural equation models with unobservable variables and measurement error. Journal of Marketing Research, 18(1), 39-50.
  69. Preacher, K.J. and Hayes, A.F. (2008). Asymptotic and resampling strategies for assessing and comparing indirect effects in multiple mediator models. Behavior Research Methods, 40, 879-891.
  70. Edwards, F. & Mercer, D. (2007). Gleaning from Gluttony: an Australian youth subculture confronts the ethics of waste. Australian Geographer, 38(3), 279-296.
  71. Goodman-Smith, F., Mirosa, R., and Mirosa, M. (2020). Understanding the E_ect of Dining and Motivational Factors on Out-Of-Home Consumer Food Waste. Sustainability, 12, 6507.
  72. Cappellini, B. (2009). The sacrifice of re-use: The travels of leftovers and family relations. Journal of Consumer Behavior, 8, 365-375.
  73. Southerton,D. & Yates,L. (2015). Exploring food waste through the lens of social practice theories: Some reflections on eating as compound practice. In: Ekstrom KM (ed.) Waste Management and Sustainable Consumption: Reflections on Consumer Waste. London; Chicago, IL: Routledge, 133-149.
  74. Lin, H. F. (2007). Effects of extrinsic and intrinsic motivation on employee knowledge sharing intentions. Journal of Information Science, 33(2), 135-149.
  75. Ding, X. H., & Huang, R. H. (2010). Effects of knowledge spillover on interorganizational resource sharing decision in collaborative knowledge creation. European Journal of Operational Research, 201, 949-959.
  76. Grether, T., Sowislo, J.F. and Wiese, B.S. (2018) Top-down or bottom-up? Prospective relations between general and domain-specific self-efficacy beliefs during a work-family transition. Personality and Individual Differences,121, 131-139.
  77. Luszczynska, A., Scholz, U. and Schwarzer, R. (2005) The general self-efficacy scale: multicultural validation studies. The Journal of Psychology, 139(5), 439-457.
  78. Saks, A. M. (1995). Longitudinal field investigation of the moderating and mediating effects of self-efficacy on the relationship between training and newcomer adjustment. Journal of Applied Psychology, 80(2), 211–225. https://doi.org/10.1037/0021-9010.80.2.211
  79. Kollmuss, A. and Agyeman, J. (2002). Mind the gap: why do people act environmentally and what are the barriers to pro-environmental behavior?. Environmental Education Research, 8(3), 239-260.
  80. Ruiz-Molina, M.E. and Gil-Saura, I. (2008). Perceived value, customer attitude and loyalty in retailing. Journal of Retail & Leisure Property, 7(4), 305-314.
  81. Podsakoff, P.M., MacKenzie, S.B., Lee, J.Y. and Podsakoff, N.P. (2003). Common method biases in behavioral research: a critical review of the literature and recommended remedies. Journal of Applied Psychology, 88(5), 879-903.

Reviewer 3 Report

Comments and Suggestions for Authors

Thank you for the opportunity to review the manuscript entitled “Sustainable Preparation Behavior for Kitchen Staff from the Viewpoint of Avoiding Food Waste”, submitted for possible publication to Foods. The purpose of the research is to explore the model of ethical sustainability, surplus food professional competence, self-efficacy, sustainable food preparation behavioral intention and sustainable food promotion behavior in the field of minimizing food waste. 

Abstract. The abstract can be improved. The authors should first introduce the context of the research. Secondly, the authors are invited to better clarify the purpose of the research and the methods adopted to carry out the analysis. Where data are collected? Which is the study area? Last, soon after the description of the main outcomes of the research (also under the quantitative perspective), could please the authors define the audience of the research? To whom is the research addressed?

Keywords. Is seems that some keywords are missing, or not? Three keywords could be increased.

As a general comment, the references should be written are required in the Instruction for Authors (MDPI). 

Introduction. The section “Introduction” must be revised. I appreciate that the authors have included some interesting quantitative information, but in the first paragraph the authors are asked to include more references on the topic. Further, the authors should define also the quantitative/qualitative environmental consequences of food waste, in a suitable paragraph (as well as additional references). So that, the authors provide a quantitative information in terms of weight, as well as an economic (already included) and environmental one. Please, consider the subsequent article for the environmental consequences of food waste: 

https://doi.org/10.1016/j.eiar.2021.106677

Soon after the sentence “The key problem of food waste lies in the catering supply chain”, the authors are invited to better clarify the significance of food waste in the hospitality industry (hotels, restaurants, etc.) and to introduce the topic of food waste in kitchens. Please, consider the subsequent articles, to better clarify such an aspect and justify the importance of the research (specifically) in kitchens: 

https://doi.org/10.1080/15378020.2019.1637220

https://doi.org/10.1080/15378020.2021.1942749

The section “Introduction” should be concluded with the information of the audience of the research. To whom is the research addressed? And what about the originality of the research?

Literature review. In the section “Literature review”, which I would better entitle “Theoretical background and hypotheses development”, I invite the authors to describe the supply chain (food supplying, food preparation, food serving, food consumption, food disposal) in hospitality kitchens, also by including a Figure. In such a Figure, the authors can better identify where food surplus, leftovers, or food waste are generated according to the specifying supply chain stage. 

Here, the authors introduce the “social cognitive theory” for the first time. If this theory represents the basis for the research, I suggest the authors including some insights also in the “Introduction”, so that readers can understand why there is a literature review on such a topic. 

I appreciate the development of the hypotheses at the basis for the research. For this reason, the title of the heading should better be “Theoretical background and hypotheses development”. 

Methodology. The section “Methodology” is rather clear, but additional information are required. First, what about the questionnaire development, the questionnaire structure, the number of questions, etc? Could the authors please include a Table in subsection “3.2. Measures”, which illustrates (and synthesized, at the same time) the structure of the questionnaire and all the variables asked to the sample? 

When the questionnaire was sent to the sample? In which period? It is important to appraise the historical value of data. 

In the subsection “Data analysis”, is it possible to refer to similar (and successful studies), which have applied similar models?

Results. In the description of the sociodemographic characteristics of the sample, is it possible to discuss on the representativeness of the sample? For instance, age or gender are representative of the Taiwan’s catering industry? If there are any references, please try to enlarge the discussion of such a brief section. 

In Table 3, could you please include in the notes the acronyms, as outlined in Figure 3?

Discussion. In the “Managerial implications” and in the “Practical implications”, is it possible to make reference to previous studies, as to compare and identify similarities/differences between the current research and previous ones?

Author Response

To Editor-in-Chief and reviewers:

Thank you for your comments and suggestions. I appreciate the time and effort that you and the reviewers have dedicated to providing your valuable feedback on my manuscript. I had major revised in this paper as final attached. Please see our comments below.

The following are Reviewer comments:

Overall

  1. The manuscript needs English editing.
  2. The references are not according to MDPI guidelines.

Ans: I had made major revise in all sections.

I had revised paper due to English language editing by MDPI. Please see the certification as below, and I also to check MDPI guideline to check all paper. Please see the revised paper. And, I also change title as “Sustainable Preparation Behavior for Kitchen Staff in order to Limit Food Waste”.

The following are our responses to the Reviewers’ 3 comments:

  1. The abstract can be improved. The authors should first introduce the context of the research. Secondly, the authors are invited to better clarify the purpose of the research and the methods adopted to carry out the analysis.
  2. Is seems that some keywords are missing, or not? Three keywords could be increased.

Ans:  I have added a statement in abstract, which should help to clear up, and I added some keywords. Please see below.

The concepts of culinary sustainability and avoiding the waste of surplus food have become important sustainability trends today. How to integrate the handling of surplus food in the catering industry is a topic of concern in the industry. Kitchen staff are the vital soul of any restaurant, and we intend to discuss how kitchen staff actually behave and to explore factors that influence their behaviors in order to develop an implementation model for food-waste prevention. Therefore, this study explored a model of ethical sustainability, professional competence, self-efficacy, sustainable food preparation objectives, and sustainable food promotion and behavior focused on limiting food waste.  This study used a questionnaire and surveyed employees who had been employed for more than 6 months in Taiwan. From May to August 2022, 500 questionnaires were distributed; 415 valid questionnaires were retrieved, yielding a 90.2% recovery rate. According to the structural equation modeling analysis between the dimensions, ethical sustainability should have had a positive influence on professional competence in food-waste prevention and self-efficacy. Professional competence in food-waste prevention affected self-efficacy and behavioral intentions during food preparation; self-efficacy also significantly affected behavioral intentions towards sustainable food preparation. Similarly, behavioral intention had a positive influence on promoting sustainable behaviors. This model was a good fit for the acquired data. Professional competence in food-waste prevention was found to be the mediating factor between ethical sustainability and the behavioral intentions towards sustainable food preparation, and self-efficacy was the mediating factor between the professional competence in food-waste prevention and the behavioral intentions towards sustainable food preparation.

Keywords: surplus food, sustainable preparation behavior, food waste, kitchen staff, ethical sustainability, self- efficacy

---------------------------------------------------------------------------

  1. The section “Introduction” must be revised. I appreciate that the authors have included some interesting quantitative information, but in the first paragraph the authors are asked to include more references on the topic.

   Here, the authors introduce the “social cognitive theory” for the first time. If this theory represents the basis for the research, I suggest the authors including some insights also in the “Introduction”, so that readers can understand why there is a literature review on such a topic. 

Ans: I had added some references and more detail statements as reviewer’s commend and major revised. Please see the revised paper.

  1. Introduction

Food is an important daily necessity. With advancements in the global economy, equipment, and food production efficiency, an abundant supply of food can provide for more people. From production to consumption, food is often discarded by consumers even though it is safe to eat because they have bought too much or have insufficient knowledge about the shelf life of food. It is estimated that one-third (1.3 billion tons) of the food produced globally for consumption is wasted every year [1,2]. The problem of food waste has been included in the World Health Organizations’ sustainable development goals (SDG 17) in order to reduce global human food waste by half and to reduce food loss by 2030. Food service and family experiences have been shown to be an appropriate way to address national and international sustainable goals [1].

The problem of surplus food management has also attracted significant attention from academics and other concerned practitioners because the normal dietary sources of a population are often interrupted due to limited resources, and experts believe that it is possible to reduce food insecurity via surplus food management. In addition to the commercial economic losses due to food production and waste, the social and environmental losses have also been significant, especially when foods that were still edible were not able to be distributed and sold, for various reasons, and were eventually discarded [3].

The Food and Agriculture Organization (FAO) of the United Nations has estimated that the annual global food waste across the entire food chain was USD 2.6 trillion. The Waste and Resources Action Plan estimated that food waste in the UK totaled around 10 million tons per year, 70% of which was avoidable [4]. Food waste is an urgent and serious issue in Western countries. The increase in the amount of food waste directly affects environmental changes, including greenhouse gas emissions and packaging pollution. In Poland, 9.2 million tons of grain have been lost every year, 53% of which was produced by consumers. In order to reduce consumer food waste, it was necessary to understand the behavioral factors [5]. The key problem of food waste has been primarily due to the supply chain, from production, preparation, and processing to transportation and consumption, all of which require considerable costs and resources. Therefore, practitioners and consumers should pay close attention to this issue from both the supply and the demand side. Reducing food waste is an important factor for sustainable development. The predicted population growth will require an adequate food system that includes the sustainable development of natural ecological resources [6,7]. Food waste occurs at all stages of the food supply chain, and the further down the supply chain, the greater the negative environmental, economic, and social impacts of food waste [8].

In restaurants, government intervention and enhanced waste disposal collection are necessary. Research has shown that changing consumer behavior, combined with corporate social responsibility, mitigated food waste [9]. The results of the study showed that many restaurants have prioritized reducing food waste. However, most of the final factors affecting food waste in restaurants were quality and specification issues during produ711ction. Generally, food waste was the worst in casual restaurants, and large meal portions resulted in more food waste [10].

The chefs are the souls of a restaurant. In addition to making good dishes, a good chef can also improve the product value while maintaining their social responsibility. Food oversupply has also been related to consumers' perceived concept of preservation. Consumers may purchase more than they need in order to save time [11, 12]. Among these food behaviors, discarding reusable ingredients as kitchen waste was the most severe (28.8% of ingredients were considered kitchen waste and discarded) [13]. Storing and sorting food in an orderly manner (for example, stacking old and new food, or sorting by frequency of use), and reordering regularly could reduce food waste [14, 15]. According to Secondi, Principato, and Laureti [16], one of the best courses of action to avoid wasting food was to estimate portion sizes correctly. Misplaced, forgotten, or expired leftovers were an issue when storing food [17].

Social cognitive theory (SCT) explains human behavior based on the dynamic interactions of individual experiences, the actions of others, and environment influences [18]. Self-efficacy is the belief in one's ability to change motivations, cognitive resources, and courses of action in order to meet the needs of a particular situation [19-22]. Research has found that self-efficacy predicted several job-related outcomes, such as work attitudes, training proficiency, and job performance. Bandura [23] pointed out that self-efficacy was the result of interactions with the external environment, personal professional competency, and learning effectiveness, which could then motivate behavior. Kitchen staff play a key role in food production from the kitchen to the table, and a sustainability-focused food revolution should be initiated for reducing carbon emissions and mitigating food waste to preserve the planet’s ecosystem.

Therefore, the attitudes of kitchen staff to avoid food waste could affect how they handle food, and sustainable cooking could provide substantial results in the future for environmental and food sustainability. This study explored the perspectives of kitchen staff towards limiting food waste by evaluating the behavioral patterns such as ethical sustainability, professional competence in food-waste prevention, self-efficacy, behavioral intentions towards sustainable food preparation, and sustainable food promotion from a practical perspective. This research could help promote sustainable cooking and food preparation, from the individual to the restaurant, and then to the catering industry. By encouraging consistent sustainability in the catering industry, we can more effectively reduce food waste.

-------------------------------------------------------------------------------------

  1. Literature review. In the section “Literature review”, which I would better entitle “Theoretical background and hypotheses development”, I invite the authors to describe the supply chain (food supplying, food preparation, food serving, food consumption, food disposal) in hospitality kitchens, also by including a Figure.

Ans: I had change title and added figure to state supply chain and food waste. Please see as below.

  1. Theoretical background and hypotheses development

2.1 Surplus food and food waste

The conceptual model of availability, surplus, recoverability, and waste (ASRW) provided a clear definition of surplus food and distinguished it from food waste by introducing the concepts of availability, surplus, recoverability, and waste. According to this model, available food was defined as "all food in the food supply chain", which consisted of three parts: "human consumption", "surplus food", and "food scraps”. There has been no universally accepted definition of food waste and surplus food because there have been different interpretations from different perspectives [24].

According to the FAO, food waste consisted of three distinct problems. Food loss (FL) was the reduction in weight or nutritional value of food originally intended for human consumption. These losses were primarily due to processes in the supply chain, such as poor management, errors, and irregular processes (such as improper farming and harvesting), resulting in reduced edible weight. Food waste (FW) was the discarding of edible food, whether expired or spoiled, due to reasons such as oversupply, lack of a sales strategy, or excessive or unplanned purchases by individual consumers. It was also the waste of edible food due to improper distribution, storage, transportation, and preparation, both in the home and in restaurants [25].

Moreover, food waste occurs at all stages of the food chain: production, distribution, retail, and consumption processes [26]. Types of waste at all stages of the food chain are shown in Figure 1. Wasted food is any food that is lost due to spoilage or being discarded. "Waste food" as opposed to "food waste" refers to food that has not been used for its intended purpose but could be further used as a valuable resource [2].

Figure 1. Food waste styles in all stages of the food chain

According to scholars [27], surplus food could be divided into four categories: (1) Unprofitable crops refer to the market phenomenon after the harvest of agricultural products, and the surplus of grain sales is generated due to a stable sale price. When production is much higher than demand, it can lead to an oversupply in the market. (2) Non-perishable processed foods are foods with a long shelf life (such as dry goods, canned foods, etc.), damaged packaging, incorrect packaging, promotions near the expiration date, etc. (3) Perishable fresh foods are fresh fruits and vegetables, bread, dairy products, refrigerated ready-to-eat foods, etc.; products that are not sold because they are close to their expiration date but are still safe to eat; and foods that agricultural producers and importers may not be able to sell because the fruit or vegetables are irregular in shape and size. (4) Perishable foods include ready-to-eat foods such as sandwiches, cakes, and pastries [27].

2.2 Social Cognitive Theory

Social cognitive theory (SCT) was first proposed by Bandura, who combined the concepts of behaviorism and social learning [18]. Behavior is generated by the interaction between the individual and the environment, rather than determined by any single aspect. These factors include environmental impacts (the overall social environment, organizational policies, and culture); individual cognition and personal factors (personal motivation, attitude); and behavior (intention and willingness), all of which interact. Figure 2 shows the dynamic formation of the person, the environment, and behavior in SCT, where personal behavior is affected by the environment and influenced by personal subjective cognitions. The psychological process of self-introspection is achieved through the inner driving force of the individual, and the person feels and reacts to their external environment due to their inner thoughts, rather than being stimulated by the environment alone. Therefore, this research hypothesized that environmental factors could influence personal factors, and personal factors could influence behavioral factors.

                      Figure 2. Interactive model of social cognitive theory

Bandura [28] suggested that persons with high self-efficacy could have more enthusiasm, broader vision, broader thinking, and, therefore, courage to accept challenges. To avoid repeating the failures of others, most individuals would use the success of others as a guide. Rimal [29] also pointed out that SCT was an important basis for predicting food handling processes. Intention and self-efficacy could predict food-handling behavior [30]. Bandura [23] also suggested that personal past performance affected self-efficacy, and higher self-efficacy could also affect future performance. Behavioral improvements could be enhanced by offering continuous reminders during food handling [31].

Bandura [23] considered SCT the most suitable for explaining dynamic human behavior and regarded cognition and self-regulation as the primary framework of causality. The development of this model could cultivate the confidence of individuals to use certain abilities. However, it could also be affected by many internal factors, such as the level of self-efficacy, which would then result in different behaviors. Self-efficacy refers to a person's belief in their own capabilities and their degree of confidence in being able to achieve a specific outcome.

2.3 Ethical Sustainability Affects Personal Behavior

Waston and Meah [32] indicated that even if consumers had an environmental motivation to avoid food waste, the real motivation to reduce food waste was the "ethic of thrift". People avoided wasting food because it caused feelings of guilt that were then combined with environmental concerns [17]. Food represented the physical and symbolic connection between humans and nature [33]. A strong negative attitude towards food waste could be explained by our respect for nature, and this could hopefully be extended to the community [34].

Therefore, food waste could be perceived as a waste of precious resources, which would be unfair from an anthropocentric perspective [35]. Food waste has been considered immoral, so people need to be called upon to change their behaviors and attitudes concerning food waste [36, 37]. Therefore, food is a valuable resource, as it is not always readily available. A change in attitude could benefit both humans and nature by reducing the negative aspects of food production and improving the balance between humans and nature [38]. The “moral” rule for food waste behavior could be "don't waste food". With this in mind, our core ethical commitment was to avoid wasting food through more sensible consumption. Doing so could protect and preserve the world in which we live [39].

As indicated by the self-perception theory, previous behaviors affected future actions. The theory suggested that if individuals were encouraged to behave in environmentally conscious ways, they were more likely to make environmentally conscious decisions [40, 41]. This theory reinforced environmental self-identity, or an individual's perception of themselves as a person who is environmentally conscious [42]. In addition, many previous studies found that social norms and the behaviors of social groups had a positive influence on behavior [33,43].

Therefore, the following hypotheses were developed:

H1: Attitudes towards ethical sustainability affected professional competence in food-waste prevention.

H2: Attitudes towards ethical sustainability affected self-efficacy.

2.4 Relationships among professional competence, self-efficacy, and food-waste prevention

Food knowledge and skills, communication skills, business knowledge, and food concepts were considered the main professional abilities of kitchen staff [44]. To conform to social trends and achieve the following objectives, Hegarty and O'Mahony [45] developed the four major fields of culinary courses, including information; culinary arts; food and life sciences; and business. The basic skills needed for kitchen staff were food science, product preparation, finance, personal management, marketing, language, and computer skills [46]. Professional competence for food-waste prevention included two aspects: hygiene knowledge and culinary knowledge. Professional attitudes towards food-waste prevention included three dimensions, namely personal moral attitudes, attitudes towards food ingredients, and attitudes towards education and training. Skill competence comprised two dimensions: dish analysis and meal planning management [47]. In the restaurant and hospitality industry, both food preparers and consumers are responsible for food waste. Food waste has also been identified as a problem in schools, kindergartens, canteens, gas stations, bars, restaurants, canteens, and hotels [48].

The concept of self-efficacy first appeared in an article by Bandura [20]. It refers to an individual’s belief whether one can successfully complete a task. The emphasis was on ability, and it could promote individual actions in certain areas and affect future activities. This aspect was related to personal values. Bandura [18] also defined self-efficacy as “in order to deal with an impending situation, an individual makes a subjective judgment on whether he or she can successfully perform the required behavior. This judgment will decide how much effort he or she will put in when he or she personally faces difficulties, and how long it will last”. That is, self-efficacy could affect whether a person would persist in an action after setting goals. This was also related to whether a person had the motivation to overcome obstacles in the learning process and the learning strategies adopted when solving problems.

The theoretical basis of perceived behavioral control was originally derived from the self-efficacy theory, so perceived behavioral control and self-efficacy were conceptually similar [49]. Many studies have employed the theory of rational action (TRA) [50] or its extended theory of planned behavior (TPB) model [51] to understand kitchen workers' behavioral intentions towards food-waste prevention. The TRA theory suggested that previous attitude (i.e., approval or disapproval of a certain behavior) and existing subjective norms (i.e., whether the implementation of a certain behavior was approved by others) were two important influences on a person's behavioral intentions to implement a certain behavior [52]. Based on these previous studies, the current study aimed to understand whether ethical sustainability and professional competence in food-waste prevention were important indicators of behavioral intentions among kitchen staff. According to TRA and TPB, a person’s attitude indicated the favorable or unfavorable evaluation of a certain behavior, so it could predict behavioral intentions [51]. Therefore, the following hypotheses were developed:

H3: Professional competence could positively influence self-efficacy in food-waste prevention.

H4: Professional competence in food-waste prevention could positively influence the sustainable behavioral intentions during food preparation.

H5: Self-efficacy could positively influence sustainable behavioral intentions during food preparation.

H6: Sustainable behavioral intentions during food preparation could positively influence the promotion of sustainable preparation.

2.5 Mediating role of professional competence in food-waste prevention and self-efficacy

Based on the iceberg model proposed by Spencer and Spencer [53], professional competence could be classified as five basic factors: motivation, traits, self-concept, knowledge, and skills. Knowledge and skills were explicit traits: Both were easily affected by the external environment, both were trainable, and both could be enhanced through education and learning. Implicit traits, such as self-concept, traits, and motivation, could not be easily explored and developed and, therefore, would need to be inherent in the individual. The capacity for efficient food handling was the most important factor in preventing food waste [54]. Self-efficacy was important during food preparation and was also linked to preventing food waste [55].

There is a considerable relationship between the self-efficacy of food handling competence and general self-efficacy. Therefore, if you can enhance your personal self-efficacy, you can also enhance your competence in food handling. It was necessary to improve professional knowledge and skills, as it could improve self-efficacy and food handling behavior [56, 57, 58]. In addition, through food handling and cooking classes, self-efficacy could be enhanced, and behavioral ability could be improved [59]. Therefore, the following hypotheses were developed:

H7: Professional competence in food-waste prevention mediated the attitudes toward ethical sustainability and the sustainable behavioral intentions during food preparation.

H8: Self-efficacy mediated the professional competence in food-waste prevention and the sustainable behavioral intentions during food preparation.

--------------------------------------------------------------------------------------------

  1. The section “Methodology” is rather clear, but additional information are required. 

 Ans: I had added more information in methodology as below.

  1. Methodology

3.1 Data Collection

A pilot study was conducted to ensure the reliability of the measurements. The first part of the questionnaire included questions related to all the variables of this study. The second part obtained the demographics of the participants, including gender, age, education, job title, work experience, and restaurant type.

To ensure the face validity of the questionnaire, we invited three professors to evaluate the validity of the predicted measurements. One hundred pilot-study questionnaires were returned with a valid return rate of 100%. The research process included item and factor analyses to determine the final questionnaire. All items in the questionnaire had good reliability (Cronbach's a > 0.70). In addition, according to the feedback of the participants, some items were revised.

This study included employees in Taiwan’s catering industry (including hotels, restaurants, and group-meal companies) as the research subjects. The convenience sampling and purposive sampling methods were adopted, and assistance was sought, through telephone interviews and e-mails, from individual practitioners in the catering industry who met the sampling criteria. From May to August 2022, questionnaires were sent and returned by mail. Out of 500 questionnaires, 415 valid questionnaires were returned, which yielded a 90.2% valid response rate.

3.2 Measures

  A questionnaire survey method was adopted for this study. All scales used to measure the constructs of this study were adopted from previous studies. The questionnaires used were divided into 6 parts: part 1, ethical sustainability, including 3 items; part 2, professional competence in food-waste prevention scale, including 29 items for 7 dimensions; part 3, self-efficacy, including 5 items; part 4, sustainable food behavioral intentions during food preparation, including 3 items; part 5, sustainable food promotion, including 4 items; and part 6, personal demographic.

For items measuring ethical sustainability, a three-item scale was adapted from the work of Lindeman and Väänänen [60]. This scale was based on a 5-point Likert scale, ranging from 1 (“strongly disagree”) to 5 (“strongly agree”). The Cronbach's α value of this scale was 0.832. The scale of professional competence in food-waste prevention was according to the professional competence in food-waste prevention scale of Ko and Lu [47], with 29 questions of 7 dimensions, and a 5-point Likert scale ranging from 1 (“strongly disagree”) to 5 (“strongly agree”). The average mean of the questions in each dimension was taken as the average mean for analysis, and the total Cronbach's α value of the scale was 0.926. To measure self-efficacy, 5 items from Chen, Gully, and Eden [61] were adapted and rated by a 5-point Likert scale, ranging from 1 (“strongly disagree”) to 5 (“strongly agree”). Finally, sustainable behavioral intentions during food preparation were measured using 10 items from Fang, Wang, and Hsu [62], and there were 3 questions. The Cronbach's α value of this scale was 0.907. The sustainable preparation promotion was adapted from the 4-item scale of Teng, Chih, and Wang [63]. The Cronbach's α value of this scale was 0.850. These items were rated by a 5-point Likert scale, ranging from 1 (“strongly disagree”) to 5 (“strongly agree”).

3.3 Data Analysis

The model comprised two components, a measurement model and a causal structural model. Before data analysis, we performed structural equation modeling (SEM) using the AMOS 21.0 application software. Whether the observed variables reflected the hypothesized latent variables and whether the measurement model adequately fit the data was performed with a confirmatory factor analysis (CFA). Path analysis was used in the study to evaluate the fit of the structural model and the hypothesized relationships among the latent variables [64]. As suggested by Anderson and Gerbing [64], we started with a CFA to determine whether all indicator variables adequately reflected their underlying structure and that the measurement model was acceptable. The fit index, comparative fit index (CFI), and the root-mean-square error of approximation (RMSEA) were used to estimate the overall model fit.

-----------------------------------------------------------------------------------------

  1. In the description of the sociodemographic characteristics of the sample, is it possible to discuss on the representativeness of the sample? In Table 3, could you please include in the notes the acronyms, as outlined in Figure 3?

Ans: I added some describe in Table 1 and 3.

  1. Results

The subjects were mostly male in an age range of 21-40 years (70.1%). The education level was mostly high school and university. Non-management positions accounted for 71.8%, working experience of 1-4 years accounted for 28.2%, and 4-12 years accounted for 35.6%. The participants worked in hotel restaurants, independent restaurants, and restaurant chains (including central kitchens) (Table 1).

Table 1  Sociodemographic details of subjects

Background variables

Category

Number of persons

Percentage (%)

Gender

Male

217

61.8

Female

134

38.2

Age

20 years old or younger

8

2.3

21-30 years old

139

39.6

31-40 years old

107

30.5

41-50 years old

65

18.5

51-60 years old

27

7.7

61 years old or older

5

1.4

Educational level

Junior high school or below

14

4

High school ∕ vocational high school

130

37

Junior college

67

19.1

University

130

37

Graduate school

10

2.8

Job position

Management

99

28.2

Non-management

252

71.8

Working years

1 year or less

24

6.8

More than 1 year to 4 years (inclusive)

99

28.2

More than 4 year to 8 years (inclusive)

64

18.2

More than 8 year to 12 years (inclusive)

61

17.4

More than 12 year to 16 years (inclusive)

34

9.7

More than 16 year to 20 years (inclusive)

29

8.3

More than 20 years 

40

11.4

Employment

Restaurant in Hotel

120

34.2

Individual Restaurant

136

38.7

Group meal/central kitchen

82

23.4

Others

13

3.7

4.1 Measurement Model

The model fit indices then indicated an acceptable overall measurement model: x 2/df = 2.77; GFI = 0.87; AGFI = 0.84; RMSEA = 0.071; SRMR = 0.043; CFI = 0.98; and NFI = 0.97 [65]. Table 2 shows that all items had statistically significant (p < 0.01) [66] factor loadings, ranging from 0.65 to 0.89. The composite reliability values for all the constructs ranged from 0.88 to 0.95, indicating that all constructs had good internal reliability [67]. Furthermore, the average extracted variance (AVE) for each construct exceeded 0.50, which established the convergent validity of the measurement model [68]. Table 3 shows that the facets between all constructs were significant (p < 0.01), ranging from 0.459 to 0.755. In addition, the square root of AVE for each construct was also higher than the cross-correlation among all variables, so this model had discriminant validity [68].

4.2 Structural Model

The structural models were x 2/df = 3.10; GFI = 0.86; AGFI = 0.83; CFI = 0.98; NFI = 0.97; RMSEA = 0.077; and SRMR = 0.067, which indicated the good fit of the structural model [65]. The path analysis showed that environmental sustainability positively affected professional competence in food-waste prevention (b = 0.51, p < 0.000) and self-efficacy (b = 0.14, p < 0.000); professional competence in food-waste prevention positively affected self-efficacy (b = 0.65, p < 0.000) and sustainable behavioral intentions during food preparation  (b = 0.43, p <0.000). Self-efficacy also positively affected sustainable behavioral intentions during food preparation (b = 0.41, p <0.000), and sustainable behavioral intentions during food preparation positively affected sustainable preparation promotion (b = 0.80, p <0.000). Therefore, the results supported H1, H2, H3, H4, H5, and H6 (Figure 2).

4.3 Mediating Effect

The mediating effect of the perceived value in hotel restaurants was tested by performing Preacher and Hayes [69] bootstrapping techniques with 95% confidence intervals, using 10,000 bootstraps to study the direct, indirect, and overall effects of the proposed model. Ethical sustainability had a significant overall impact (standardized coefficient = 0.328, p < 0.05) on sustainable preparation promotion (Table 4). Ethical sustainability had a positive significant direct effect (standardized coefficient = 0.140, p < 0.05) and indirect effect (standardized coefficient = 0.329, p < 0.01) on self-efficacy. These results indicated that professional competence in food-waste prevention played a partial mediating role in the relationships between ethical sustainability and self-efficacy and supported H7 of this study. Similarly, professional competence in food-waste prevention had a significant direct effect (standardized coefficient = 0.425, p < 0.01) and indirect effect (standardized coefficient = 0.329, p < 0.01) on sustainable behavioral intentions during food preparation. These results indicated that self-efficacy had a partial mediating role in the relationships between professional competence in food-waste prevention and sustainable behavioral intentions during food preparation. Therefore, H8 was supported.

Table 2 Measurement model estimated parameters

Items

Std. factor loadings

Cronbach’s α

CR

AVE

E. Ethical sustainability

.832

.836

.630

E1. I will choose to use the food produced through a method that does not harm the animals (for example, enough space for raising).

.77

E2. I can consider environmentally friendly ways to prepare food and try to avoid food waste.

.79

E3. I will consider ways that will not disrupt the ecological balance to avoid waste in the food preparation process.

.82

C. Professional competence in food-waste prevention a

.926

.930

.655

C1 Hygiene knowledge

.80

C2 Cooking knowledge

.85

C3 Personal moral attitude

.72

C4 Attitude towards ingredients

.81

C5 Education and training attitude

.83

C6 Dish analysis skills

.81

C7 Planning and management skills

.84

S. Self-efficacy

.907

.910

.670

S1. I can set a goal to avoid food waste, and I think I can get the desired result.

.76

S2. I will be able to successfully overcome several challenges to avoid food waste.

.83

S3. I believe that I can effectively perform several different tasks to avoid food waste.

.84

S4 Compared with others, I can complete most tasks well to avoid food waste.

.82

S5. Even if I encounter difficulties in food-waste prevention, I can try my best to achieve the goal.

.84

I. Sustainable behavioral intentions during food preparation

.879

.843

.696

I1. I am willing to take food preparation activities for a healthy and sustainable diet.

.78

I2. I will encourage others to participate in healthy and sustainable food preparation together.

.86

I3. I will adjust and change my food preparation habits to promote sustainable positive effects of food preparation on health and the environment.

.86

B. Sustainable preparation promotion behavior

.850

.856

.600

B1. I implement green procurement (e.g., purchase local food, support green diet).

.65

B2. I implement and supervise others to reduce food waste in preparation.

.77

B3. I participate in the action of a healthy and sustainable diet.

.89

B4. I support the sustainable food/snacks activities organized by the government.

.77

a Twenty-nine items measured for professional competence load onto seven factors.

Table 3 Relevance among various dimensions and the correlations among all the constructs

Variable

Average mean

Standard deviation

E1

C

S

I

B

E. Ethical sustainability

4.10

.58

.8182

C. Professional competence in food-waste prevention

4.39

.55

.459

.789

S. Self-efficacy

4.12

.51

.712

.497

.806

I. Sustainable behavioral intentions during food preparation

4.23

.61

.680

.537

.665

.844

B. Sustainable preparation promotion behavior

4.13

.61

.680

.519

.721

.755

.777

**p < 0.01; N =351

1E: Ethical sustainability; C: Professional competence in food-waste prevention; S: Self-efficacy; I: Sustainable behavioral intentions during food preparation; B: Sustainable preparation promotion

2The square root of the AVE for discriminant validity is italicized along the diagonal.

-------------------------------------------------------------------------------------------------------------

  1. In the “Managerial implications” and in the “Practical implications”, is it possible to make reference to previous studies, as to compare and identify similarities/differences between the current research and previous ones?

Ans:  I added some references to text, please see as below. 

5.3 Management Implications

This study shed light the factors influencing food waste prevention in the catering industry and could help the government and catering stakeholders to made specific recommendations for promoting food-waste prevention. The results showed that the ethical sustainability attitudes of kitchen staff affected professional competence in food-waste prevention and self-efficacy. Ethical sustainability among kitchen staff was related to their individual experiences and work/education environments. Therefore, the awareness and education provided by companies and governments concerning food-waste prevention could indirectly affect the concept of ethical sustainability among kitchen staff. Influencing the knowledge, attitudes, and skills of professional competence in food-waste prevention, and continuously promoting better strategies for surplus food, are important factors for governmental and corporate sustainability promotion. Self-efficacy indicates the individual believes that they can successfully perform a certain task or challenge, and they have the self-confidence to complete the task successfully [18, 23]. We need more consumers who are aware of and committed to food-waste prevention, and we need strong policymakers with the right strategies to promote food-waste prevention systematically. In addition, the cultivation of professional competence in food-waste prevention could affect self-efficacy and could have a significant impact on behavioral intentions through self-efficacy. For example, a company could regularly hold competitions for the creative preparation of surplus food in order to encourage self-reflection in catering practitioners [78]. Competitions or mutual internal audits of surplus food and food-waste prevention among catering practitioners could be helpful to understand limitations and share effective approaches, which could then further improve the reduction of food waste company-wide.

5.4 Practical Implications

It has been shown that the greater the professional competence of employees, the greater the increase in self-efficacy. Sustainability issues have become an important consideration in the catering industry in recent years. The responsibility and prevention of food waste are not only related to consumers but, rather, are closely related to the policies and factors that govern the entire supply chain. It is an encouraging sign that food-waste prevention is now a major socio-economic issue globally.

Implicit traits could be adjusted through a longer period of education and training, psychological counseling, or accumulated practical training [37]. According to our results, the level of performance depends on the level of self-efficacy and professional competence. More time must be spent on the psychological development of professional competence before effectiveness can be achieved. Related courses such as sustainability ethics, workplace ethics, and professional ethics are also indispensable components. In these courses, such topics can be discussed via case studies, peer experience sharing, and industry experience seminars for exploring individual perceptions that contribute to the improvement of self-efficacy and overall behavior motivation.

Kitchen staff are crucial for food preparation and management in the catering industry. The knowledge–attitude–behavior model [79] suggested that increased environmental knowledge could enhance environmental awareness and concern, leading to more environmentally conscious behaviors. This knowledge also generated positive thoughts, concerns, and practices [80]. In other words, the environment changed the person, and only personal changes could lead to changes in attitudes towards surplus food and more sustainable efforts when processing ingredients and preparing food. These could, then, aid the catering industry in food-waste prevention, sustainable ethics, the sustainable use of ingredients, sustainable cooking, and environmental protection.

Sustainable Preparation Behavior for Kitchen Staff in order to Limit Food Waste

Abstract

The concepts of culinary sustainability and avoiding the waste of surplus food have become important sustainability trends today. How to integrate the handling of surplus food in the catering industry is a topic of concern in the industry. Kitchen staff are the vital soul of any restaurant, and we intend to discuss how kitchen staff actually behave and to explore factors that influence their behaviors in order to develop an implementation model for food-waste prevention. Therefore, this study explored a model of ethical sustainability, professional competence, self-efficacy, sustainable food preparation objectives, and sustainable food promotion and behavior focused on limiting food waste.  This study used a questionnaire and surveyed employees who had been employed for more than 6 months in Taiwan. From May to August 2022, 500 questionnaires were distributed; 415 valid questionnaires were retrieved, yielding a 90.2% recovery rate. According to the structural equation modeling analysis between the dimensions, ethical sustainability should have had a positive influence on professional competence in food-waste prevention and self-efficacy. Professional competence in food-waste prevention affected self-efficacy and behavioral intentions during food preparation; self-efficacy also significantly affected behavioral intentions towards sustainable food preparation. Similarly, behavioral intention had a positive influence on promoting sustainable behaviors. This model was a good fit for the acquired data. Professional competence in food-waste prevention was found to be the mediating factor between ethical sustainability and the behavioral intentions towards sustainable food preparation, and self-efficacy was the mediating factor between the professional competence in food-waste prevention and the behavioral intentions towards sustainable food preparation.

Keywords: surplus food, sustainable preparation behavior, food waste, kitchen staff, ethical sustainability, self- efficacy

  1. Introduction

Food is an important daily necessity. With advancements in the global economy, equipment, and food production efficiency, an abundant supply of food can provide for more people. From production to consumption, food is often discarded by consumers even though it is safe to eat because they have bought too much or have insufficient knowledge about the shelf life of food. It is estimated that one-third (1.3 billion tons) of the food produced globally for consumption is wasted every year [1,2]. The problem of food waste has been included in the World Health Organizations’ sustainable development goals (SDG 17) in order to reduce global human food waste by half and to reduce food loss by 2030. Food service and family experiences have been shown to be an appropriate way to address national and international sustainable goals [1].

The problem of surplus food management has also attracted significant attention from academics and other concerned practitioners because the normal dietary sources of a population are often interrupted due to limited resources, and experts believe that it is possible to reduce food insecurity via surplus food management. In addition to the commercial economic losses due to food production and waste, the social and environmental losses have also been significant, especially when foods that were still edible were not able to be distributed and sold, for various reasons, and were eventually discarded [3].

The Food and Agriculture Organization (FAO) of the United Nations has estimated that the annual global food waste across the entire food chain was USD 2.6 trillion. The Waste and Resources Action Plan estimated that food waste in the UK totaled around 10 million tons per year, 70% of which was avoidable [4]. Food waste is an urgent and serious issue in Western countries. The increase in the amount of food waste directly affects environmental changes, including greenhouse gas emissions and packaging pollution. In Poland, 9.2 million tons of grain have been lost every year, 53% of which was produced by consumers. In order to reduce consumer food waste, it was necessary to understand the behavioral factors [5]. The key problem of food waste has been primarily due to the supply chain, from production, preparation, and processing to transportation and consumption, all of which require considerable costs and resources. Therefore, practitioners and consumers should pay close attention to this issue from both the supply and the demand side. Reducing food waste is an important factor for sustainable development. The predicted population growth will require an adequate food system that includes the sustainable development of natural ecological resources [6,7]. Food waste occurs at all stages of the food supply chain, and the further down the supply chain, the greater the negative environmental, economic, and social impacts of food waste [8].

In restaurants, government intervention and enhanced waste disposal collection are necessary. Research has shown that changing consumer behavior, combined with corporate social responsibility, mitigated food waste [9]. The results of the study showed that many restaurants have prioritized reducing food waste. However, most of the final factors affecting food waste in restaurants were quality and specification issues during produ711ction. Generally, food waste was the worst in casual restaurants, and large meal portions resulted in more food waste [10].

The chefs are the souls of a restaurant. In addition to making good dishes, a good chef can also improve the product value while maintaining their social responsibility. Food oversupply has also been related to consumers' perceived concept of preservation. Consumers may purchase more than they need in order to save time [11, 12]. Among these food behaviors, discarding reusable ingredients as kitchen waste was the most severe (28.8% of ingredients were considered kitchen waste and discarded) [13]. Storing and sorting food in an orderly manner (for example, stacking old and new food, or sorting by frequency of use), and reordering regularly could reduce food waste [14, 15]. According to Secondi, Principato, and Laureti [16], one of the best courses of action to avoid wasting food was to estimate portion sizes correctly. Misplaced, forgotten, or expired leftovers were an issue when storing food [17].

Social cognitive theory (SCT) explains human behavior based on the dynamic interactions of individual experiences, the actions of others, and environment influences [18]. Self-efficacy is the belief in one's ability to change motivations, cognitive resources, and courses of action in order to meet the needs of a particular situation [19-22]. Research has found that self-efficacy predicted several job-related outcomes, such as work attitudes, training proficiency, and job performance. Bandura [23] pointed out that self-efficacy was the result of interactions with the external environment, personal professional competency, and learning effectiveness, which could then motivate behavior. Kitchen staff play a key role in food production from the kitchen to the table, and a sustainability-focused food revolution should be initiated for reducing carbon emissions and mitigating food waste to preserve the planet’s ecosystem.

Therefore, the attitudes of kitchen staff to avoid food waste could affect how they handle food, and sustainable cooking could provide substantial results in the future for environmental and food sustainability. This study explored the perspectives of kitchen staff towards limiting food waste by evaluating the behavioral patterns such as ethical sustainability, professional competence in food-waste prevention, self-efficacy, behavioral intentions towards sustainable food preparation, and sustainable food promotion from a practical perspective. This research could help promote sustainable cooking and food preparation, from the individual to the restaurant, and then to the catering industry. By encouraging consistent sustainability in the catering industry, we can more effectively reduce food waste.

  1. Theoretical background and hypotheses development

2.1 Surplus food and food waste

The conceptual model of availability, surplus, recoverability, and waste (ASRW) provided a clear definition of surplus food and distinguished it from food waste by introducing the concepts of availability, surplus, recoverability, and waste. According to this model, available food was defined as "all food in the food supply chain", which consisted of three parts: "human consumption", "surplus food", and "food scraps”. There has been no universally accepted definition of food waste and surplus food because there have been different interpretations from different perspectives [24].

According to the FAO, food waste consisted of three distinct problems. Food loss (FL) was the reduction in weight or nutritional value of food originally intended for human consumption. These losses were primarily due to processes in the supply chain, such as poor management, errors, and irregular processes (such as improper farming and harvesting), resulting in reduced edible weight. Food waste (FW) was the discarding of edible food, whether expired or spoiled, due to reasons such as oversupply, lack of a sales strategy, or excessive or unplanned purchases by individual consumers. It was also the waste of edible food due to improper distribution, storage, transportation, and preparation, both in the home and in restaurants [25].

Moreover, food waste occurs at all stages of the food chain: production, distribution, retail, and consumption processes [26]. Types of waste at all stages of the food chain are shown in Figure 1. Wasted food is any food that is lost due to spoilage or being discarded. "Waste food" as opposed to "food waste" refers to food that has not been used for its intended purpose but could be further used as a valuable resource [2].

Figure 1. Food waste styles in all stages of the food chain

According to scholars [27], surplus food could be divided into four categories: (1) Unprofitable crops refer to the market phenomenon after the harvest of agricultural products, and the surplus of grain sales is generated due to a stable sale price. When production is much higher than demand, it can lead to an oversupply in the market. (2) Non-perishable processed foods are foods with a long shelf life (such as dry goods, canned foods, etc.), damaged packaging, incorrect packaging, promotions near the expiration date, etc. (3) Perishable fresh foods are fresh fruits and vegetables, bread, dairy products, refrigerated ready-to-eat foods, etc.; products that are not sold because they are close to their expiration date but are still safe to eat; and foods that agricultural producers and importers may not be able to sell because the fruit or vegetables are irregular in shape and size. (4) Perishable foods include ready-to-eat foods such as sandwiches, cakes, and pastries [27].

2.2 Social Cognitive Theory

Social cognitive theory (SCT) was first proposed by Bandura, who combined the concepts of behaviorism and social learning [18]. Behavior is generated by the interaction between the individual and the environment, rather than determined by any single aspect. These factors include environmental impacts (the overall social environment, organizational policies, and culture); individual cognition and personal factors (personal motivation, attitude); and behavior (intention and willingness), all of which interact. Figure 2 shows the dynamic formation of the person, the environment, and behavior in SCT, where personal behavior is affected by the environment and influenced by personal subjective cognitions. The psychological process of self-introspection is achieved through the inner driving force of the individual, and the person feels and reacts to their external environment due to their inner thoughts, rather than being stimulated by the environment alone. Therefore, this research hypothesized that environmental factors could influence personal factors, and personal factors could influence behavioral factors.

                      Figure 2. Interactive model of social cognitive theory

Bandura [28] suggested that persons with high self-efficacy could have more enthusiasm, broader vision, broader thinking, and, therefore, courage to accept challenges. To avoid repeating the failures of others, most individuals would use the success of others as a guide. Rimal [29] also pointed out that SCT was an important basis for predicting food handling processes. Intention and self-efficacy could predict food-handling behavior [30]. Bandura [23] also suggested that personal past performance affected self-efficacy, and higher self-efficacy could also affect future performance. Behavioral improvements could be enhanced by offering continuous reminders during food handling [31].

Bandura [23] considered SCT the most suitable for explaining dynamic human behavior and regarded cognition and self-regulation as the primary framework of causality. The development of this model could cultivate the confidence of individuals to use certain abilities. However, it could also be affected by many internal factors, such as the level of self-efficacy, which would then result in different behaviors. Self-efficacy refers to a person's belief in their own capabilities and their degree of confidence in being able to achieve a specific outcome.

2.3 Ethical Sustainability Affects Personal Behavior

Waston and Meah [32] indicated that even if consumers had an environmental motivation to avoid food waste, the real motivation to reduce food waste was the "ethic of thrift". People avoided wasting food because it caused feelings of guilt that were then combined with environmental concerns [17]. Food represented the physical and symbolic connection between humans and nature [33]. A strong negative attitude towards food waste could be explained by our respect for nature, and this could hopefully be extended to the community [34].

Therefore, food waste could be perceived as a waste of precious resources, which would be unfair from an anthropocentric perspective [35]. Food waste has been considered immoral, so people need to be called upon to change their behaviors and attitudes concerning food waste [36, 37]. Therefore, food is a valuable resource, as it is not always readily available. A change in attitude could benefit both humans and nature by reducing the negative aspects of food production and improving the balance between humans and nature [38]. The “moral” rule for food waste behavior could be "don't waste food". With this in mind, our core ethical commitment was to avoid wasting food through more sensible consumption. Doing so could protect and preserve the world in which we live [39].

As indicated by the self-perception theory, previous behaviors affected future actions. The theory suggested that if individuals were encouraged to behave in environmentally conscious ways, they were more likely to make environmentally conscious decisions [40, 41]. This theory reinforced environmental self-identity, or an individual's perception of themselves as a person who is environmentally conscious [42]. In addition, many previous studies found that social norms and the behaviors of social groups had a positive influence on behavior [33,43].

Therefore, the following hypotheses were developed:

H1: Attitudes towards ethical sustainability affected professional competence in food-waste prevention.

H2: Attitudes towards ethical sustainability affected self-efficacy.

2.4 Relationships among professional competence, self-efficacy, and food-waste prevention

Food knowledge and skills, communication skills, business knowledge, and food concepts were considered the main professional abilities of kitchen staff [44]. To conform to social trends and achieve the following objectives, Hegarty and O'Mahony [45] developed the four major fields of culinary courses, including information; culinary arts; food and life sciences; and business. The basic skills needed for kitchen staff were food science, product preparation, finance, personal management, marketing, language, and computer skills [46]. Professional competence for food-waste prevention included two aspects: hygiene knowledge and culinary knowledge. Professional attitudes towards food-waste prevention included three dimensions, namely personal moral attitudes, attitudes towards food ingredients, and attitudes towards education and training. Skill competence comprised two dimensions: dish analysis and meal planning management [47]. In the restaurant and hospitality industry, both food preparers and consumers are responsible for food waste. Food waste has also been identified as a problem in schools, kindergartens, canteens, gas stations, bars, restaurants, canteens, and hotels [48].

The concept of self-efficacy first appeared in an article by Bandura [20]. It refers to an individual’s belief whether one can successfully complete a task. The emphasis was on ability, and it could promote individual actions in certain areas and affect future activities. This aspect was related to personal values. Bandura [18] also defined self-efficacy as “in order to deal with an impending situation, an individual makes a subjective judgment on whether he or she can successfully perform the required behavior. This judgment will decide how much effort he or she will put in when he or she personally faces difficulties, and how long it will last”. That is, self-efficacy could affect whether a person would persist in an action after setting goals. This was also related to whether a person had the motivation to overcome obstacles in the learning process and the learning strategies adopted when solving problems.

The theoretical basis of perceived behavioral control was originally derived from the self-efficacy theory, so perceived behavioral control and self-efficacy were conceptually similar [49]. Many studies have employed the theory of rational action (TRA) [50] or its extended theory of planned behavior (TPB) model [51] to understand kitchen workers' behavioral intentions towards food-waste prevention. The TRA theory suggested that previous attitude (i.e., approval or disapproval of a certain behavior) and existing subjective norms (i.e., whether the implementation of a certain behavior was approved by others) were two important influences on a person's behavioral intentions to implement a certain behavior [52]. Based on these previous studies, the current study aimed to understand whether ethical sustainability and professional competence in food-waste prevention were important indicators of behavioral intentions among kitchen staff. According to TRA and TPB, a person’s attitude indicated the favorable or unfavorable evaluation of a certain behavior, so it could predict behavioral intentions [51]. Therefore, the following hypotheses were developed:

H3: Professional competence could positively influence self-efficacy in food-waste prevention.

H4: Professional competence in food-waste prevention could positively influence the sustainable behavioral intentions during food preparation.

H5: Self-efficacy could positively influence sustainable behavioral intentions during food preparation.

H6: Sustainable behavioral intentions during food preparation could positively influence the promotion of sustainable preparation.

2.5 Mediating role of professional competence in food-waste prevention and self-efficacy

Based on the iceberg model proposed by Spencer and Spencer [53], professional competence could be classified as five basic factors: motivation, traits, self-concept, knowledge, and skills. Knowledge and skills were explicit traits: Both were easily affected by the external environment, both were trainable, and both could be enhanced through education and learning. Implicit traits, such as self-concept, traits, and motivation, could not be easily explored and developed and, therefore, would need to be inherent in the individual. The capacity for efficient food handling was the most important factor in preventing food waste [54]. Self-efficacy was important during food preparation and was also linked to preventing food waste [55].

There is a considerable relationship between the self-efficacy of food handling competence and general self-efficacy. Therefore, if you can enhance your personal self-efficacy, you can also enhance your competence in food handling. It was necessary to improve professional knowledge and skills, as it could improve self-efficacy and food handling behavior [56, 57, 58]. In addition, through food handling and cooking classes, self-efficacy could be enhanced, and behavioral ability could be improved [59]. Therefore, the following hypotheses were developed:

H7: Professional competence in food-waste prevention mediated the attitudes toward ethical sustainability and the sustainable behavioral intentions during food preparation.

H8: Self-efficacy mediated the professional competence in food-waste prevention and the sustainable behavioral intentions during food preparation.

  1. Methodology

3.1 Data Collection

A pilot study was conducted to ensure the reliability of the measurements. The first part of the questionnaire included questions related to all the variables of this study. The second part obtained the demographics of the participants, including gender, age, education, job title, work experience, and restaurant type.

To ensure the face validity of the questionnaire, we invited three professors to evaluate the validity of the predicted measurements. One hundred pilot-study questionnaires were returned with a valid return rate of 100%. The research process included item and factor analyses to determine the final questionnaire. All items in the questionnaire had good reliability (Cronbach's a > 0.70). In addition, according to the feedback of the participants, some items were revised.

This study included employees in Taiwan’s catering industry (including hotels, restaurants, and group-meal companies) as the research subjects. The convenience sampling and purposive sampling methods were adopted, and assistance was sought, through telephone interviews and e-mails, from individual practitioners in the catering industry who met the sampling criteria. From May to August 2022, questionnaires were sent and returned by mail. Out of 500 questionnaires, 415 valid questionnaires were returned, which yielded a 90.2% valid response rate.

3.2 Measures

  A questionnaire survey method was adopted for this study. All scales used to measure the constructs of this study were adopted from previous studies. The questionnaires used were divided into 6 parts: part 1, ethical sustainability, including 3 items; part 2, professional competence in food-waste prevention scale, including 29 items for 7 dimensions; part 3, self-efficacy, including 5 items; part 4, sustainable food behavioral intentions during food preparation, including 3 items; part 5, sustainable food promotion, including 4 items; and part 6, personal demographic.

For items measuring ethical sustainability, a three-item scale was adapted from the work of Lindeman and Väänänen [60]. This scale was based on a 5-point Likert scale, ranging from 1 (“strongly disagree”) to 5 (“strongly agree”). The Cronbach's α value of this scale was 0.832. The scale of professional competence in food-waste prevention was according to the professional competence in food-waste prevention scale of Ko and Lu [47], with 29 questions of 7 dimensions, and a 5-point Likert scale ranging from 1 (“strongly disagree”) to 5 (“strongly agree”). The average mean of the questions in each dimension was taken as the average mean for analysis, and the total Cronbach's α value of the scale was 0.926. To measure self-efficacy, 5 items from Chen, Gully, and Eden [61] were adapted and rated by a 5-point Likert scale, ranging from 1 (“strongly disagree”) to 5 (“strongly agree”). Finally, sustainable behavioral intentions during food preparation were measured using 10 items from Fang, Wang, and Hsu [62], and there were 3 questions. The Cronbach's α value of this scale was 0.907. The sustainable preparation promotion was adapted from the 4-item scale of Teng, Chih, and Wang [63]. The Cronbach's α value of this scale was 0.850. These items were rated by a 5-point Likert scale, ranging from 1 (“strongly disagree”) to 5 (“strongly agree”).

3.3 Data Analysis

The model comprised two components, a measurement model and a causal structural model. Before data analysis, we performed structural equation modeling (SEM) using the AMOS 21.0 application software. Whether the observed variables reflected the hypothesized latent variables and whether the measurement model adequately fit the data was performed with a confirmatory factor analysis (CFA). Path analysis was used in the study to evaluate the fit of the structural model and the hypothesized relationships among the latent variables [64]. As suggested by Anderson and Gerbing [64], we started with a CFA to determine whether all indicator variables adequately reflected their underlying structure and that the measurement model was acceptable. The fit index, comparative fit index (CFI), and the root-mean-square error of approximation (RMSEA) were used to estimate the overall model fit.

  1. Results

The subjects were mostly male in an age range of 21-40 years (70.1%). The education level was mostly high school and university. Non-management positions accounted for 71.8%, working experience of 1-4 years accounted for 28.2%, and 4-12 years accounted for 35.6%. The participants worked in hotel restaurants, independent restaurants, and restaurant chains (including central kitchens) (Table 1).

Table 1  Sociodemographic details of subjects

Background variables

Category

Number of persons

Percentage (%)

Gender

Male

217

61.8

Female

134

38.2

Age

20 years old or younger

8

2.3

21-30 years old

139

39.6

31-40 years old

107

30.5

41-50 years old

65

18.5

51-60 years old

27

7.7

61 years old or older

5

1.4

Educational level

Junior high school or below

14

4

High school ∕ vocational high school

130

37

Junior college

67

19.1

University

130

37

Graduate school

10

2.8

Job position

Management

99

28.2

Non-management

252

71.8

Working years

1 year or less

24

6.8

More than 1 year to 4 years (inclusive)

99

28.2

More than 4 year to 8 years (inclusive)

64

18.2

More than 8 year to 12 years (inclusive)

61

17.4

More than 12 year to 16 years (inclusive)

34

9.7

More than 16 year to 20 years (inclusive)

29

8.3

More than 20 years 

40

11.4

Employment

Restaurant in Hotel

120

34.2

Individual Restaurant

136

38.7

Group meal/central kitchen

82

23.4

Others

13

3.7

4.1 Measurement Model

The model fit indices then indicated an acceptable overall measurement model: x 2/df = 2.77; GFI = 0.87; AGFI = 0.84; RMSEA = 0.071; SRMR = 0.043; CFI = 0.98; and NFI = 0.97 [65]. Table 2 shows that all items had statistically significant (p < 0.01) [66] factor loadings, ranging from 0.65 to 0.89. The composite reliability values for all the constructs ranged from 0.88 to 0.95, indicating that all constructs had good internal reliability [67]. Furthermore, the average extracted variance (AVE) for each construct exceeded 0.50, which established the convergent validity of the measurement model [68]. Table 3 shows that the facets between all constructs were significant (p < 0.01), ranging from 0.459 to 0.755. In addition, the square root of AVE for each construct was also higher than the cross-correlation among all variables, so this model had discriminant validity [68].

4.2 Structural Model

The structural models were x 2/df = 3.10; GFI = 0.86; AGFI = 0.83; CFI = 0.98; NFI = 0.97; RMSEA = 0.077; and SRMR = 0.067, which indicated the good fit of the structural model [65]. The path analysis showed that environmental sustainability positively affected professional competence in food-waste prevention (b = 0.51, p < 0.000) and self-efficacy (b = 0.14, p < 0.000); professional competence in food-waste prevention positively affected self-efficacy (b = 0.65, p < 0.000) and sustainable behavioral intentions during food preparation  (b = 0.43, p <0.000). Self-efficacy also positively affected sustainable behavioral intentions during food preparation (b = 0.41, p <0.000), and sustainable behavioral intentions during food preparation positively affected sustainable preparation promotion (b = 0.80, p <0.000). Therefore, the results supported H1, H2, H3, H4, H5, and H6 (Figure 2).

4.3 Mediating Effect

The mediating effect of the perceived value in hotel restaurants was tested by performing Preacher and Hayes [69] bootstrapping techniques with 95% confidence intervals, using 10,000 bootstraps to study the direct, indirect, and overall effects of the proposed model. Ethical sustainability had a significant overall impact (standardized coefficient = 0.328, p < 0.05) on sustainable preparation promotion (Table 4). Ethical sustainability had a positive significant direct effect (standardized coefficient = 0.140, p < 0.05) and indirect effect (standardized coefficient = 0.329, p < 0.01) on self-efficacy. These results indicated that professional competence in food-waste prevention played a partial mediating role in the relationships between ethical sustainability and self-efficacy and supported H7 of this study. Similarly, professional competence in food-waste prevention had a significant direct effect (standardized coefficient = 0.425, p < 0.01) and indirect effect (standardized coefficient = 0.329, p < 0.01) on sustainable behavioral intentions during food preparation. These results indicated that self-efficacy had a partial mediating role in the relationships between professional competence in food-waste prevention and sustainable behavioral intentions during food preparation. Therefore, H8 was supported.

Table 2 Measurement model estimated parameters

Items

Std. factor loadings

Cronbach’s α

CR

AVE

E. Ethical sustainability

.832

.836

.630

E1. I will choose to use the food produced through a method that does not harm the animals (for example, enough space for raising).

.77

E2. I can consider environmentally friendly ways to prepare food and try to avoid food waste.

.79

E3. I will consider ways that will not disrupt the ecological balance to avoid waste in the food preparation process.

.82

C. Professional competence in food-waste prevention a

.926

.930

.655

C1 Hygiene knowledge

.80

C2 Cooking knowledge

.85

C3 Personal moral attitude

.72

C4 Attitude towards ingredients

.81

C5 Education and training attitude

.83

C6 Dish analysis skills

.81

C7 Planning and management skills

.84

S. Self-efficacy

.907

.910

.670

S1. I can set a goal to avoid food waste, and I think I can get the desired result.

.76

S2. I will be able to successfully overcome several challenges to avoid food waste.

.83

S3. I believe that I can effectively perform several different tasks to avoid food waste.

.84

S4 Compared with others, I can complete most tasks well to avoid food waste.

.82

S5. Even if I encounter difficulties in food-waste prevention, I can try my best to achieve the goal.

.84

I. Sustainable behavioral intentions during food preparation

.879

.843

.696

I1. I am willing to take food preparation activities for a healthy and sustainable diet.

.78

I2. I will encourage others to participate in healthy and sustainable food preparation together.

.86

I3. I will adjust and change my food preparation habits to promote sustainable positive effects of food preparation on health and the environment.

.86

B. Sustainable preparation promotion behavior

.850

.856

.600

B1. I implement green procurement (e.g., purchase local food, support green diet).

.65

B2. I implement and supervise others to reduce food waste in preparation.

.77

B3. I participate in the action of a healthy and sustainable diet.

.89

B4. I support the sustainable food/snacks activities organized by the government.

.77

a Twenty-nine items measured for professional competence load onto seven factors.

Table 3 Relevance among various dimensions and the correlations among all the constructs

Variable

Average mean

Standard deviation

E1

C

S

I

B

E. Ethical sustainability

4.10

.58

.8182

C. Professional competence in food-waste prevention

4.39

.55

.459

.789

S. Self-efficacy

4.12

.51

.712

.497

.806

I. Sustainable behavioral intentions during food preparation

4.23

.61

.680

.537

.665

.844

B. Sustainable preparation promotion behavior

4.13

.61

.680

.519

.721

.755

.777

**p < 0.01; N =351

1E: Ethical sustainability; C: Professional competence in food-waste prevention; S: Self-efficacy; I: Sustainable behavioral intentions during food preparation; B: Sustainable preparation promotion

2The square root of the AVE for discriminant validity is italicized along the diagonal.

Figure 3. Path analysis of each construct for overall structure model

(E: Ethical sustainability; C: Professional competence in food-waste prevention; S: Self-efficacy; I: Sustainable behavioral intentions during food preparation; B: Sustainable preparation promotion)

Table 4 The total, indirect, and direct effects of the hypothetical model

Bootstrapping

Estimate

Bias-corrected

percentile 95% CI

Lower

Upper

Two-tailed

significance

Indirect effect

Ethical sustainability→Professional competence in food-waste prevention →Self-efficacy

.329

.224

.432

.007**

Professional competence in food-waste prevention →Self-efficacy→ Sustainable behavioral intentions during food preparation

.266

.181

.397

.004**

Direct effect

Ethical sustainability→Professional competence in food-waste prevention

.510

.385

.603

.019*

Ethical sustainability→Self-efficacy

.140

.032

.277

.021*

Professional competence in food-waste prevention →Self-efficacy

.645

.483

.740

.009**

Professional competence in food-waste prevention → Sustainable behavioral intentions during food preparation

.425

.250

.617

.006**

Self-efficacy→ Sustainable behavioral intentions during food preparation

.413

.245

.577

.018*

Sustainable behavioral intentions during food preparation→ Sustainable preparation promotion

.798

.721

.859

.009**

Total effect

Ethical sustainability→Sustainable preparation promotion

.328

.234

.420

.018*

  1. Conclusion and Discussion

5.1 Conclusion

According to the data analysis results, the dimensions of ethical sustainability, professional competence in food-waste prevention, self-efficacy, sustainable preparation intention, and sustainable preparation promotion were all found to be significantly and positively correlated. This study first analyzed the CFA among the various dimensions, and the results showed that this model had a good fit. All indices passed the standards, indicating a compliant fit. This model could interpret the impact of each dimension. Ethical sustainability positively affected professional competence in food-waste prevention and self-efficacy, professional competence in food-waste prevention positively affected self-efficacy and sustainable behavioral intentions during food preparation. Similarly, self-efficacy positively affected sustainable behavioral intentions during food preparation, and sustainable behavioral intentions during food preparation significantly and positively affected sustainable preparation promotion. Therefore, H1, H2, H3, H4, H5, and H6 all had empirical support. Moreover, professional competence in food-waste prevention was the factor mediating ethical sustainability to sustainable behavioral intentions during food preparation, while self-efficacy was the factor mediating professional competence in food-waste prevention to sustainable behavioral intentions during food preparation. Both H7 and H8 were supported.

5.2 Discussion

Ethical sustainability positively affected professional competence in food-waste prevention and self-efficacy, while professional competence in food-waste prevention was the mediating factor between ethical sustainability and sustainable behavioral intentions during food preparation. Edwards and Mercer [70] strongly suggested that lifestyle choices, such as diet and career choices, had an ethical basis. Waston and Meah [32] explained that the primary driver for lower food waste was an “ethic of thrift”, which was the ethical motivation to act appropriately and “be thrifty”. The ethical rule for food waste behavior was "don't waste food". Therefore, the core ethical commitment in the study was to avoid wasting food, thus making consumption more reasonable [39]. Therefore, kitchen staff must have a personal imperative to prevent food waste in order to establish relevant behaviors and build professional competence in that arena.

Based on social cognitive theory in the context of food waste, females were more cognitively advanced than males. Females more closely monitored reducing food waste, so they were more motivated [71]. This contributed to their identification of more motivational factors that could have had a positive effect on their behavior. The environment affected the individuals and, in turn, their behavior. If actual catering producers were more aware of food waste, it could help enhance their professional competence in food-waste prevention.

The research on food waste has shown that preventing food waste required different culinary skills and knowledge, so leftovers food could be re-purposed [72, 73]. During storage, leftovers were often misplaced or left in the refrigerator for too long and spoiled [14, 15, 17]. Therefore, professional competence in food preparation is important.

 Professional competence in food-waste prevention positively affected self-efficacy and sustainable behavioral intentions during food preparation. Self-efficacy was the mediating factor between professional competence in food-waste prevention and sustainable behavioral intentions during food preparation. Lin [74] defined knowledge-sharing as a culture of social interaction, and the composition of this culture included knowledge exchange, experiential inheritance, and mutual learning among employees because the growth of knowledge could increase self-efficacy. Ding and Huang [75] suggested that mutual assistance, cooperation, and knowledge-sharing were critical for obtaining a competitive advantage. Bandura [20] suggested that individuals with high self-efficacy could have more enthusiasm, broader vision, broader thinking, and thus, more courage to accept challenges, and these behaviors were predominantly guided by the successful experiences of others. Self-efficacy and behavioral intentions could, indeed, predict food handling behaviors [30].

 Self-efficacy in processing food was important and correlated with general self-efficacy. Therefore, relevant educational policies should support consumers in developing self-efficacy related to sustainable food handling. Previous research pointed to the need for self-efficacy to enhance knowledge and skills, yet building self-efficacy required the inclusion of food handling and cooking classes in school curricula based on real-world experiences and the social marketing of public policy [45, 56, 58,  64].

Self-efficacy could be conceptualized with different levels of linkages, ranging from general self-efficacy (a person's ability to exercise self-control in different situations) to domain-specific self-efficacy (related to a person's ability to control behavior in specific settings (such as work and home)) to task-related self-efficacy (such as confidence in being able to correctly sort leftover waste) [76]. Domain- and task-related self-efficacy was positively correlated with general self-efficacy [77]. We explored the different hierarchies of self-efficacy in food waste in order to provide insight into the root causes of food waste. Therefore, we investigated the causes of household food waste by focusing on the effect of self-efficacy on consumer food waste [76]. Therefore, enhancing professional competence in food-waste prevention could improve self-efficacy and help promote more sustainable practices around food preparation and surplus food.  

5.3 Management Implications

This study shed light the factors influencing food waste prevention in the catering industry and could help the government and catering stakeholders to made specific recommendations for promoting food-waste prevention. The results showed that the ethical sustainability attitudes of kitchen staff affected professional competence in food-waste prevention and self-efficacy. Ethical sustainability among kitchen staff was related to their individual experiences and work/education environments. Therefore, the awareness and education provided by companies and governments concerning food-waste prevention could indirectly affect the concept of ethical sustainability among kitchen staff. Influencing the knowledge, attitudes, and skills of professional competence in food-waste prevention, and continuously promoting better strategies for surplus food, are important factors for governmental and corporate sustainability promotion. Self-efficacy indicates the individual believes that they can successfully perform a certain task or challenge, and they have the self-confidence to complete the task successfully [18, 23]. We need more consumers who are aware of and committed to food-waste prevention, and we need strong policymakers with the right strategies to promote food-waste prevention systematically. In addition, the cultivation of professional competence in food-waste prevention could affect self-efficacy and could have a significant impact on behavioral intentions through self-efficacy. For example, a company could regularly hold competitions for the creative preparation of surplus food in order to encourage self-reflection in catering practitioners [78]. Competitions or mutual internal audits of surplus food and food-waste prevention among catering practitioners could be helpful to understand limitations and share effective approaches, which could then further improve the reduction of food waste company-wide.

5.4 Practical Implications

It has been shown that the greater the professional competence of employees, the greater the increase in self-efficacy. Sustainability issues have become an important consideration in the catering industry in recent years. The responsibility and prevention of food waste are not only related to consumers but, rather, are closely related to the policies and factors that govern the entire supply chain. It is an encouraging sign that food-waste prevention is now a major socio-economic issue globally.

Implicit traits could be adjusted through a longer period of education and training, psychological counseling, or accumulated practical training [37]. According to our results, the level of performance depends on the level of self-efficacy and professional competence. More time must be spent on the psychological development of professional competence before effectiveness can be achieved. Related courses such as sustainability ethics, workplace ethics, and professional ethics are also indispensable components. In these courses, such topics can be discussed via case studies, peer experience sharing, and industry experience seminars for exploring individual perceptions that contribute to the improvement of self-efficacy and overall behavior motivation.

Kitchen staff are crucial for food preparation and management in the catering industry. The knowledge–attitude–behavior model [79] suggested that increased environmental knowledge could enhance environmental awareness and concern, leading to more environmentally conscious behaviors. This knowledge also generated positive thoughts, concerns, and practices [80]. In other words, the environment changed the person, and only personal changes could lead to changes in attitudes towards surplus food and more sustainable efforts when processing ingredients and preparing food. These could, then, aid the catering industry in food-waste prevention, sustainable ethics, the sustainable use of ingredients, sustainable cooking, and environmental protection.

5.5 Research Limitations and Future Studies

There were some limitations that should be considered. First, we primarily focused on those working in hotels and restaurants, as well as those working for group-meal providers, as a whole. To understand the differences in their environments, they should be considered separately. In the future, chain restaurants and central catering factories may be considered as the subjects, and different results may be obtained since other products in these catering sectors have been more standardized.

Due to the survey method used to collect data at a specific point in time, the results may be somewhat biased because the research instrument was a self-report survey [81]. Therefore, future studies should design more rigorous research procedures, preferably collecting data on predictor and outcome variables from different respondents. Finally, this study involved quantitative research, which could be supplemented by qualitative research in the future to understand issues associated with its implementation. We could have collected more factors affecting food waste in order to explore the actual situation of sustainable preparation behavior, such as the types of service and the ranks of kitchen staff, as well as to discuss the factors regulating sustainable preparation behavior.

References

  1. Amicarelli, V., Lagioia, G., and Bux, C. (2021). Global warming potential of food waste through the life cycle assessment: An analytical review. Environmental Impact Assessment Review, 91(11), 106677.
  2. Food and Agriculture Organization (FAO) (2022). Food Losses and Food Waste. Available online: http://www.fao.org/food-loss-and-food-waste/flw-data) (accessed on 10 Oct, 2022).
  3. Tarasuk, V., & Eakin, J. (2005). Food Assistance Through "Surplus" Food: Insights from an Ethnographic Study of Food Bank Work. Agriculture and Human Values, 177-186.
  4. WRAP (2022). Food Surplus and Waste in the UK Key Facts. Available online: https://wrap.org.uk/resources/report/food-surplus-and-waste-uk-key-facts (accessed on 20 Aug, 2022).
  5. Jungowska, J., Kulczy´ nski,B., Sidor, A., Gramza-MichaÅ‚owska, A. (2021). Assessment of Factors Affecting the Amount of Food Waste in Households Run by Polish Women Aware of Well-Being. Sustainability, 13, 976. https://doi.org/ 10.3390/su13020976
  6. Foley, J. A., Ramankutty, N., Brauman, K. A., Cassidy, E. S., Gerber, J. S., Johnston, M., & Zaks, D. P. M. (2011). Solutions for a cultivated planet. Nature, 478(7369), 337–342.
  7. Willett, W., Rockström, J., Loken, B., Springmann, M., Lang, T., Vermeulen, S., Garnett, T., Tilman, D.,Declerck, F., Wood, A., Jonell, M., Clark, M., Gordon, L., Fanzo, J., Hawkes, C., Zurayk, R., Rivera, J., Vries, W., Sibanda, L., & Murray, C. (2019). Food in the Anthropocene: the EAT–Lancet Commission on healthy diets from sustainable food systems. The Lancet. 393.(10170). 447-492.
  8. Stenmarck, Â., Jensen, C., Quested, T., Moates, G., Buksti, M., Cseh, B., & Scherhaufer, S. (2016). Estimates of European food waste levels. IVL Swedish Environmental Research Institute.
  9. Stirnimann, A. and Zizka, L. (2022). Waste not, want not: Managerial attitudes towards mitigating food waste in the Swiss-German restaurant industry. Journal of Foodservice Business Research, 25(3), 302–328
  10. McAdams, B., Massow, M., Gallant, M., and Hayhoe, M.A. (2019). A cross industry evaluation of food waste in restaurants. Journal of Foodservice Business Research, 22(5), 449–466.https://doi.org/10.1080/15378020.2019.1637220
  11. Ganglbauer, E., Fitzpatrick, G., Comber, R. (2013). Negotiating food waste: using a practice lens to inform design. ACM Trans. Computer Human Interaction, 20, 1-25.
  12. Graham-Rowe, E., Jessop, D.C., and Sparks, P. (2014). Identifying motivations and barriers to minimizing household food waste. Resources, Conservation and Recycling, 84, 15-23.
  13. Wang, Y.F. (2016). Improving Culinary Education by Examining the Green Culinary Behaviors of Hospitality College Students. Journal of Hospitality & Tourism Education, 28(1), 1-9, DOI: 10.1080/10963758.2015.1127167
  14. Farr-Wharton, G., Foth, M., Choi, J.H.J. (2014). Identifying factors that promote consumer behaviours causing expired domestic food waste. Journal of Consumer Behavior,13, 393-402.
  15. Waitt, G., Phillips, C. (2016). Food waste and domestic refrigeration: a visceral and material approach. Social & Cultural Geography, 17, 359-379.
  16. Secondi, L., Principato, L., and Laureti, T. (2015). Household food waste behaviour in EU- 27 countries: a multilevel analysis. Food Policy, 56, 25-40.
  17. Blichfeldt, B.S., Mikkelsen, M., and Gram, M.(2015). When it stops being food: The edibility, ideology, procrastination, objectification and internalization of household food waste. Food, Culture & Society, 18, 89-105.
  18. Bandura, A. (1982). Self-efficacy mechanism in human agency. American Psychologist, 37, 122-147.
  19. Wood, R., & Bandura, A. (1989). Impact of conceptions of ability on self-regulatory mechanisms and complex decision making. Journal of Personality and Social Psychology, 56,407-415.
  20. Bandura, A. (1977). Self-Efficacy: Toward a Unifying Theory of Behavioral Change. Psychological Review, 84(2), 191-215.
  21. Gist, M. E., & Mitchell, T. R. (1992). Self-efficacy: A theoretical analysis of its determinants and malleability. Academy of Management Review, 17, 183-211.
  22. Stajkovic, A. D., & Luthans, F. (1998). Self-efficacy and work-related performance: A metaanalysis. Psychological Bulletin, 124, 240-261.
  23. Bandura, A. (1986). Social foundations of thought and action. Englewood Cliffs, NJ: Prentice-Hall.
  24. Garrone, P., Melacini, M., & Perego, A. (2013). Feed the Hungry: The Potential of Surplus Food Recovery. Milan, Italy: Edizioni AngeloGuerini e Associati SpA.
  25. Parfitt, J., Barthel, M., & Macnaughton, S. (2010). Food waste within food supply chains: Quantification and potential for change to 2050. Philosophical Transactions of the Royal Society B, 365, 3065–3081.
  26. Göbel, C., Langen, N., Blumentha, A., Teitscheid, P., and Ritter, G. (2015). Cutting Food Waste through Cooperation along the food Supply Chain. Sustainability, 7, 1431-1438.
  27. Hawkes, C., & Webster, J. (2000). Too much and too little? debates on surplus food redistribution. London: Sustain.
  28. Bandura, A. (1977). Social learning theory. Englewood Cliff, NJ: Prentice Hall.
  29. Rimal, R.N. (2000). Closing the knowledge-behavior gap in health promotion: The mediating role of self-efficacy. Health Communication, 12(3), 219–237.
  30. Bearth, A., Cousin, M. -E., & Siegrist, M. (2014). Investigating novice cooks' behavior change: Avoiding cross-contamination. Food Control, 40, 26–31.
  31. Mullan, B., Allom, V., Fayn, K., and Johnston, I. (2014). Building habit strength: A pilot intervention designed to improve food-safety behavior. Food Research International, 66, 274-278.
  32. Watson, M., & Meah, A. (2012). Food, waste and safety: Negotiating conflicting social anxieties into the practices of domestic provisioning. The Sociological Review, 60(S2), 102–120.
  33. Goldstein, N. J., Cialdini, R. B., & Griskevicius, V. (2008). A room with a viewpoint: Using norms to motivate environmental conservation in hotels. Journal of Consumer Research, 35, 472-482. doi:10.1086/586910
  34. Glendinning, C. (1994). My name is Chellis and I’m in recovery from Western civilization. Gabriola Island: New Catalyst Books.
  35. Abram, D. (1996) The spell of the sensuous. New York: Vintage Books.
  36. MacMillan, T. (2009) What is wrong with waste? Food Ethics, 4 (3), p. 4.
  37. Stancu, V., Haugaard, P. and Lähteenmäki, L. (2016) Determinants of consumer food waste behaviour: two routes to food waste. Appetite, 96, 7-17.
  38. Gjerris, M., & Gaiani, S. (2013). Household food waste in Nordic countries:Estimations and ethical implications. Nordic Journal of Applied Ethics, 7 (1),6-23.
  39. Lehtokunnas, T., Mattila, M., Narvanen, E., and Mesiranta, N. (2020). Towards a circular economy in food consumption: Food waste reduction practices as ethical work. Journal of Consumer Culture, https://doi.org/10.1177/1469540520926252
  40. Lacasse, K. (2016). Don’t be satisfied, identify! Strengthening positive spillover by connecting pro-environmental behaviors to an “environmentalist” label. Journal of Environmental Psychology, 48, 149-158. doi:10.1016/j.jenvp.2016.09.006
  41. van der Werff, E., Steg, L., & Keizer, K. (2014). Follow the signal: When past proenvironmental actions signal who you are. Journal of Environmental Psychology, 40, 273-282. doi:10.1016/j.jenvp.2014.07.004
  42. Whitmarsh, L., & O’Neill, S. (2010). Green identity, green living? The role of pro-environmental self-identity in determining consistency across diverse proenvironmental behaviours. Journal of Environmental Psychology, 30, 305-314. doi:10.1016/j.jenvp.2010.01.003
  43. Cialdini, R. B., Kallgren, C. A., & Reno, R. R. (1991). A focus theory of normative conduct: A theoretical refinement and reevaluation of the role of norms in human behavior. Advances in Experimental Social Psychology, 24, 201-234. doi:10.1016/S0065-2601(08)60330-5
  44. Harrington, R.J., Mandabach, K.H., VanLeeuwen, D., and Thibodeaux, W. (2005). A multi-lens framework explaining structural differences across foodservice and culinary education. International Journal of Hospitality Management, 24, 198-218.
  45. Hegarty, J. A., & O’Mahony, G. B. (2001). Gastronomy: A phenomenon of cultural expressionism and an aesthetic for living. International Journal of Hospitality Management, 20, 3-13.
  46. Agut, S., Grau, R., and Peiro, J.M. (2003). Competency needs among managers from Spanish hotels and restaurants and their training demands. International Journal of Hospitality Management, 22, 281-295.
  47. Ko,W.H. and Lu, M.Y. (2021). Developing a professional competence scale for kitchen staff: Food value and availability for surplus food. International Journal of Hospitality Management, 95, 102926.
  48. Tekin, Ö.A.; Ilyasov, A. (2017). The Food Waste in Five-Star Hotels: A Study on Turkish Guests’ Attitudes. Journal of Tourism and Gastronomy Studies, 5, 13–31.
  49. Ajzen, I. (2002) Perceived behavioral control, self-efficacy, locus of control, and the theory of planned behavior. Journal of Applied Social Psychology, 32(4), 665-683.
  50. Ajzen, I. and Fishbein, M. (1980). Understanding Attitudes and Predicting Social Behavior, Prentice-Hall, Englewood Cliffs, NJ.
  51. Ajzen, I. (1991). The theory of planned behavior. Organizational Behavior and Human Decision Processes, 50(2), 179-211.
  52. Chan, E.S., Hon, A.H., Chan, W. and Okumus, F. (2014). What drives employees’ intentions to implement green practices in hotels? the role of knowledge, awareness, concern and ecological behavior. International Journal of Hospitality Management, 40, 20-28.
  53. Spencer, L. M., & Spencer, S. M. (1993). Competence at work- Models for Superior Performance. NY: John Wiley & Sons.
  54. Romani, S., Grappi, S., Bagozzi, R.P. and Barone, A.M. (2018) Domestic food practices. A study of food management behaviors and the role of food preparation planning in reducing waste. Appetite, 121, 215-227.
  55. Lavelle, F., McGowan, L., Spence, M., Caraher, M., Raats, M.M., Hollywood, L., McDowell, D., McCloat, A., Mooney, E. and Dean, M. (2016) Barriers and facilitators to cooking from 'scratch' using basic or raw ingredients. A qualitative interview study. Appetite, 107, 383-391.
  56. Thyberg, K.L. and Tonjes, D.J. (2016) Drivers of food waste and their implications for sustainable policy development. Resources, Conservation and Recycling, 106, 110-123.
  57. Hebrok, M. and Boks, C. (2017) Household food waste. Drivers and potential intervention points for design—an extensive review. Journal of Cleaner Production, 151, 380-392.
  58. Schanes, K., Dobernig, K. and Gözet, B. (2018) Food waste matters—a systematic review of household food waste practices and their policy implications. Journal of Cleaner Production, 182, 978-991.
  59. Andreasen, A.R. (2002) Marketing social marketing in the social change marketplace. Journal of Public Policy & Marketing, 21(1), 3-13.
  60. Lindeman, M. & Väänänen, M. (2000). Measurement of ethical food choice motives. Appetite, 34(1),55-59.
  61. Chen, G., Gully, S.M., Eden, D. (2001).Validation of a New General Self-Efficacy Scale. Organizational Research Methods, 4(1), 62-83.
  62. Fang, W.T., Ng, E., Wang, C.M., and Hsu, M.L. (2017). Normative Beliefs, Attitudes, and Social Norms: People Reduce Waste as an Index of Social Relationships When Spending Leisure Time. Sustainability, 9, 1696.
  63. Teng, C.C., Chih, C. and Wang, Y.C. (2020). Decisional Factors Driving Household Food Waste Prevention: Evidence from Taiwanese Families. Sustainability, 12, 6666; doi:10.3390/su12166666.
  64. Anderson, J.C. and Gerbing, D.W. (1988). Structural equation modeling in practice: a review and recommended two-step approach. Psychological Bulletin, 103(3), 411-423.
  65. Hair, J.F., Anderson, R.E., Tatham, R.L. and Black, W.C. (2010). Multivariate Data Analysis, 7th ed., Prentice Hall, Englewood Cliffs, NJ.
  66. Hu, L.T. and Bentler, P.M. (1999). Cutoff criteria for fit indexes in covariance structure analysis: conventional criteria versus new alternatives. Structural Equation Modeling: A Multidisciplinary Journal, 6(1), 1-55.
  67. Jöreskog, K.G. and Sörbom, D. (1989), LISREL 7: A Guide to the Program and Applications, SPSS, Chicago, IL.
  68. Fornell, C. and Larcker, D.F. (1981). Evaluating structural equation models with unobservable variables and measurement error. Journal of Marketing Research, 18(1), 39-50.
  69. Preacher, K.J. and Hayes, A.F. (2008). Asymptotic and resampling strategies for assessing and comparing indirect effects in multiple mediator models. Behavior Research Methods, 40, 879-891.
  70. Edwards, F. & Mercer, D. (2007). Gleaning from Gluttony: an Australian youth subculture confronts the ethics of waste. Australian Geographer, 38(3), 279-296.
  71. Goodman-Smith, F., Mirosa, R., and Mirosa, M. (2020). Understanding the E_ect of Dining and Motivational Factors on Out-Of-Home Consumer Food Waste. Sustainability, 12, 6507.
  72. Cappellini, B. (2009). The sacrifice of re-use: The travels of leftovers and family relations. Journal of Consumer Behavior, 8, 365-375.
  73. Southerton,D. & Yates,L. (2015). Exploring food waste through the lens of social practice theories: Some reflections on eating as compound practice. In: Ekstrom KM (ed.) Waste Management and Sustainable Consumption: Reflections on Consumer Waste. London; Chicago, IL: Routledge, 133-149.
  74. Lin, H. F. (2007). Effects of extrinsic and intrinsic motivation on employee knowledge sharing intentions. Journal of Information Science, 33(2), 135-149.
  75. Ding, X. H., & Huang, R. H. (2010). Effects of knowledge spillover on interorganizational resource sharing decision in collaborative knowledge creation. European Journal of Operational Research, 201, 949-959.
  76. Grether, T., Sowislo, J.F. and Wiese, B.S. (2018) Top-down or bottom-up? Prospective relations between general and domain-specific self-efficacy beliefs during a work-family transition. Personality and Individual Differences,121, 131-139.
  77. Luszczynska, A., Scholz, U. and Schwarzer, R. (2005) The general self-efficacy scale: multicultural validation studies. The Journal of Psychology, 139(5), 439-457.
  78. Saks, A. M. (1995). Longitudinal field investigation of the moderating and mediating effects of self-efficacy on the relationship between training and newcomer adjustment. Journal of Applied Psychology, 80(2), 211–225. https://doi.org/10.1037/0021-9010.80.2.211
  79. Kollmuss, A. and Agyeman, J. (2002). Mind the gap: why do people act environmentally and what are the barriers to pro-environmental behavior?. Environmental Education Research, 8(3), 239-260.
  80. Ruiz-Molina, M.E. and Gil-Saura, I. (2008). Perceived value, customer attitude and loyalty in retailing. Journal of Retail & Leisure Property, 7(4), 305-314.
  81. Podsakoff, P.M., MacKenzie, S.B., Lee, J.Y. and Podsakoff, N.P. (2003). Common method biases in behavioral research: a critical review of the literature and recommended remedies. Journal of Applied Psychology, 88(5), 879-903.

Round 2

Reviewer 1 Report

Comments and Suggestions for Authors

Dear authors,

thank you for providing the revised version of your manuscript.

I still have several remarks:

Abstract:

Remove or revise this sentence. It does not have scientific soundness."This model was a good fit for the acquired data."

You should include the aim of your manuscript in the abstract as well.

Introduction:

You must provide a reference for this statement. "The problem of food waste has been included in the World Health Organizations’ sustainable develop- ment goals (SDG 17) in order to reduce global human food waste by half and to reduce food loss by 2030"

Results:

Figure 3. is still not a reader-friendly one. Something more it is very difficult to understand and incomprehensible.

5. Conclusion and Discussion:

It is not appropriate the Conclusion to be before Discussion.

The Conclusion must be separated in different section.

Something more, your conclusion is not well constructed and in fact it does not provide any conclusion.

References: must be revised according to the MDPI guidelines. 

Comments on the Quality of English Language

 Minor editing of English language required

Author Response

To reviewers 1:

Thank you for your comments and suggestions. I appreciate the time and effort that the reviewers have dedicated to providing your valuable feedback on my manuscript. I had major revised in this paper as final attached again as reviewers commend. Please see the comments below.

The following are our responses to the Reviewers’ 1 comments:

  1. Remove or revise this sentence. "This model was a good fit for the acquired data." You should include the aim of your manuscript in the abstract as well.

Ans:  I had removed this sentence and describe the aim in the abstract. Please see below.

Abstract

The concepts of culinary sustainability and avoiding the waste of surplus food have become important sustainability trends today. How to integrate the handling of surplus food in the catering industry is a topic of concern in the industry. Kitchen staff are the vital soul of any restaurant, and we intend to discuss how kitchen staff actually behave and to explore factors that influence their behaviors in order to develop an implementation model for food-waste prevention. Therefore, this study explored a model of ethical sustainability, professional competence, self-efficacy, sustainable food preparation objectives, and sustainable food promotion and behavior focused on limiting food waste. Using structural equation modeling (SEM) to understand the relationship between various constructs. This study used a questionnaire and surveyed employees who had been employed for more than 6 months in Taiwan. From May to August 2022, 500 questionnaires were distributed; 415 valid questionnaires were retrieved, yielding a 90.2% recovery rate. According to the structural equation modeling analysis between the dimensions, ethical sustainability should have had a positive influence on professional competence for food-waste prevention and self-efficacy. Professional competence for food-waste prevention affected self-efficacy and behavioral intentions during food preparation; self-efficacy also significantly affected behavioral intentions towards sustainable food preparation. Similarly, behavioral intention had a positive influence on promoting sustainable behaviors. There is a significant relationship between all constructs in this study. Professional competence for food-waste prevention was found to be the mediating factor between ethical sustainability and the behavioral intentions towards sustainable food preparation, and self-efficacy was the mediating factor between the professional competence for food-waste prevention and the behavioral intentions towards sustainable food preparation.

---------------------------------------------------------------------------

  1. Introduction:You must provide a reference for this statement. "The problem of food waste has been included in the World Health Organizations’ sustainable development goals (SDG 17) in order to reduce global human food waste by half and to reduce food loss by 2030"

Ans: I had added. Please see the revised paper as below.

Food is an important daily necessity. With advancements in the global economy, equipment, and food production efficiency, an abundant supply of food can provide for more people. From production to consumption, food is often discarded by consumers even though it is safe to eat because they have bought too much or have insufficient knowledge about the shelf life of food. About 1.3 billion tons of food is wasted in the world every year, accounting for about 1/3 of production [1,2]. The problem of food waste has been included in the World Health Organizations’ sustainable development goals (SDG 17) in order to reduce global human food waste by half and to reduce food loss by 2030 [2]. Food service and family experiences have been shown to be an appropriate way to address national and international sustainable goals [1].

-------------------------------------------------------------------------------------

  1. Results: Figure 3. is still not a reader-friendly one. Something more it is very difficult to understand.

Ans : I had revised again, please see as below.

Figure 3. Structural model for path analysis of ethical sustainability, professional competence for food-waste prevention, self-efficacy and sustainable behavioral intentions during food preparation

  1. Conclusion and Discussion: It is not appropriate the Conclusion to be before Discussion. The Conclusion must be separated in different section.

Ans:  I had revised as below.

  1. Discussion

Ethical sustainability positively affected professional competence for food-waste prevention and self-efficacy, while professional competence for food-waste prevention was the mediating factor between ethical sustainability and sustainable behavioral intentions during food preparation. Edwards and Mercer [70] strongly suggested that lifestyle choices, such as diet and career choices, had an ethical basis. Waston and Meah [32] explained that the primary driver for lower food waste was an “ethic of thrift”, which was the ethical motivation to act appropriately and “be thrifty”. The ethical rule for food waste behavior was "don't waste food". Therefore, the core ethical commitment in the study was to avoid wasting food, thus making consumption more reasonable [39]. Therefore, kitchen staff must have a personal imperative to prevent food waste in order to establish relevant behaviors and build professional competence in that arena.

Based on social cognitive theory in the context of food waste, females were more cognitively advanced than males. Females more closely monitored reducing food waste, so they were more motivated [71]. This contributed to their identification of more motivational factors that could have had a positive effect on their behavior. The environment affected the individuals and, in turn, their behavior. If actual catering producers were more aware of food waste, it could help enhance their professional competence for food-waste prevention.

The research on food waste has shown that preventing food waste required different culinary skills and knowledge, so leftovers food could be re-purposed [72, 73]. During storage, leftovers were often misplaced or left in the refrigerator for too long and spoiled [14, 15, 17]. Therefore, professional competence in food preparation is important.

 Professional competence for food-waste prevention positively affected self-efficacy and sustainable behavioral intentions during food preparation. Self-efficacy was the mediating factor between professional competence for food-waste prevention and sustainable behavioral intentions during food preparation. Lin [74] defined knowledge-sharing as a culture of social interaction, and the composition of this culture included knowledge exchange, experiential inheritance, and mutual learning among employees because the growth of knowledge could increase self-efficacy. Ding and Huang [75] suggested that mutual assistance, cooperation, and knowledge-sharing were critical for obtaining a competitive advantage. Bandura [20] suggested that individuals with high self-efficacy could have more enthusiasm, broader vision, broader thinking, and thus, more courage to accept challenges, and these behaviors were predominantly guided by the successful experiences of others. Self-efficacy and behavioral intentions could, indeed, predict food handling behaviors [30].   Relevant educational policies should support consumers in developing self-efficacy related to sustainable food handling. Previous research points to the need for self-efficacy to enhance knowledge and skills, but building self-efficacy needs to be based on actual experience [45, 56, 58,  64].

Self-efficacy could be conceptualized with different levels of linkages, ranging from general self-efficacy (a person's ability to exercise self-control in different situations) to specific self-efficacy (related to a person's ability to control behavior in specific settings (like as work)) to task-related self-efficacy (such as confidence in being able to correctly sort leftover waste) [76]. Domain- and task-related self-efficacy was positively correlated with general self-efficacy [77]. There are different levels of self-efficacy in food waste, and only by deeply understanding the root causes of food waste can we truly understand the factors of household food waste [76]. Therefore, enhancing professional competence for food-waste prevention could improve self-efficacy and help promote more sustainable practices around food preparation and surplus food. 

  1. Conclusion and Suggestions

6.1 Conclusion

According to the data analysis results, the dimensions of ethical sustainability, professional competence for food-waste prevention, self-efficacy, sustainable preparation intention, and sustainable preparation promotion were all found to be significantly and positively correlated. This study first analyzed the CFA among the various dimensions, and the results showed that this model had a good fit. All indices passed the standards, indicating a compliant fit. This model could interpret the impact of each dimension. Ethical sustainability positively affected professional competence for food-waste prevention and self-efficacy, professional competence for food-waste prevention positively affected self-efficacy and sustainable behavioral intentions during food preparation. Similarly, self-efficacy positively affected sustainable behavioral intentions during food preparation, and sustainable behavioral intentions during food preparation significantly and positively affected sustainable preparation promotion. Therefore, H1, H2, H3, H4, H5, and H6 all had empirical support. Moreover, professional competence for food-waste prevention was the factor mediating ethical sustainability to sustainable behavioral intentions during food preparation, while self-efficacy was the factor mediating professional competence for food-waste prevention to sustainable behavioral intentions during food preparation. Both H7 and H8 were supported.

6.2 Management Suggestions and Implications

This study shed light the factors influencing food waste prevention in the catering industry and could help the government and catering stakeholders to made specific recommendations for promoting food-waste prevention. The results showed that the ethical sustainability attitudes of kitchen staff affected professional competence for food-waste prevention and self-efficacy. Ethical sustainability among kitchen staff was related to their individual experiences and work/education environments. Therefore, the awareness and education provided by companies and governments concerning food-waste prevention could indirectly affect the concept of ethical sustainability among kitchen staff. Influencing the knowledge, attitudes, and skills of professional competence for food-waste prevention, and continuously promoting better strategies for surplus food, are important factors for governmental and corporate sustainability promotion. Self-efficacy indicates the individual believes that they can successfully perform a certain task or challenge, and they have the self-confidence to complete the task successfully [18, 23]. We need more consumers who are aware of and committed to food-waste prevention, and we need strong policymakers with the right strategies to promote food-waste prevention systematically. In addition, the cultivation of professional competence for food-waste prevention could affect self-efficacy and could have a significant impact on behavioral intentions through self-efficacy. For example, a company could regularly hold competitions for the creative preparation of surplus food in order to encourage self-reflection in catering practitioners [78]. Competitions or mutual internal audits of surplus food and food-waste prevention among catering practitioners could be helpful to understand limitations and share effective approaches, which could then further improve the reduction of food waste company-wide.

6.3 Practical Suggestions and Implications

It has been shown that the greater the professional competence of employees, the greater the increase in self-efficacy. Sustainability issues have become an important consideration in the catering industry in recent years. The responsibility and prevention of food waste are not only related to consumers but, rather, are closely related to the policies and factors that govern the entire supply chain. It is an encouraging sign that food-waste prevention is now a major socio-economic issue globally.

Implicit traits could be adjusted through a longer period of education and training, psychological counseling, or accumulated practical training [37]. According to our results, the level of performance depends on the level of self-efficacy and professional competence. More time must be spent on the psychological development of professional competence before effectiveness can be achieved. Related courses such as sustainability ethics, workplace ethics, and professional ethics are also indispensable components. In these courses, such topics can be discussed via case studies, peer experience sharing, and industry experience seminars for exploring individual perceptions that contribute to the improvement of self-efficacy and overall behavior motivation.

Kitchen staff are crucial for food preparation and management in the catering industry. The knowledge–attitude–behavior model [79] suggested that increased environmental knowledge could enhance environmental awareness and concern, leading to more environmentally conscious behaviors. This knowledge also generated positive thoughts, concerns, and practices [80]. In other words, the environment changed the person, and only personal changes could lead to changes in attitudes towards surplus food and more sustainable efforts when processing ingredients and preparing food. These could, then, aid the catering industry in food-waste prevention, sustainable ethics, the sustainable use of ingredients, sustainable cooking, and environmental protection.

6.4 Research Limitations and Future Studies

There were some limitations that should be considered. First, we primarily focused on those working in hotels and restaurants, as well as those working for group-meal providers, as a whole. To understand the differences in their environments, they should be considered separately. In the future, chain restaurants and central catering factories may be considered as the subjects, and different results may be obtained since other products in these catering sectors have been more standardized.

Since survey methods are used to collect data over a short period of time, the results may be somewhat biased because the research instrument was a self-report survey [81]. Therefore, future studies should design more rigorous research procedures, the data vary due to different data sources Finally, this study involved quantitative research, which could be supplemented by qualitative research in the future to understand issues associated with its implementation. We could have collected more factors affecting food waste in order to explore the actual situation of sustainable preparation behavior, such as the types of service and the ranks of kitchen staff, as well as to discuss the factors regulating sustainable preparation behavior. 

-----------------------------------------------------------------------------------------

  1. References: must be revised according to the MDPI guidelines.

Ans: I had revised all references, please see revised paper.

Reviewer 2 Report

Comments and Suggestions for Authors

Dear authors,

I read with interest your article now titled "Sustainable preparation behaviors for kitchen staff to limit food waste" and I find your presentation interesting and helpful in understanding some of the dynamics related to professional cooking and food waste. I even believe that the study should be useful for future research in the field but, in my humble opinion, the current form of your manuscript is deficient in many of the following aspects:

- definition of the objective of the study. The purpose you propose (i.e. "This study explored the perspectives of kitchen staff towards limiting food waste by evaluating the behavioral patterns such as ethical sustainability, professional competence in food-waste prevention, self-efficacy, behavioral intentions towards sustainable food preparation, and sustainable food promotion from a practical perspective.") accurate and convincing, and thus the significance of your findings seems to be lost.

- All tables are not well explained. I would suggest explaining more thoroughly the contents of the tables that introduce more information about the results.

- Figure 3 is presented as an analysis path closed between two tables and its meaning is not so clear: the link with a part of the text is not evident.

- The structure of the "Conclusion and discussion" section is strange: the discussion subsection follows the "Conclusion" subsection and other sections follow...

- The eating pattern seems not to be followed: some errors occur

Finally, the most critical point of the work is indicated. Some of the results you have presented may have already been published in a previous work or entered into an open database. I suggest you check this situation very carefully.

Therefore, as it stands, I believe this version of the paper should be heavily edited and, only after a check of the contents, could be eligible for publication in an IF (Q1) journal such as Foods.

Best regards

Author Response

To reviewers 2:

Thank you for your comments and suggestions. I appreciate the time and effort that the reviewers have dedicated to providing your valuable feedback on my manuscript. I had major revised in this paper as final attached again as reviewers commend. Please see the comments below.

The following are our responses to the Reviewers’ 2 comments:

  1. Need to definition of the objective of the study. The purpose you propose (i.e. "This study explored the perspectives of kitchen staff towards limiting food waste by evaluating the behavioral patterns such as ethical sustainability, professional competence in food-waste prevention, self-efficacy, behavioral intentions towards sustainable food preparation, and sustainable food promotion from a practical perspective.") accurate and convincing, and thus the significance of your findings seems to be lost.

Ans:  I had revised the sentences again. I also approved the findings and the model is good in this study. The abstract could found the finding.

Abstract

The concepts of culinary sustainability and avoiding the waste of surplus food have become important sustainability trends today. How to integrate the handling of surplus food in the catering industry is a topic of concern in the industry. Kitchen staff are the vital soul of any restaurant, and we intend to discuss how kitchen staff actually behave and to explore factors that influence their behaviors in order to develop an implementation model for food-waste prevention. Therefore, this study explored a model of ethical sustainability, professional competence, self-efficacy, sustainable food preparation objectives, and sustainable food promotion and behavior focused on limiting food waste. Using structural equation modeling (SEM) to understand the relationship between various constructs. This study used a questionnaire and surveyed employees who had been employed for more than 6 months in Taiwan. From May to August 2022, 500 questionnaires were distributed; 415 valid questionnaires were retrieved, yielding a 90.2% recovery rate. According to the structural equation modeling analysis between the dimensions, ethical sustainability should have had a positive influence on professional competence for food-waste prevention and self-efficacy. Professional competence for food-waste prevention affected self-efficacy and behavioral intentions during food preparation; self-efficacy also significantly affected behavioral intentions towards sustainable food preparation. Similarly, behavioral intention had a positive influence on promoting sustainable behaviors. There is a significant relationship between all constructs in this study. Professional competence for food-waste prevention was found to be the mediating factor between ethical sustainability and the behavioral intentions towards sustainable food preparation, and self-efficacy was the mediating factor between the professional competence for food-waste prevention and the behavioral intentions towards sustainable food preparation.

---------------------------------------------------------------------------

  1. All tables are not well explained. I would suggest explaining more thoroughly the contents of the tables that introduce more information about the results.

Ans: I had added some statement in all table. Please see the revised paper.

-------------------------------------------------------------------------------------

  1. Figure 3 is presented as an analysis path closed between two tables and its meaning is not so clear: the link with a part of the text is not evident.

Ans : I had revised again, please see as below.

Figure 3. Structural model for path analysis of ethical sustainability, professional competence for food-waste prevention, self-efficacy and sustainable behavioral intentions during food preparation

  1. The structure of the "Conclusion and discussion" section is strange: the discussion subsection follows the "Conclusion" subsection and other sections follow...

Ans:  I had revised as below.

  1. Discussion

Ethical sustainability positively affected professional competence for food-waste prevention and self-efficacy, while professional competence for food-waste prevention was the mediating factor between ethical sustainability and sustainable behavioral intentions during food preparation. Edwards and Mercer [70] strongly suggested that lifestyle choices, such as diet and career choices, had an ethical basis. Waston and Meah [32] explained that the primary driver for lower food waste was an “ethic of thrift”, which was the ethical motivation to act appropriately and “be thrifty”. The ethical rule for food waste behavior was "don't waste food". Therefore, the core ethical commitment in the study was to avoid wasting food, thus making consumption more reasonable [39]. Therefore, kitchen staff must have a personal imperative to prevent food waste in order to establish relevant behaviors and build professional competence in that arena.

Based on social cognitive theory in the context of food waste, females were more cognitively advanced than males. Females more closely monitored reducing food waste, so they were more motivated [71]. This contributed to their identification of more motivational factors that could have had a positive effect on their behavior. The environment affected the individuals and, in turn, their behavior. If actual catering producers were more aware of food waste, it could help enhance their professional competence for food-waste prevention.

The research on food waste has shown that preventing food waste required different culinary skills and knowledge, so leftovers food could be re-purposed [72, 73]. During storage, leftovers were often misplaced or left in the refrigerator for too long and spoiled [14, 15, 17]. Therefore, professional competence in food preparation is important.

 Professional competence for food-waste prevention positively affected self-efficacy and sustainable behavioral intentions during food preparation. Self-efficacy was the mediating factor between professional competence for food-waste prevention and sustainable behavioral intentions during food preparation. Lin [74] defined knowledge-sharing as a culture of social interaction, and the composition of this culture included knowledge exchange, experiential inheritance, and mutual learning among employees because the growth of knowledge could increase self-efficacy. Ding and Huang [75] suggested that mutual assistance, cooperation, and knowledge-sharing were critical for obtaining a competitive advantage. Bandura [20] suggested that individuals with high self-efficacy could have more enthusiasm, broader vision, broader thinking, and thus, more courage to accept challenges, and these behaviors were predominantly guided by the successful experiences of others. Self-efficacy and behavioral intentions could, indeed, predict food handling behaviors [30].   Relevant educational policies should support consumers in developing self-efficacy related to sustainable food handling. Previous research points to the need for self-efficacy to enhance knowledge and skills, but building self-efficacy needs to be based on actual experience [45, 56, 58,  64].

Self-efficacy could be conceptualized with different levels of linkages, ranging from general self-efficacy (a person's ability to exercise self-control in different situations) to specific self-efficacy (related to a person's ability to control behavior in specific settings (like as work)) to task-related self-efficacy (such as confidence in being able to correctly sort leftover waste) [76]. Domain- and task-related self-efficacy was positively correlated with general self-efficacy [77]. There are different levels of self-efficacy in food waste, and only by deeply understanding the root causes of food waste can we truly understand the factors of household food waste [76]. Therefore, enhancing professional competence for food-waste prevention could improve self-efficacy and help promote more sustainable practices around food preparation and surplus food. 

  1. Conclusion and Suggestions

6.1 Conclusion

According to the data analysis results, the dimensions of ethical sustainability, professional competence for food-waste prevention, self-efficacy, sustainable preparation intention, and sustainable preparation promotion were all found to be significantly and positively correlated. This study first analyzed the CFA among the various dimensions, and the results showed that this model had a good fit. All indices passed the standards, indicating a compliant fit. This model could interpret the impact of each dimension. Ethical sustainability positively affected professional competence for food-waste prevention and self-efficacy, professional competence for food-waste prevention positively affected self-efficacy and sustainable behavioral intentions during food preparation. Similarly, self-efficacy positively affected sustainable behavioral intentions during food preparation, and sustainable behavioral intentions during food preparation significantly and positively affected sustainable preparation promotion. Therefore, H1, H2, H3, H4, H5, and H6 all had empirical support. Moreover, professional competence for food-waste prevention was the factor mediating ethical sustainability to sustainable behavioral intentions during food preparation, while self-efficacy was the factor mediating professional competence for food-waste prevention to sustainable behavioral intentions during food preparation. Both H7 and H8 were supported.

6.2 Management Suggestions and Implications

This study shed light the factors influencing food waste prevention in the catering industry and could help the government and catering stakeholders to made specific recommendations for promoting food-waste prevention. The results showed that the ethical sustainability attitudes of kitchen staff affected professional competence for food-waste prevention and self-efficacy. Ethical sustainability among kitchen staff was related to their individual experiences and work/education environments. Therefore, the awareness and education provided by companies and governments concerning food-waste prevention could indirectly affect the concept of ethical sustainability among kitchen staff. Influencing the knowledge, attitudes, and skills of professional competence for food-waste prevention, and continuously promoting better strategies for surplus food, are important factors for governmental and corporate sustainability promotion. Self-efficacy indicates the individual believes that they can successfully perform a certain task or challenge, and they have the self-confidence to complete the task successfully [18, 23]. We need more consumers who are aware of and committed to food-waste prevention, and we need strong policymakers with the right strategies to promote food-waste prevention systematically. In addition, the cultivation of professional competence for food-waste prevention could affect self-efficacy and could have a significant impact on behavioral intentions through self-efficacy. For example, a company could regularly hold competitions for the creative preparation of surplus food in order to encourage self-reflection in catering practitioners [78]. Competitions or mutual internal audits of surplus food and food-waste prevention among catering practitioners could be helpful to understand limitations and share effective approaches, which could then further improve the reduction of food waste company-wide.

6.3 Practical Suggestions and Implications

It has been shown that the greater the professional competence of employees, the greater the increase in self-efficacy. Sustainability issues have become an important consideration in the catering industry in recent years. The responsibility and prevention of food waste are not only related to consumers but, rather, are closely related to the policies and factors that govern the entire supply chain. It is an encouraging sign that food-waste prevention is now a major socio-economic issue globally.

Implicit traits could be adjusted through a longer period of education and training, psychological counseling, or accumulated practical training [37]. According to our results, the level of performance depends on the level of self-efficacy and professional competence. More time must be spent on the psychological development of professional competence before effectiveness can be achieved. Related courses such as sustainability ethics, workplace ethics, and professional ethics are also indispensable components. In these courses, such topics can be discussed via case studies, peer experience sharing, and industry experience seminars for exploring individual perceptions that contribute to the improvement of self-efficacy and overall behavior motivation.

Kitchen staff are crucial for food preparation and management in the catering industry. The knowledge–attitude–behavior model [79] suggested that increased environmental knowledge could enhance environmental awareness and concern, leading to more environmentally conscious behaviors. This knowledge also generated positive thoughts, concerns, and practices [80]. In other words, the environment changed the person, and only personal changes could lead to changes in attitudes towards surplus food and more sustainable efforts when processing ingredients and preparing food. These could, then, aid the catering industry in food-waste prevention, sustainable ethics, the sustainable use of ingredients, sustainable cooking, and environmental protection.

6.4 Research Limitations and Future Studies

There were some limitations that should be considered. First, we primarily focused on those working in hotels and restaurants, as well as those working for group-meal providers, as a whole. To understand the differences in their environments, they should be considered separately. In the future, chain restaurants and central catering factories may be considered as the subjects, and different results may be obtained since other products in these catering sectors have been more standardized.

Since survey methods are used to collect data over a short period of time, the results may be somewhat biased because the research instrument was a self-report survey [81]. Therefore, future studies should design more rigorous research procedures, the data vary due to different data sources Finally, this study involved quantitative research, which could be supplemented by qualitative research in the future to understand issues associated with its implementation. We could have collected more factors affecting food waste in order to explore the actual situation of sustainable preparation behavior, such as the types of service and the ranks of kitchen staff, as well as to discuss the factors regulating sustainable preparation behavior. 

-----------------------------------------------------------------------------------------

  1. Finally, the most critical point of the work is indicated. Some of the results you have presented may have already been published in a previous work or entered into an open database. I suggest you check this situation very carefully.

Ans: I had revised and careful check all texts, and I use Turnitin to check. The similar results as 7%, and below 1% for each reference.
